# Modulating mitofusins to control mitochondrial function and signaling

Emmanouil Zacharioudakis[1,2,3,4,5], Bogos Agianian [1,2,3,4,5], Vasantha Kumar MV[1,2,3,4,5], Nikolaos Biris [1,2,3,4,5], Thomas P. Garner[1,2,3,4,5], Inna Rabinovich-Nikitin[6,7,8], Amanda T. Ouchida[1,2,3,4,5], Victoria Margulets[6,7,8], Lars Ulrik Nordstrøm[1], Joel S. Riley [9,10], Igor Dolgalev [11,12,13], Yun Chen[2,3,4,14,15], Andre J. H. Wittig[2,3,4,14,15], Ryan Pekson [2,3,4,14,15], Chris Mathew[1], Peter Wei[1], Aristotelis Tsirigos[11,12,13], Stephen W. G. Tait [9,10], Lorrie A. Kirshenbaum[6,7,8], Richard N. Kitsis [2,3,4,14,15] & Evripidis Gavathiotis [1,2,3,4,5✉]

Mitofusins reside on the outer mitochondrial membrane and regulate mitochondrial fusion, a physiological process that impacts diverse cellular processes. Mitofusins are activated by conformational changes and subsequently oligomerize to enable mitochondrial fusion. Here, we identify small molecules that directly increase or inhibit mitofusins activity by modulating mitofusin conformations and oligomerization. We use these small molecules to better understand the role of mitofusins activity in mitochondrial fusion, function, and signaling. We find that mitofusin activation increases, whereas mitofusin inhibition decreases mitochondrial fusion and functionality. Remarkably, mitofusin inhibition also induces minority mitochondrial outer membrane permeabilization followed by sub-lethal caspase-3/7 activation, which in turn induces DNA damage and upregulates DNA damage response genes. In this context, apoptotic death induced by a second mitochondria-derived activator of caspases (SMAC) mimetic is potentiated by mitofusin inhibition. These data provide mechanistic insights into the function and regulation of mitofusins as well as small molecules to pharmacologically target mitofusins.

[1] Department of Biochemistry, Albert Einstein College of Medicine, Bronx, NY, USA. [2] Department of Medicine, Albert Einstein College of Medicine, Bronx, NY, USA. [3] Wilf Family Cardiovascular Research Institute, Albert Einstein College of Medicine, Bronx, NY, USA. [4] Albert Einstein Cancer Center, Albert Einstein College of Medicine, Bronx, NY, USA. [5] Institute for Aging Research, Albert Einstein College of Medicine, Bronx, NY, USA. [6] Department of Physiology and Pathophysiology, Max Rady College of Medicine, Faculty of Health Sciences, University of Manitoba, Winnipeg, MB, Canada. [7] Department of Pharmacology and Therapeutics, Max Rady College of Medicine, Faculty of Health Sciences, University of Manitoba, Winnipeg, MB, Canada. [8] Institute of Cardiovascular Sciences, St. Boniface Research Centre, Winnipeg, MB, Canada. [9] Cancer Research UK Beatson Institute, Glasgow, UK. [10] Institute of Cancer Sciences, University of Glasgow, Glasgow, UK. [11] Department of Pathology, New York University School of Medicine, New York, NY, USA. [12] Laura & Isaac Perlmutter Cancer Center, New York University School of Medicine, New York, NY, USA. [13] Applied Bioinformatics Laboratories, New York University School of Medicine, New York, NY, USA. [14] Department of Cell Biology, Albert Einstein College of Medicine, Bronx, NY, USA. [15] Einstein-Mount Sinai Diabetes Research Center, Albert Einstein College of Medicine, Bronx, NY, USA. ✉email: evripidis.gavathiotis@einsteinmed.edu

Mitochondria fuse, divide, and interact with other organelle structures to regulate cellular fitness and fate while they produce the majority of energy to sustain cellular activity[1,2]. They are highly dynamic organelles constantly undergoing the physiological process of fusion and fission which regulates mitochondrial morphology and dynamics[3]. Among different cell types or within the same type of cells, mitochondria morphology varies among small spheres, short rods, or long tubules[4]. This dynamism allows mitochondria to exchange components (e.g. lipid membranes, proteins), promote repair and removal of defective mitochondria, thus maintaining mitochondrial function and quality[5]. Furthermore, mitochondrial fusion and fission enables mitochondria to meet cellular energy demands in response to environmental stimuli[2,6]. Fused mitochondria often lead to increased, oxidative phosphorylation and mitochondrial membrane potential[7]. In contrast, fragmented mitochondria often correlate with reduced function, decreased mitochondrial membrane potential, and oxidative phosphorylation[7,8].

Mitochondrial fusion is a two step process; the first step requires the fusion of the outer mitochondrial membrane (OMM), which is mediated by mitofusin-1 (MFN1) and mitofusin-2 (MFN2)[4,9–12]. The second step requires the fusion of the inner mitochondrial membrane, which is mediated by optic atrophy-1 (OPA1)[13,14]. Loss of either MFN1/2 or OPA1 proteins results in a network of hyper-fragmented mitochondria[8,14]. While both MFN1 and MFN2 deficient cells display a clearly fragmented mitochondrial network, MFN1 knockout cells display severe mitochondrial fragmentation with the formation of small spheres of similar size and MFN2 knockout cells display mitochondrial spheres or ovals of variable but larger size[4,11,15]. Overexpression of either MFN1 or MFN2 in wild type cells leads to extensive mitochondrial clustering in the perinuclear area[9,15,16]. However, overexpression of MFN1 in MFN2 knockout cells and MFN2 in MFN1 knockout cells restores mitochondrial fusion highlighting the degree of redundancy between the MFN1/2 proteins[4].

Biochemical and structural studies have provided insights into how MFN1/2 accomplish mitochondrial tethering, although the exact mechanism behind this process remains elusive[17–20]. MFN1 and MFN2 share high sequence homology, possessing an N-terminal GTPase domain, a coiled-coiled heptad repeat (HR1) domain, a short transmembrane domain responsible for anchoring MFNs on the OMM, and a second coiled-coiled heptad repeat (HR2) domain located in the C-terminus[17,18]. Homo- or hetero-dimerization of MFN1/2 mediates the tethering of the OMMs of adjacent mitochondria[21,22]. Interestingly, Koshiba et al. reported that dimers of the HR2 domain of MFN1 can be formed in vitro, in which the HR2 domains interact with each other in an anti-parallel fashion, suggesting that this interaction plays an important role in the dimerization of MFN1/2 from adjacent mitochondria[18,19,21]. Consistent with this model, we previously showed that MFN2 can adopt an anti-tethering or a pro-tethering conformation, in the latter the HR2 domain is exposed in the cytoplasm where it can interact in an anti-parallel manner with another exposed HR2 domain from adjacent mitochondria and promote MFN1/2 oligomerization[18,19]. Others proposed that MFN1/2 oligomerization occurs through the trans dimerization of the GTPase domains[17,23]. These models are not mutually exclusive and it is likely that MFN1/2 oligomerization requires the intermolecular interactions of both GTPase and HR2 domains. Furthermore, an alternative model for mitochondrial tethering proposed that MFN1/2 oligomerization is triggered by the dimerization of the HR2 domains in the intermembrane mitochondrial space prior to the oligomerization of their corresponding GTPase domains[24].

In this work, we aim to identify small molecules that directly and specifically modulate MFN1/2 fusogenic activity and better understand the function of MFN1/2 and their effects in mitochondrial dynamics and signaling. Using structural and biochemical insights for direct modulation of MFN1/2 conformations, we develop rational pharmacophore approaches to perform in silico screening of small molecules. These screening strategies provide us with small molecules capable of activating or inhibiting the fusogenic activity of MFN1/2 by manipulating the MFN1/2 tethering permissive structure. Using these molecules as molecular tools we set out to investigate the direct role of MFN1/2 in mitochondrial fusion and regulation of other biological processes related to mitochondrial morphology such as metabolic activity and apoptotic priming. Our study yields mechanistic insights about the role of MFN1/2 in mitochondrial function and apoptotic signaling, as well as a rational strategy to pharmacologically target mitofusins using small molecules.

## Results

**A small molecule activator of mitochondrial fusion via mitofusins.** We previously generated structural models of full length MFN2 in the anti-tethering and pro-tethering conformation that are consistent with the crystal structures of the MFN1 anti-tethering conformation and HR2–HR2 interactions (Fig. 1a)[17–19]. This informed us about the intramolecular HR1–HR2 interactions that occur in the anti-tethering conformation and the intermolecular HR2–HR2 interactions that occur in the pro-tethering conformation (Fig. 1a). Previously, a helical peptide (367–384Gly) from the MFN2 HR1 residues 367–384 was found to promote the pro-tethering conformation of MFN2 and mitochondrial fusion in cells[18]. Therefore, we envisioned that a small molecule-based peptidomimetic of 367–384Gly would modulate the interconversion of anti- and pro-tethering conformations towards the pro-tethering conformation and subsequently promote mitochondrial fusion (Fig. 1b). Visual inspection of the intramolecular HR1–HR2 interactions in the anti-tethering conformation of the human MFN2, in the 367–384 region, identified several hydrophobic interactions between the following HR1-amino acids: Val372, Met376, and HR2-amino acids: Leu724, Leu727, and Ala731 and a possible hydrogen bond between the HR1-amino acid: His380 and HR2-amino acid: Asp725 (Fig. 1c). Furthermore, molecular dynamics simulations of the MFN2 structural model showed that the aforementioned HR2 residues present weak spatial fluctuations over time (Supplementary Figs. 1, 2a–c). Additionally, these residues are conserved among different species of MFN2 and MFN1 (Supplementary Fig. 3a).

To identify a small molecule activator of MFN2, we generated an in silico pharmacophore model to screen small molecules that can mimic the aforementioned HR1 residues and bind to the corresponding interacting HR2 residues (Fig. 1c, d). The in silico pharmacophore model comprises the following features: three hydrophobic groups, an aromatic ring and a hydrogen bond donor based on the side chains of the HR1 residues: Val372, Met376, and His380 (Fig. 1d). An in silico library of $13.8 \times 10^6$ commercially available small molecules was screened to fit the pharmacophore model using PHASE software (Supplementary Fig. 3b)[25]. The top 1000 hits of the in silico screen were clustered for diversity and analyzed based on their interactions with residues of the HR2 domain and molecular properties. Moreover, a number of filters (e.g., elimination of hits with poor ADME properties) were applied to provide small molecules with drug-like properties[26]. Finally, a subset of 18 putative Mitofusin Activator Small Molecules (MASMs) was selected for

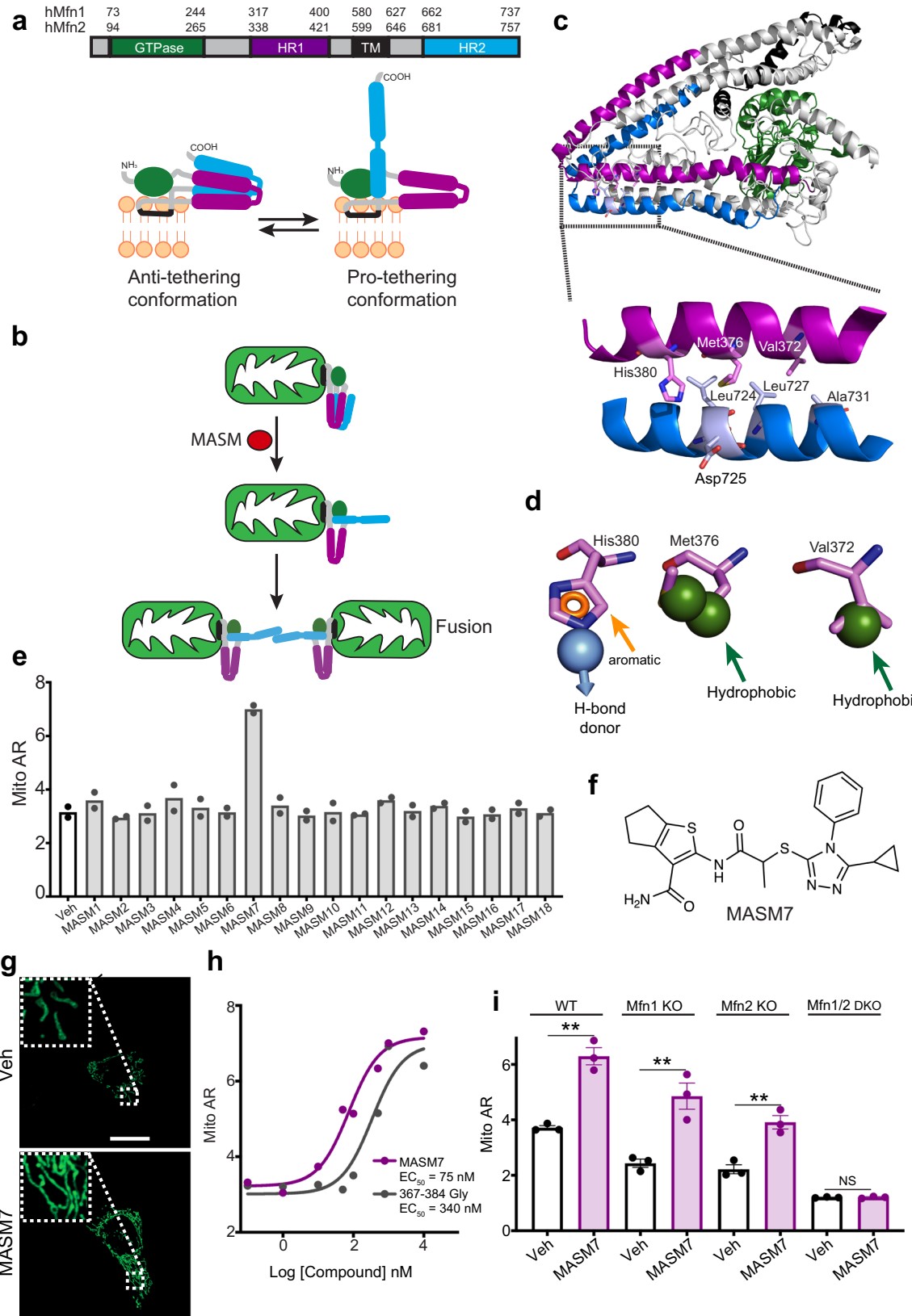

experimental validation based on their fit to the pharmacophore model and molecular diversity of the scaffolds.

Selected hits were evaluated for their capacity to promote mitochondrial fusion in cells using mitochondrial aspect ratio (mitochondrial length/width) as a readout (Fig. 1e). From this set of compounds, MASM7 increased mitochondrial aspect ratio and

emerged as the most effective compound in promoting mitochondrial fusion (Fig. 1e–g). Interestingly, titration of MASM7 showed more potent induction of mitochondrial fusion in cells than the 367–384Gly peptide (Fig. 1h). MASM7 possesses functional groups that fulfill the five criteria of the pharmacophore model (Supplementary Fig. 3c). To validate this, we

**Fig. 1 A small molecule activator of mitofusins. a** Domain model of MFN2 showing GTPase (green), transmembrane region (black), HR1 (purple), and HR2 (blue). Amino acid numbers corresponding to human MFN1 and MFN2. Cartoon depiction of MFN2 structure showing anti-tethering (closed) conformation and pro-tethering (open) conformation[18]. **b** Cartoon depiction of conformational activation of MFNs and fusion of the OMM. MASMs promote the pro-tethering conformation of MFNs and subsequently mitochondrial fusion. **c** Ribbon presentation of the human full-length MFN2 structural model highlighting specific interactions between the HR1 (purple) and HR2 (blue) helical segments. **d** Pharmacophore hypothesis based on the sidechains of the HR1-amino acids: Val372, Met376, His380 interacting with HR2 as in (**c**) comprising of three hydrophobic points, one aromatic ring, and one hydrogen bond donor. **e** Screening of putative MASMs in cells using mitochondrial aspect ratio (Mito AR) as readout. Cells were treated with MASMs (1 μM, 2 h). Data represent mean of two independent biological replicates. **f** Chemical structure of MASM7. **g** Confocal micrographs of MEFs treated with MASM7 (1 μM, 2 h). Mitochondria were stained with Mitotracker green. Scale bar 20 μm. Micrograph is representative of $n = 3$ independent experiments. **h** MASM7 and 367–384Gly concentration-responsively increased Mito AR in MEFs. Cells were treated with MASM7 or 367–384 Gly at the indicated concentrations for 2 h. Data represent mean of two independent biological replicates. **i** Quantification of Mito AR of WT, *Mfn1* KO, *Mfn2* KO, and *Mfn1/Mfn2* DKO MEFs treated with MASM7 (1 μM, 2 h). Data represent mean ± SEM of three independent biological replicates. Statistics were obtained using two-tailed unpaired t-test: *$p < 0.05$, **$p < 0.01$, ***$p < 0.001$, ****$p < 0.0001$. Source data are provided as a Source Data file.

investigated structure activity relationships around the MASM7 scaffold. A series of MASM7 analogs each of which harbors alterations in the aforementioned functional groups were evaluated for their capacity to promote mitochondrial fusion in cells (Supplementary Fig. 3d, e). Loss of the thiophene ring diminished the ability of MASM19 to promote mitochondrial fusion. Expansion of the ring from cyclopentane to cyclohexane slightly decreased the capacity of MASM20 to promote fusion. Moreover, the replacement of the methyl group of MASM7 with hydrogen decreased drastically the capacity of MASM21 to promote fusion. Interestingly, replacement of the cyclopropane group of MASM7 with bulky substituents (e.g., benzene) was detrimental for the activity of MASM22, while replacements with less bulky substituents (e.g., furan group or methyl-furan) were better tolerated in the cases of MASM23 and MASM24. Taken all together these data suggest that MASM7 possesses functional groups that satisfy the pharmacophore model to interact with mitofusins and promote mitochondrial fusion.

Next we sought to determine whether MASM7 altered mitochondrial dynamics in MFN1/2 dependent manner. Knockout of *Mfn1* or *Mfn2* reduced the capacity of MASM7 to promote mitochondrial fusion in cells compared to WT cells (Fig. 1i and Supplementary Fig. 3f). Moreover, knockout of both *Mfn1* and *Mfn2* completely abolished the capacity of MASM7 to promote mitochondrial fusion in cells (Fig. 1i and Supplementary Fig. 3f). Collectively, these data suggest that MASM7 can promote mitochondrial fusion by directly activating either MFN2 or MFN1 and its activity is greater when both MFN2 and MFN1 are expressed. This result is in agreement with the high sequence homology between MFN1 and MFN2 and our pharmacophore model for MFN2 activation based on conserved residues that regulate HR1-HR2 interactions (Supplementary Fig. 3a). Notably, MASM7 was capable of promoting mitochondrial fusion in *Drp1* KO MEFs excluding the possibility of inhibiting DRP1 and fission as a potential mechanism for its activity to promote fusion (Supplementary Fig. 3g)[27]. Furthermore, molecular docking and molecular dynamic simulations of MASM7 or MASM22 to the full-length MFN2 structural model showed that MASM7 but not MASM22 could form and retain the interactions with the predicted residues of the HR2 domain during the simulation time (Supplementary Fig. 4a, b, d). Consistently, MASM22 had a higher RMSD than MASM7 during the simulation time (Supplementary Fig. 4e). This data come in agreement with our SAR studies around the MASM7 scaffold and validate our pharmacophore model hypothesis (Fig. 1c, d and Supplementary Fig. 3c).

**A small molecule inhibitor of mitochondrial fusion via mitofusins.** Intermolecular interactions of the HR2 domains of

MFN1/2 from adjacent mitochondria mediate MFN1/2 oligomerization[18,19]. We, therefore, posited that a small molecule capable of inhibiting such intermolecular HR2–HR2 interactions could act as an inhibitor of MFN1/2 oligomerization and subsequently mitochondrial fusion (Supplementary Fig. 5a). Moreover, previously a helical peptide (398–418Gly) from the MFN2 HR1 residues 398–418 was found to inhibit mitochondrial fusion in cells by inhibiting HR2–HR2 intermolecular interactions and subsequently MFN1/2 oligomerization[18]. This prompted us to visually inspect the interactions of the HR1-residues 398–418 with the HR2 residues to gain structural insights for small molecule mimicry (Fig. 2a). Specifically, hydrophobic interactions were observed between the HR1-residues: Leu408, Ala412, and the HR2-residues: Leu692, Val688, and a possible hydrogen bond between the HR1-amino acid: Tyr415 and HR2-amino acid: Ser685 (Fig. 2a). Furthermore, molecular dynamics simulations of the MFN2 structural model showed that the aforementioned HR2 residues present weak spatial fluctuations over time (Supplementary Fig. 2d–f). Of note, these HR1 residues are conserved among different species of MFN2 and MFN1 (Supplementary Fig. 3a). Hence, we envisioned that a small molecule capable of recapitulating the interactions of the aforementioned HR1-amino acids would compete with intermolecular HR2–HR2 interactions and inhibit MFN1/2 oligomerization.

To identify such a small molecule we used a similar strategy as with the discovery of MASM7 (Supplementary Fig. 3b). We generated an in silico pharmacophore model to screen small molecules that would specifically mimic the side chains of the HR1 residues: Leu408, Ala412, and Tyr415 and bind to the corresponding HR2 residues (Fig. 2a and Supplementary Fig. 2d). Our pharmacophore model includes two hydrophobic interactions, an aromatic ring, and a hydrogen bond donor/acceptor (Fig. 2b). Next, we screened the in silico library of $13.8 \times 10^6$ commercially available small molecules using the same strategy and selection criteria as with MASMs (Supplementary Fig. 3b). Likewise a set of 21 putative Mitochondrial Fusion Inhibitors (MFIs) was selected for experimental validation based on their fit to the pharmacophore model and the molecular diversity of their scaffolds.

Selected hits were screened for their capacity to inhibit mitochondrial fusion in cells by monitoring mitochondrial morphology and using mitochondrial aspect ratio as a readout (Fig. 2c). Strikingly, MFI8 reduced significantly mitochondrial aspect ratio and emerged as the most effective compound in inhibiting mitochondrial fusion and subsequently promoting mitochondrial fission (Fig. 2d, e). Titration of MFI8 showed a comparable concentration-dependent reduction of the mitochondrial aspect ratio to that of 398–418 Gly peptide (Fig. 2f). MFI8 has a small structure but possesses functional groups that could fulfill the four criteria of the pharmacophore model used

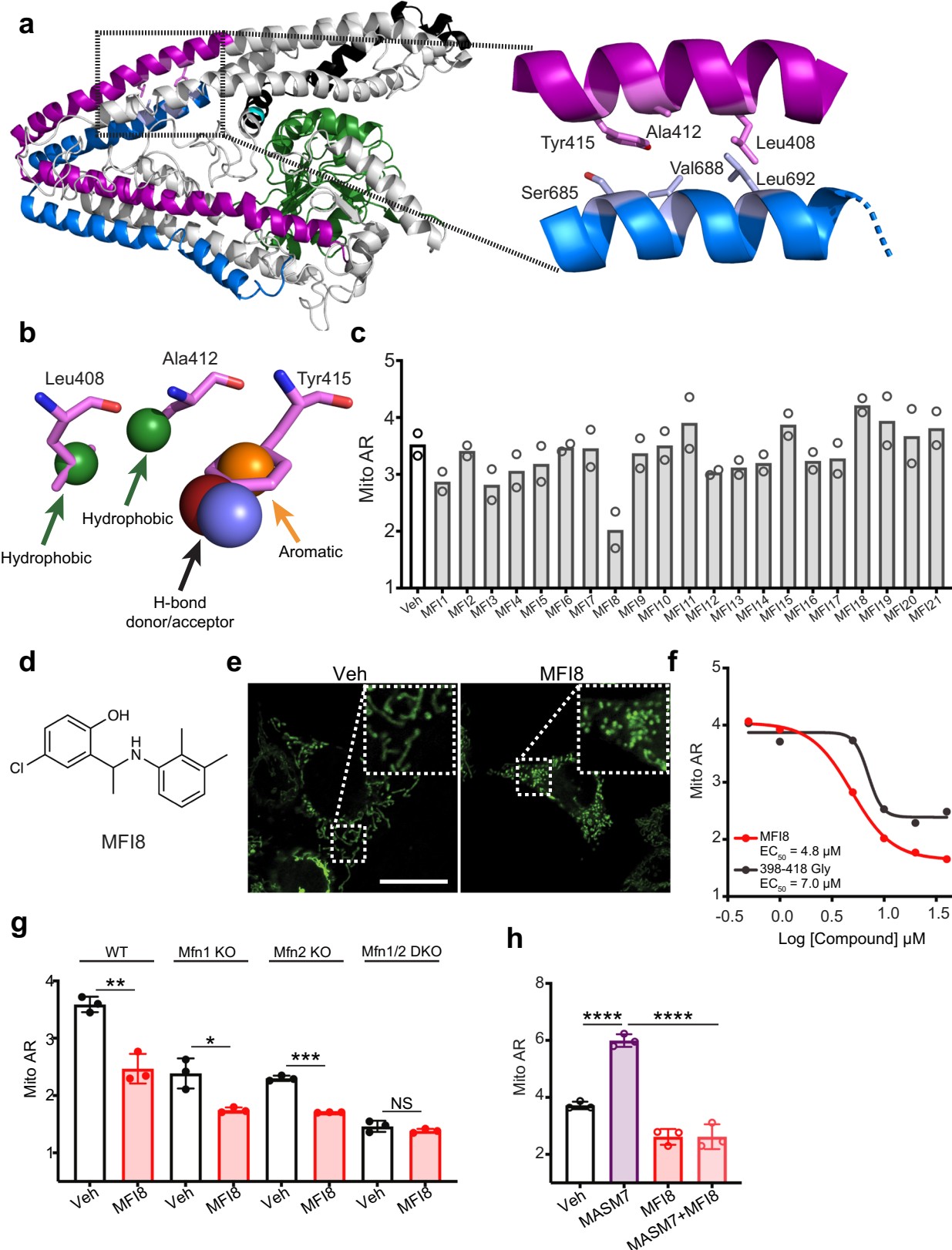

for the in silico screen (Supplementary Fig. 5b). To further validate that MFI8 meets the four criteria of the pharmacophore model hypothesis, we investigated structure activity relationships around the MFI8 scaffold. A series of MFI8 analogs were generated and evaluated for their capacity to promote mitochondrial fragmentation in cells (Supplementary Fig. 5c, d).

Substitution of the chlorine with hydrogen in the aromatic ring of MFI22 significantly reduced the capacity of the compound to promote mitochondrial fragmentation, suggesting that loss of chlorine affects the electron density of the aromatic ring and impairs its interaction. Loss of the methyl group in the aliphatic chain that connects the two phenyl rings had a minimal effect on

**Fig. 2 A small molecule inhibitor of mitofusins. a** Ribbon presentation of the human full-length MFN2 structural model highlighting specific interactions between the HR1 (purple) and HR2 (blue) helical segments. **b** Pharmacophore hypothesis based on the sidechains of the HR1-amino acids: Leu408, Ala412, Tyr415 interacting with HR2 as in (**a**) comprising 2 hydrophobic points, one aromatic ring, and one hydrogen bond donor or acceptor. **c** Screening of putative MFIs using Mito AR as readout. Cells were treated with MFIs (10 μM, 6 h). Data represent mean of two independent biological replicates. **d** Chemical structure of MFI8. **e** Confocal micrographs of MEFs treated with MFI8 (20 μM, 6 h). Mitochondria were stained with Mitotracker green. Scale bar 20 μm. Micrograph is representative of $n = 3$ independent experiments. **f** MFI8 and 398–418Gly concentration-responsively decreased Mito AR in MEFs. Cells were treated with MFI8 or 398–418Gly at the indicated concentrations for 6 h. Data represent mean of two independent biological replicates. **g** Quantification of Mito AR of WT, *Mfn1* KO, *Mfn2* KO, and *Mfn1/Mfn2* DKO MEFs treated with MFI8 (20 μM, 6 h). Data represent mean ± SEM of three independent biological replicates. Statistics were obtained using two-tailed unpaired t-test: \*$p < 0.05$, \*\*$p < 0.01$, \*\*\*$p < 0.001$, \*\*\*\*$p < 0.0001$. **h** Quantification of Mito AR of MEFs treated with MASM7 (1 μM), MFI8 (20 μM), and the combination of MASM7 and MFI8 for 6 h. Data represent mean ± SEM of three independent biological replicates. Statistics were obtained using one way ANOVA: \*$p < 0.05$, \*\*$p < 0.01$, \*\*\*$p < 0.001$, \*\*\*\*$p < 0.0001$. Source data are provided as a Source Data file.

the activity of MFI23. Furthermore, loss of each methyl group in the phenyl ring of MFI8 was not tolerated as MFI24 and MFI25 mildly reduced mitochondrial aspect ratio, suggesting the importance of those two methyl groups in mediating hydrophobic contacts for MFI8. Importantly, loss of the hydroxyl group of the phenolic ring was dentrimental for the activity of MFI26, suggesting that MFI8 forms a crucial hydrogen bond with the HR2 domain. Taken together, these data suggested that MFI8 possesses functional groups that meet the criteria of the pharmacophore model to interact with mitofusins and promote mitochondrial fission.

Next we investigated whether MFI8 inhibited mitochondrial fusion in a MFN1/2 dependent manner using mitochondrial aspect ratio as a readout. MFI8 was still capable of reducing mitochondrial aspect ratio when either *Mfn1* or *Mfn2* was knocked out (Fig. 2g and Supplementary Fig. 5e). In contrast, double knockout of *Mfn1* and *Mfn2* completely abolished MFI8 from reducing mitochondrial aspect ratio (Fig. 2g and Supplementary Fig. 5e). Collectively, these data indicate that MFI8 can promote mitochondrial fission by inhibiting mitochondrial fusion and interfering with the formation of either homotypic or heterotypic MFN1/2 complexes. Importantly, MFI8 reduced mitochondrial aspect ratio even when it was co-treated with MASM7 in MEFs (Fig. 2h and Supplementary Fig. 5f). Such result supports the idea that MFI8 operates on the tethering permissive structure of MFNs and inhibits MFNs oligomerization by reducing the HR2–HR2 intermolecular interactions. Moreover, molecular docking and molecular dynamic simulations of MFI8 or MFI25 to the full-length MFN2 structural model showed that MFI8 but not MFI25 could form and retain the interactions with the predicted residues of the HR2 domain during the simulation time (Supplementary Fig. 6a, b, d). Consistently, MFI25 had a higher RMSD than MFI8 during the simulation time (Supplementary Fig. 6e). This data come in agreement with our SAR studies around the MFI8 scaffold and validate our pharmacophore model hypothesis (Fig. 2a, b and Supplementary Fig. 5b).

**MASM7 and MFI8 bind to the HR2 domain of MFN2.** To confirm that MASM7 and MFI8 directly interact with the HR2 domain, we produced MFN2-HR2 domain (residues 678–757) and evaluated their interaction in comparison with 367–384Gly and 398–418Gly peptides using microscale thermophoresis (MST). MASM7 and MFI8 demonstrated direct binding to the HR2 domain of MFN2 with Kds in the low micrormolar range and that was comparable to the binding interactions of 367–384Gly and 398–418Gly peptides, respectively (Fig. 3a). Furthermore, MASM23 and MFI23 demonstrated direct binding to the HR2 domain of MFN2 (Supplementary Fig. 7), in line with our previous results that showed that these small molecules are capable of increasing or decreasing the mitochondrial aspect ratio, respectively (Supplementary Figs. 3e and 5d). In contrast,

MASM19, MASM21, MASM22, MFI22, MFI25, and MFI26 did not demonstrate measurable binding to the HR2 domain of MFN2 (Supplementary Fig. 7), in agreement with the inability or weak activity of these compounds to promote mitochondrial fusion or fission, respectively (Supplementary Figs. 3e and 5d).

We also used an $^{15}$N-labeled MFN2-HR2 domain to record heteronuclear single quantum coherence (HSQC) spectra with or without MASM7 or MFI8 and evaluated their binding interaction to MFN2-HR2 in comparison with the 367–384Gly and 398–418Gly peptides. HSQC NMR spectra of the recombinant MFN2-HR2 domain showed evidence of a folded conformation. Titration of MASM7 induced peak broadening and shifting of select cross peaks of HR2 residues in the HSQC spectra, demonstrating direct binding to the HR2 domain (Supplementary Fig. 8a). The same cross peaks broadened and shifted upon titration of the 367–384Gly peptide, indicating that both MASM7 and 367–384Gly interact with similar HR2 residues (Supplementary Fig. 8b). To further confirm that the interaction of MASM7 is specific to the expected binding region of HR2, a double mutant of HR2, D725A/L727A, was generated that should disrupt binding of MASM7 based on its proposed interaction with the HR2 domain (Fig. 1c). Titration of MASM7 to an $^{15}$N-labeled MFN2-HR2 D725A/L727A mutant under the same conditions showed a lack of peak shifts or peak broadening indicating the inability of MASM7 to bind to the MFN2-HR2 D725A/L727A mutant (Supplementary Fig. 8c).

Titration of MFI8 to the $^{15}$N-labeled MFN2-HR2 revealed peak broadening and shifting of select cross peaks of HR2 residues in the HSQC spectra, showing that MFI8 directly interacts with the HR2 domain of MFN2 (Supplementary Fig. 9a). Additionally, the same cross peaks shifted upon titration of the 398–418Gly peptide, indicating interaction with similar HR2 residues for both MFI8 and 398–418Gly (Supplementary Fig. 9b). Notably, titration of MFI8 to $^{15}$N-labeled MFN2-HR2 S685A mutant under the same conditions showed a lack of peak shifts or peak broadening indicating the inability of MFI8 to bind to the MFN2-HR2 S685A mutant (Supplementary Fig. 9c).

Next we sought to determine whether MASM7 or MFI8 also interact with the HR2 domain of MFN2 in cells. MASM7 markedly increased mitochondrial aspect ratio in *Mfn1/Mfn2* DKO MEFs when reconstituted with WT *Mfn2*, but not when reconstituted with D725A/L727A or L727A *Mfn2* mutants, underscoring that MASM7 specifically targets the HR2 domain of MFN2 in cells (Fig. 3b and Supplementary Fig. 10a, c). Likewise, MFI8 significantly decreased mitochondrial aspect ratio in *Mfn1/Mfn2* DKO MEFs when reconstituted with WT *Mfn2*, but not when reconstituted with S685A or L692A *Mfn2* mutants, suggesting that MFI8 interacts specifically with the HR2 domain of *Mfn2* in cells (Fig. 3c, Supplementary Fig. 10b, c). In line with these experiments, the same mutations on the HR2 domain reduced the capacity of MASM7 and MFI8 to retain the

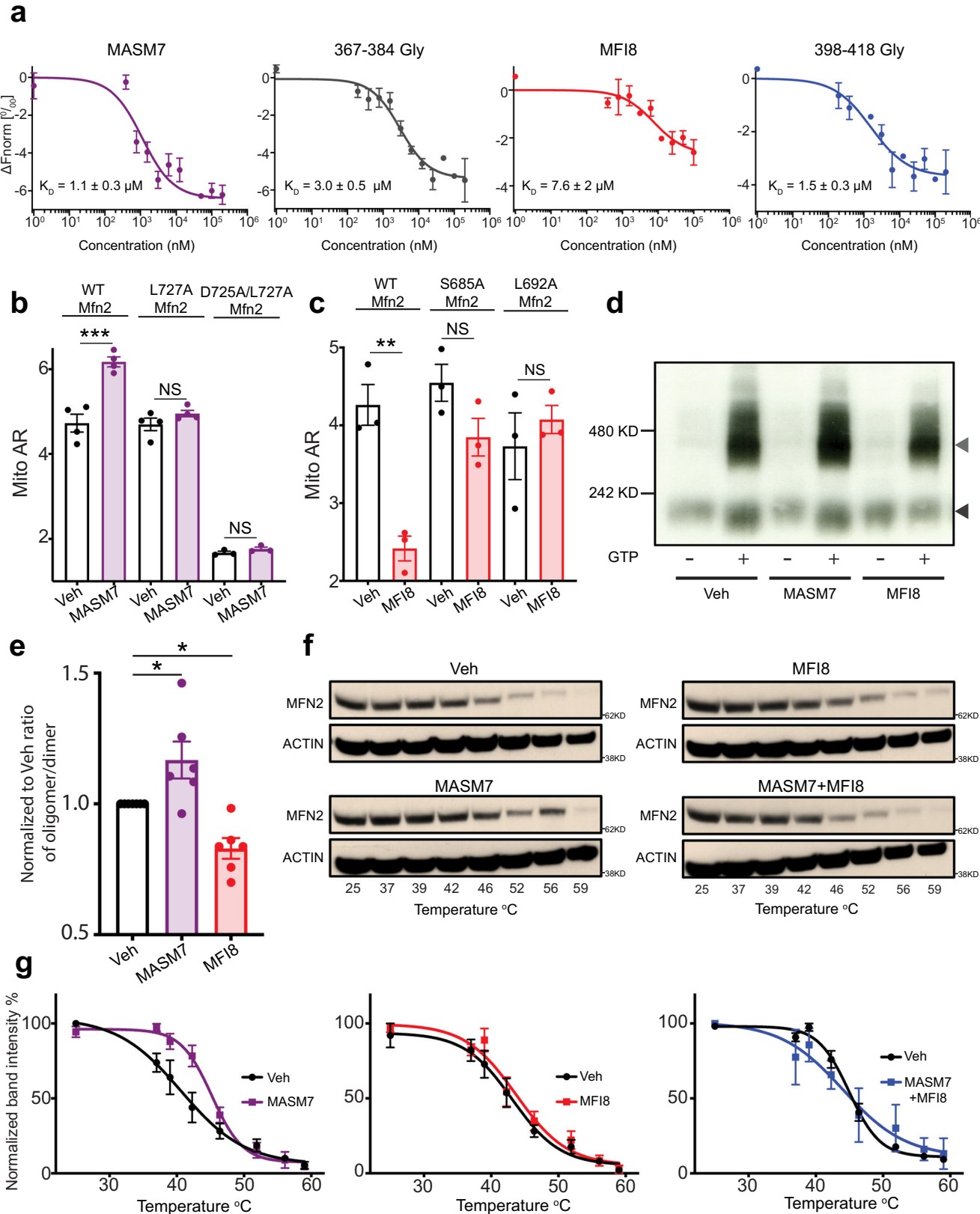

interactions with the HR2 domain during molecular dynamic simulations (Supplementary Figs. 4c–e, 6c–e). Collectively, MST, NMR, molecular modeling and mitochondrial morphology data supported that MASM7 and MFI8 specifically interact with the HR2 domain as predicted from our pharmacophore models.

**MASM7 and MFI8 modulate MFN conformation and complexes.** Next, we investigated whether MASM7 and MFI8 can modulate MFN2 oligomerization in its native membrane environment. Thus, we treated isolated mitochondria from MEFs with MASM7 or MFI8 and monitored the capacity of MFN2 to form oligomers using blue native gel electrophoresis (BN-PAGE)[28].

**Fig. 3 MASM7 and MFI8 interact specifically with the HR2 domain of MFN2. a** Plots of mean corrected normalized fluorescence ($\Delta F_{norm}$: $F_{nomr}$ bound $-$ $F_{norm}$ unbound) from MST signal analysis of titrations of HR2 with indicated compounds and peptides. Data represent mean $\pm$ SEM from three replicate experiments. **b** Quantification of Mito AR of *Mfn1/Mfn2 DKO* MEFs reconstituted with WT, L727A, and D725A/L727A *Mfn2* and treated with MASM7 (1 μM, 2 h). Data represent mean $\pm$ SEM of three independent biological replicates. Statistics were obtained using two-tailed unpaired t-test: *$p < 0.05$, **$p < 0.01$, ***$p < 0.001$, ****$p < 0.0001$. **c** Quantification of Mito AR of *Mfn1/Mfn2 DKO* MEFs reconstituted with WT, S685A and L692A *Mfn2* and treated with MFI8 (20 μM, 6 h). Data represent mean $\pm$ SEM of three independent biological replicates. Statistics were obtained using two-tailed unpaired t-test: *$p < 0.05$, **$p < 0.01$, ***$p < 0.001$, ****$p < 0.0001$. **d** Separation of MFN2 oligomers from isolated mitochondria treated with GTP, MASM7, and MFI8 as indicated using BN-PAGE electophoresis. Gray arrowhead marks 450 kD MFN2 oligomer and black arrowhead marks MFN2 dimer. **e** Quantification of the proportion of the MFN2 oligomer observed in the ~450 kD band to the MFN2 dimer band after treatment with GTP (**d**). Data represent mean $\pm$ SEM of seven independent biological replicates. Statistics were obtained using one way ANOVA: *$p < 0.05$, **$p < 0.01$, ***$p < 0.001$, ****$p < 0.0001$. **f** Cellular engagement (CETSA) of MFN2 by MASM7, MFI8, and the combination of both. A representative blot from three independent experiments is shown. **g** Quantification of temperature-dependent normalized MFN2 levels obtained by densitometry with corresponding fitted curves. Data represent mean $\pm$ SEM from $n = 3$ independent experiments. Source data are provided as a Source Data file.

MFN2 migrated as a dimer in the absence of GTP, while incubation with GTP promoted higher order oligomers ~450 kD. Indeed, treatment of MASM7 in isolated mitochondria increased the ratio of higher order oligomers ~450 kD to dimers upon GTP binding, whereas MFI8 reduced the ratio (Fig. 3d, e). Importantly, neither MASM7 nor MFI8 altered the total MFN2 levels when treated under the same conditions in isolated mitochondria (Supplementary Fig. 11a).

Next, we used cellular extract thermal shift assay (CETSA), an established assay for small molecule-protein target engagement in intact cells[29,30]. Interestingly, MASM7 induced stabilization of MFN2 as it increased its melting point ᵀᵐ by 2.3 °C (Veh: $T_m = 42.9 \pm 0.8$ °C vs MASM7: $T_m = 45.2 \pm 0.5$ °C) (Fig. 3f, g). Furthermore, we obtained the melting curves of MFN2 with or without MASM7 from *Mfn2* KO MEFs reconstituted with WT, L727A, and D725A/L727A *Mfn2*. Notably, MASM7 induced stabilization of WT MFN2 (Veh: $T_m = 42.2 \pm 0.6$ °C vs MASM7: $T_m = 45.3 \pm 2.3$ °C), but not of L727A *Mfn2* (Veh: $T_m = 43.8 \pm 0.3$ °C vs MASM7: $T_m = 44.1 \pm 0.4$ °C) or D725A/L727A *Mfn2* (Veh: $T_m = 44.1 \pm 1.7$ °C vs MASM7: $T_m = 43.3 \pm 1.7$ °C) (Supplementary Fig. 11b–d).

In contrast, MFI8 alone had no effect on the $T_m$ of MFN2 (Veh: $T_m = 42.9 \pm 0.8$ °C vs MFI8: $T_m = 43.7 \pm 1.1$ °C) (Fig. 3f, g). Given that MFI8 binds in vitro and in cells on the HR2 domain of MFN2 (Fig. 3a, c), reduces the ratio of higher order MFN2 oligomers to dimers (Fig. 3d, e) and inhibits mitochondrial fusion in a MFN1/2 dependent manner (Fig. 2g), we speculated that MFI8 engages better with the pro-tethering conformation of MFN2. To test this hypothesis we co-treated cells with MFI8 and MASM7, where the latter compound was found to promote the pro-tethering conformation and increase the Tm of MFN2. Interestingly, MFI8 reduced the capacity of MASM7 to promote MFN2 stabilization (Veh: $T_m = 42.9 \pm 0.8$ °C vs MFI8 + MASM7: $T_m = 44.05 \pm 2.6$ °C) (Fig. 3f, g). This result is consistent with the data showing that MFI8 can still promote mitochondrial fragmentation even when co-treated with MASM7 (Fig. 2h and Supplementary Fig. 5f).

It is noteworthy that neither MASM7 nor MFI8 altered *Mfn1* and *Mfn2* gene expression and their corresponding protein levels (Supplementary Fig. 12a, b). Furthermore, neither MASM7 nor MFI8 altered Tomm20 protein levels, mitotracker green intensity, and mitochondrial to nuclear DNA ratio, suggesting that none of the compounds altered mitochondrial biomass (Supplementary Fig. 12c–f). In line with this result, no alteration in the gene expression levels of mitochondrial biogenesis markers was detected upon MASM7 or MFI8 treatment (Supplementary Fig. 12g).

## Modulation of mitofusins activity alters mitochondrial respiration and functionality. Previous reports showed that knockdown of *Mfn2* or knockout of both *Mfn1/2* reduced but it

did not completely diminish respiration in cells[7,8], suggesting that mitochondria from cells that lack both MFN1/2 can still use respiration to produce ATP, albeit to a lesser extent than WT cells. Given that MASM7 positively and MFI8 negatively regulate MFN1/2' fusogenic activity, we examined whether they can affect mitochondrial respiration. Strikingly, MASM7 increased basal and maximal respiration and mitochondrial ATP production in WT but not in *Mfn1/Mfn2 DKO* MEFs (Fig. 4a, b). On the other hand, MFI8 reduced respiration and mitochondrial ATP production in WT but not in *Mfn1/Mfn2 DKO* MEFs (Fig. 4c, d). Importantly, MFI8 did not alter the ratio of state III/ state II respiration of isolated mitochondria, suggesting that MFI8 does not impact directy the electron transport chain (ETC) or is an non-specific uncoupler to isolated mitochondria but it rather reduces mitochondrial respiration by modulating mitochondrial dynamics (Supplementary Fig. 13a). These results support the idea that the fusogenic activity of MFN1/2 positively correlates with mitochondrial respiration.

To further support this idea we reconstituted WT *Mfn2* or β-Galactosidase (βGal) in *Mfn1/Mfn2 DKO* MEFs and evaluated the mitochondrial membrane potential using TMRE staining as a readout (Supplementary Fig. 13b). Interestingly, cells that expressed WT *Mfn2* possessed a higher mitochondrial membrane potential compared to cells that expressed βGal. Consistently, MASM7 concentration responsively increased mitochondrial membrane potential in WT MEFs (Fig. 4e). In line with this result, MASM7 significantly increased mitochondrial membrane potential of *Mfn1/Mfn2 DKO* MEFs when reconstituted with WT *Mfn2* and to a lesser extent when reconstituted with L727A *Mfn2* (Supplementary Fig. 13c). Importantly, MASM7 was not able to increase membrane potential when treated to isolated mitochondria in vitro in conditions where mitochondrial fusion cannot take place (Supplementary Fig. 13d). Moreover, MASM7-induced increase in mitochondrial membrane potential was revoked when cells were co-treated with myxothiazol or rotenone (Supplementary Fig. 13e, f), suggesting that the increase in the membrane potential upon MASM7 treatment is derived from an increased activity of the ETC. On the other hand, MFI8 concentration responsively decreased mitochondrial membrane potential in WT MEFs (Fig. 4f). It is noteworthy that MFI8 significantly decreased mitochondrial membrane potential of *Mfn1/Mfn2 DKO* MEFs when reconstituted with WT MFN2 but the effect was reduced when reconstituted with L692A MFN2 (Supplementary Fig. 13g). Strikingly, MFI8 reduced the gene expression of several nuclear-encoded subunits of the respiratory complexes as revealed by RNA-seq analysis, while MASM7 altered the expression of selected genes of the respiratory complexes rather than inducing a consistent trend (Fig. 4g). These findings are consistent with the idea that MFI8 promotes mitochondrial dysfunction and highlight that alterations in mitochondrial dynamics can affect gene transcription.

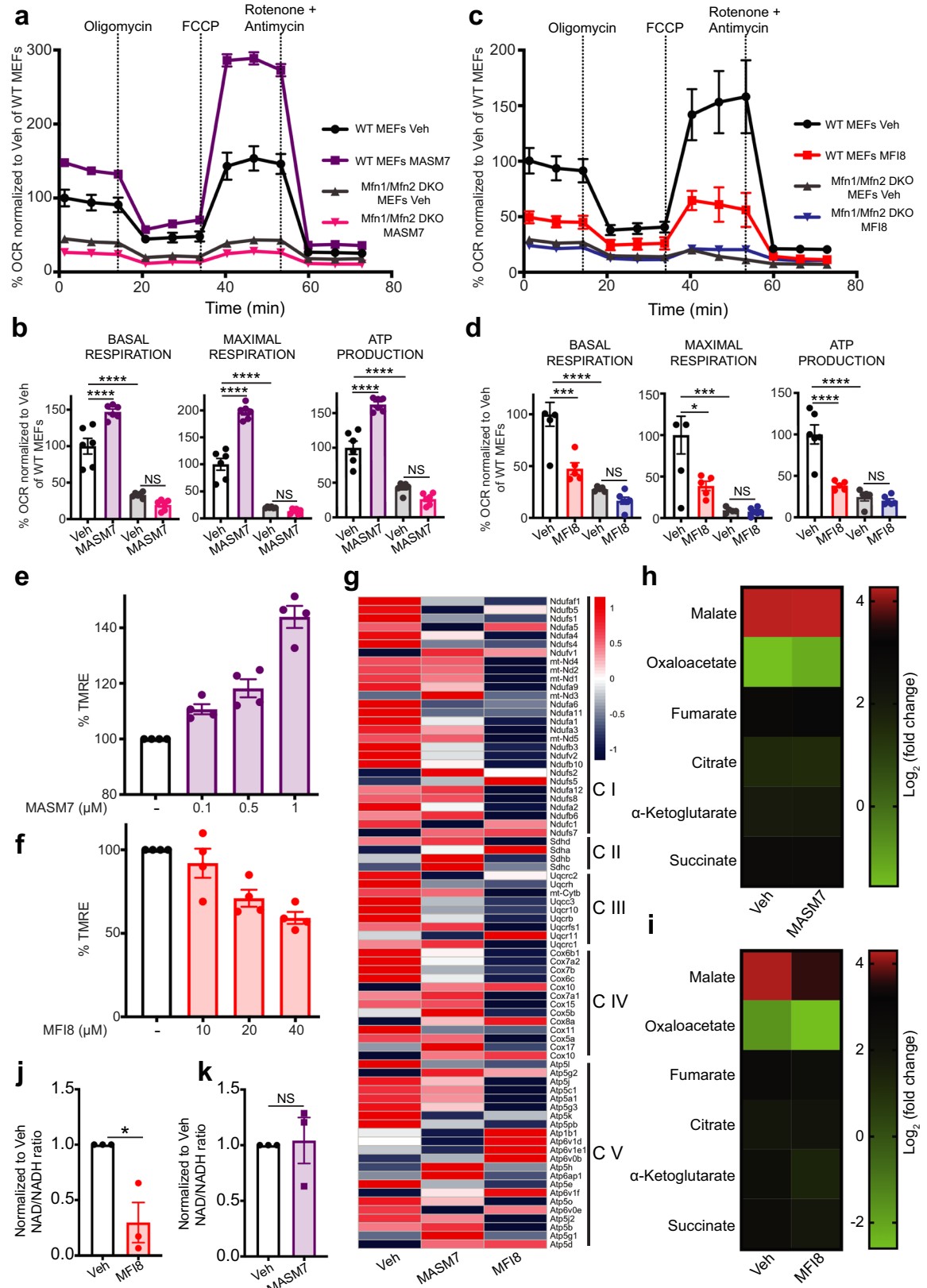

**Fig. 4 MASM7 and MFI8 alter mitochondrial functionality. a** A representative OCR trace of WT or *Mfn1/Mfn2* DKO MEFs from three independent experiments is shown. OCR was normalized to Veh of WT MEFs for $t = 1.31$ min. Data represent mean ± SD from $n = 6$ replicates. **b** Bar graphs showing the quantification of basal, maximal respiration and ATP production of (**a**). Data represent mean ± SEM from $n = 6$ replicates. **c** A representative OCR trace of WT or *Mfn1/Mfn2* DKO MEFs from three independent experiments is shown. OCR was normalized to Veh of WT MEFs for $t = 1.31$ min. Data represent mean ± SD from $n = 5$ replicates. **d** Bar graphs showing the quantification of basal, maximal respiration, and ATP production of (**c**). Data represent mean ± SEM from $n = 5$ replicates. **e** MASM7 concentration responsively increased membrane potential in MEFs. Cells were treated with the indicated concentrations of MASM7 for 6 h. Data represent mean ± SEM of four independent biological replicates. **f** MFI8 concentration responsively decreased membrane potential in MEFs. Cells were treated with the indicated concentrations of MFI8 for 6 h. Data represent mean ± SEM of four independent biological replicates. **g** Relative variance-stabilized expression levels of nuclear-encoded mitochondrial genes belonging to respiratory complexes. **h** Heatmap showing the alteration in the metabolites of the TCA cycle. Heatmap was generated by quantitative metabolomics from three independent biological replicates. **i** Heatmap showing the alteration in the metabolites of the TCA cycle. Heatmap was generated by quantitative metabolomics from three independent biological replicates. **j** NAD/NADH ratio. Data represent mean ± SEM of three independent biological replicates. **k** NAD/NADH ratio. Data represent mean ± SEM of three independent biological replicates. Statistics were obtained using one way ANOVA for panels (**b**, **d**) or two-tailed unpaired t-test for panel (**j**): *$p < 0.05$, **$p < 0.01$, ***$p < 0.001$, ****$p < 0.0001$. Cells were treated with MASM7 (1 µM, 6 h) for panels (**a**, **g**, **h**, **k**) or MFI8 (20 µM, 6 h) for panels (**c**, **g**, **i**, **j**). Source data are provided as a Source Data file.

Next, we sought to determine whether such alterations can have an impact on the metabolites of the TCA cycle[31]. Surprisingly, MASM7 had no significant effect in the majority of the metabolites of the TCA cycle (Fig. 4h). On the contrary, MFI8 markedly reduced several metabolites such as malate, oxaloacetate, and α-ketoglutarate (Fig. 4i). Consistently, MFI8 decreased the total NAD$^+$/NADH ratio, which is consistent with reduced oxidative capacity (Fig. 4j). In contrast, MASM7 had no effect on total NAD$^+$/NADH ratio (Fig. 4k). Collectively, these data suggest that MFN1/2 acts as regulators of mitochondrial functionality and that mitochondrial fusion positively correlates with mitochondrial functionality.

**Inhibition of mitofusins activity induces minority MOMP and primes cells for cell death.** Previous reports highlighted the importance of mitochondrial shape and function in the execution and sensitivity of the cells to various types of cell death, such as apoptosis[32–38]. Thus, we were prompted to use our compounds to better understand how mitochondrial shape regulates apoptotic signaling. Remarkably, MFI8 concentration-responsively increased caspase-3/7 activity, a key component of apoptotic signaling (Fig. 5a), whereas MASM7 had no effect on it (Fig. 5b). Furthermore, an analog of MFI8 that is unable to inhibit MFN1/2 fusogenic activity and induce aberrant mitochondrial fragmentation (Supplementary Fig. 5c, d), MFI22, did not increase caspase-3/7 activity (Fig. 5c). Consistently, deletion of *Mfn1/Mfn2* also impaired the capacity of MFI8 to increase caspase-3/7 activity (Fig. 5d). These data suggest that MFI8 induces caspase 3/7 activation in a MFNs-dependent manner. Deletion of *Apaf-1* was detrimental for the capacity of MFI8 to increase caspase 3/7 activity, suggesting that apoptosome formation is crucial for the MFI8-induced caspase 3/7 activation (Fig. 5e). Furthermore, we also analyzed cytosolic and mitochondrial fractions upon treatment with MFI8 and found that cytochrome c was released to the cytosol, albeit at modest levels (Fig. 5f). Importantly, MFI8 unlike with the combination of BH3 mimetics (ABT-737/S63845) was not able to induce cytochrome c release when treated to *Mfn1/Mfn2* DKO MEFs (Fig. 5g). Moreover, ABT-737/S63845 combination but not MFI8 decreased mitochondrial membrane potential when treated to *Mfn1/Mfn2* DKO MEFs (Fig. 5h). Collectively, these data suggest that MFI8 induces cytochrome c release and decreases membrane potential in a mitofusin-dependent manner. Interestingly, MFI8 did not increase the percentage of dead cells upon caspase 3/7 activation (Fig. 5i). Consistently, neither MFI8 nor MASM7 decreased cellular viability over the course of 72 h (Fig. 5j).

Since cytochrome *c* release and caspase-3/7 activation was detected upon MFI8 treatment, we posited that inhibition of

mitochondrial fusion by MFI8 could induce mitochondrial outer membrane permeabilization (MOMP). Using an established cell imaging assay in U2OS cells that expressed fluorescent cytosolic (CytoGFP) and mitochondrial (MitoCherry) probes, we were able to quantify MOMP[39]. Notably, MFI8 induced MOMP, albeit at modest levels (Fig. 5k). Accordingly, MFI8 but not MASM7 induced caspase-3/7 activation in U2OS cells (Fig. 5l). Of note, MFI8 also inhibited mitochondrial fusion in U2OS cells (Fig. 5m). Taken all together our data suggest that MFI8 induced minority MOMP and subsequent activation of caspases 3/7, albeit at sub-lethal levels. Previously, Ichim et al. showed that minority MOMP induces genomic instability[39]. Hence, we sought to evaluate the capacity of MFI8 in inducing DNA damage using γH2AX as a readout[40]. Interestingly, MFI8 increased γH2AX foci in WT MEFs but not in *Mfn1/Mfn2* DKO MEFs, demonstrating that MFI8 induces DNA damage in a mitofusin-dependent manner (Fig. 6a–c). Of note, MASM7 did not induce DNA damage in any of the cell lines (Fig. 6a–c). Consistently, MFI8 upregulated several genes that are involved in DNA damage response in MEFs (Fig. 6d). Moreover, MFI8 induced DNA damage in U2OS cells (Fig. 6e, f). Importantly, co-treatment of a pan-caspase inhibitor, Q-VD-OPh, with MFI8 abolished the capacity of the latter to induce DNA damage in U2OS (Fig. 6g).

We further investigated if induction of minority MOMP by MFI8 can be used to enhance the capacity of another pro-apopotic agent to induce cell death. Thus, we used BV6, a bivalent SMAC mimetic that induces caspase-dependent cell death predominantly via XIAP inhibition. Notably, deletion of *Mfn1/Mfn2* sensitized cells to BV6 treatment (Fig. 7a and Supplementary Figs. 14, 15). Consistently, MFI8 potentiated the capacity of BV6 to induce cell death (Fig. 7b and Supplementary Fig. 14). The effect of MFI8 was specific to MFN1/2 inhibition as MFI22 did not sensitize cells to BV6 treatment (Fig. 7c and Supplementary Fig. 14). Importantly, deletion of *Mfn1/Mfn2* and *Apaf-1* abolished the capacity of MFI8 to sensitize cells to BV6 treatment (Fig. 7d, e and Supplementary Fig. 15). Collectively, these results demonstrate that inhibition of MFN1/2 fusogenic activity induces minority MOMP and sub-lethal caspase-3/7 activation, which leads to DNA damage and can sensitize cells to apoptotic cell death in combination with a SMAC-mimetic.

**Discussion**

Investigating the mechanisms of mitofusins regulation enables a deeper understanding of the critical function of mitofusins in mitochodrial dynamics, signaling, and related pathways such as apoptosis. Here, using structural and biochemical insights, we identified small molecules that are capable of regulating directly either positively or negatively MFN1/2 activity and subsequently

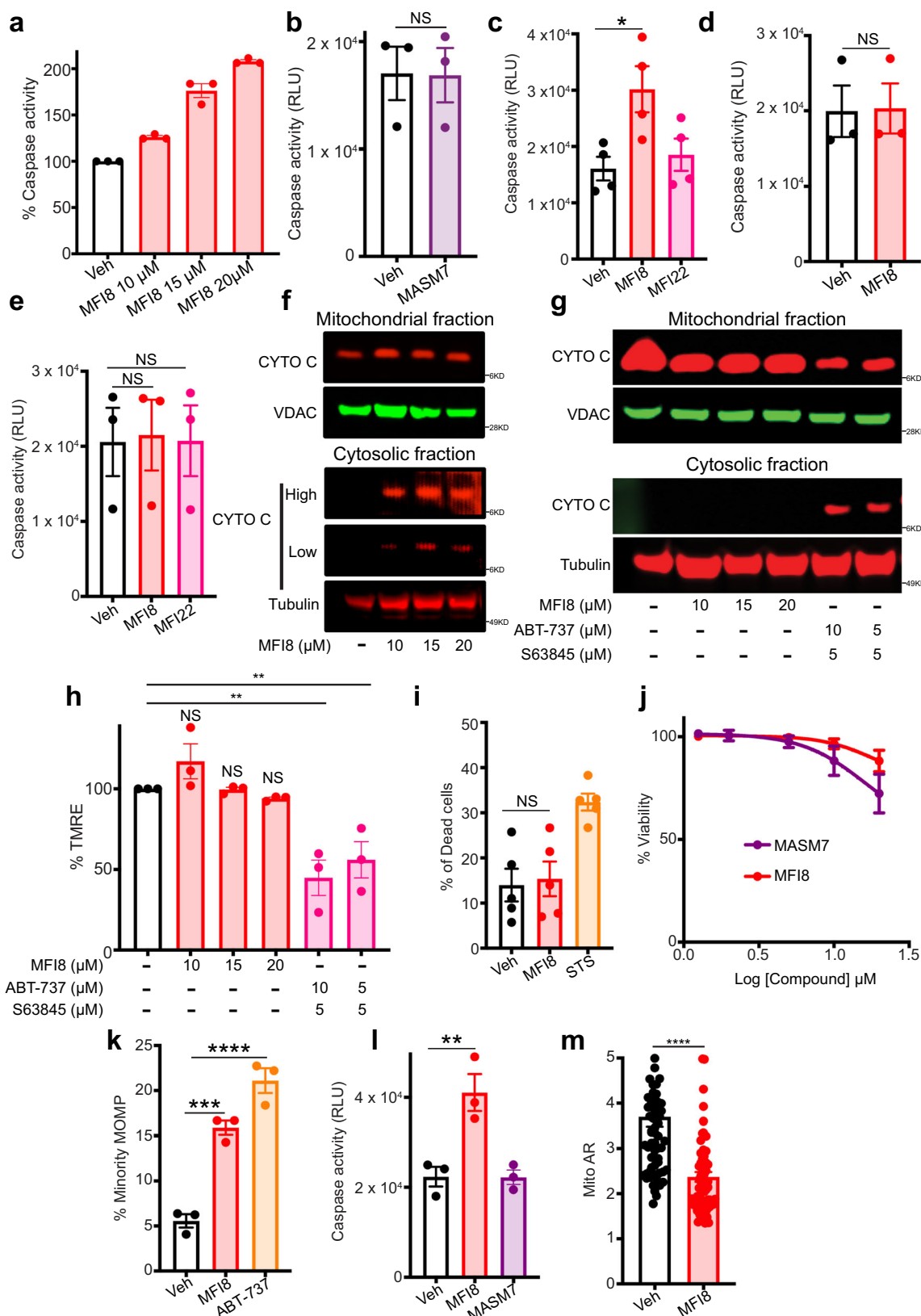

mitochondrial fusion. Our studies demonstrated direct binding of the mitofusin activator MASM7 and mitofusin inhibitor MFI8 to the recombinant HR2 domain of MFN2 and to intact MFN2 localized at mitochondria in cells. Binding and activity data come in agreement with our pharmacophore models where MASM7 and MFI8 were designed to interact with the HR2 residues based on the side chains of HR1 residues. MASM7 promotes the pro-tethering conformation of MFN1/2 to enable mitochondrial fusion, whereas MFI8 impedes mitochondrial fusion by directly interfering with the tethering permissive conformation of MFN1/2. MASM7 and MFI8 were found to increase or decrease, respectively, the GTP-dependent MFN2 higher-order oligomers,

**Fig. 5 MFI8 induces minority MOMP and cytochrome c release leading to caspase 3/7 activation. a** Caspase 3/7 assay in WT MEFs treated with MFI8 for 6 h. Data represent mean ± SEM of three independent biological replicates. **b** Caspase 3/7 assay in WT MEFs treated with MASM7 (1 μM, 6 h). Data represent mean ± SEM of three independent biological replicates. Caspase 3/7 assay in WT (**c**), *Mfn1/Mfn2* DKO (**d**) and *Apaf-1* KO (**e**) MEFs. Cells were treated with MFI8 or MFI22 (20 μM, 6 h). Data represent mean ± SEM of three independent biological replicates. **f** Western blot analysis of cytochrome c release from MEFs treated with MFI8 for 6 h. Blot is representative of $n = 2$ independent experiments. **g** Western blot analysis of cytochrome c release from *Mfn1/Mfn2* DKO MEFs treated with MFI8 for 6 h. ABT-737 and S63845 were used as positive controls for the assay. Blot is representative of $n = 2$ independent experiments. **h** Mitochondrial membrane potential assay in *Mfn1/Mfn2* DKO MEFs treated with MFI8 and ABT-737/S63845 for 6 h. Data represent mean ± SEM of three independent biological replicates. **i** Evaluation of dead cells using Annexin V/PI staining via flow cytometry. Cells positive to Annexin V, PI, or both, were considered as dead. Data represent mean ± SEM of five independent biological replicates. **j** Viability assay in MEFs treated with MASM7 or MFI8 for 72 h. Data represent mean ± SEM of three independent biological replicates. **k** U2OS cells expressing CytoGFP/MitoCherry were treated with MFI8 (20 μM, 6 h) or ABT-737 (10 μM, 3 h) and minority MOMP was quantified. Data represent mean ± SEM of three independent biological replicates. **l** Caspase 3/7 assay in U2OS cells treated with MFI8 (20 μM, 6 h) or MASM7 (1 μM, 6 h). Data represent mean ± SEM of three independent biological replicates. **m** Quantification of Mito AR in U2OS cells treated with MFI8 (20 μM, 6 h). Data represent mean ± SEM of $n = 65$ mitochondria. Statistics were obtained using one way ANOVA for (panels **c, e, h, i, k, l**) or two-tailed unpaired t-test for panels (**b, d, m**): *$p < 0.05$, **$p < 0.01$, ***$p < 0.001$, ****$p < 0.0001$. Source data are provided as a Source Data file.

demonstrating these small molecules can modulate the levels of pro-fusion oligomers, and therefore the extent of fusion among mitochondria. These data also support that the activity of MASM7 and MFI8 through their interactions with the HR2 domain of MFN1/2[17,18,20,23] is compatible with the proposed GTP-mediated dimerization of MFN1/2[17,18,20,23]. Our data indicated that MASM7 can activate both MFN2 and MFN1, while MFI8 can inhibit both MFN2 and MFN1. This can be attributed to the high sequence homology between MFN1 and MFN2, and the conservation of the residues that are located in the binding region of each small molecule between MFN1 and MFN2.

Our studies establish a direct link between MFN1/2 and mitochondrial functionality. MASM7-induced fused mitochondria were found to have increased membrane potential, mitochondrial respiration, and ATP production. On the contrary, MFI8-induced fragmented mitochondria were found to have decreased membrane potential, mitochondrial respiration, and ATP production. These findings are in line with previous reports showing that deletion of *MFN1/2* reduces mitochondrial respiration and membrane potential[8,41]. Here our data suggest a direct connection of MFN1/2 and mitochondrial fusion with improved mitochondrial function. However, the molecular mechanism that regulates the crosstalk between mitochondrial shape and function is still elusive. One possibility is that mitochondrial shape may affect the ETC by regulating the formation of respiratory complexes and supercomplexes[7,41]. An alternative explanation may place mitochondrial quality control in between mitochondrial shape and function as a regulating factor of functional mitochondria, which is known to be affected by mitochondrial shape and MFN1/2 levels[42]. Indeed, expression of MFN1/2 has been positively correlated with upregulation of mitophagy, suggesting that increased mitochondrial fusion promotes clearance of damaged mitochondria[43]. Interestingly, TCA cycle metabolites were affected only when mitochondrial fusion was inhibited by MFI8. Our data suggest that inhibition of mitochondrial fusion will reduce the activity of TCA cycle in response to decreased respiration.

Previous reports that linked mitochondrial dynamics with apoptosis typically used a cell death stimulus to induce MOMP and activate the apoptotic machinery[44–46]. For example, cytochrome c release, a post MOMP event, preceded mitochondrial fragmentation in cells that were treated with ABT-737, suggesting that mitochondrial fragmentation is a secondary effect in response to MOMP[45]. Our data demonstrate that minority MOMP can be induced in response to the direct inhibition of MFN1/2 and mitochondrial fusion. Although, the molecular mechanism that connects minority MOMP and mitochondrial shape will need further elucidation, our data place mitochondrial

dynamics as a regulating factor of MOMP in the absence of a cell death stimulus.

Although inhibition of mitochondrial fusion induced minority MOMP, sub-lethal activation of caspases-3/7, and DNA damage as downstream events of minority MOMP, it did not induce cell death. Such inshight provides a new opportunity to use inhibition of MFN1/2 activity to sensitize cells to other agents promoting apoptosis. Here, we demonstrated that BV6, a SMAC mimetic that operates downstream of MOMP, when combined with MFI8 can synergistically induce cell death. Similar combinatorial strategies can have broad applications in cancer therapy by enhancing the activity of pro-apoptotic drugs to induce caspase-mediated apoptosis or evade resistance mechanisms[46–48]. Furthermore, inhibition of mitochondrial fusion altered the transcription of several genes that are involved in DNA repair and respiration, suggesting that changes in mitochondrial shape can impact gene transcription of diverse gene sets. Our findings suggest that these are possible homeostatic mechanisms that cells use to counteract mitochondrial stress.

MFN1/2 modulators reported here allow temporal manipulation of the fusogenic activity of MFN1/2 in a reversible fashion. This is in contrast to other small molecules that have been reported such as the drug Leflunomide, which alters MFN1/2 protein levels through loss of pyrimidine synthesis and are likely to affect non-fusogenic functions of MFN1/2[49]. Importantly, the rational discovery of MASM7 and MFI8 raises the possibility of developing novel therapeutics for disorders/syndromes where impaired mitochondrial dynamics contributes to pathogenesis. Defective MFN2 mutants have been associated with development of Charcot-Marie-Tooth disease type 2A (CMT2A) and imbalances in mitochondrial dynamics have been linked to metabolic disorders such as type II diabetes, obesity, neurodegeneration, cancer, and aging[50]. Several mutations on the GTPase, HR1 and HR2 domain of the MFN2 have been identified from patient samples and correlated with the development of the CMT2A disease. Interestingly, M376, a residue that was used for the development of the pharmacophore model for the discovery of the MFN2 activators, has been found to be mutated in patient samples to Val[23,51]. Moreover, Leu724, a residue that is located in the binding region of the MFN2 activators, has been found to be mutated in patient samples to Pro[23]. These patient mutations highlight the importance of those residues in HR1 and HR2 domains for the function of the protein. Notably, MFN2 activators have been shown to rescue mitochondrial defects of CMT2A in preclinical models raising the opportunity for the development of novel therapeutics[52].

In summary, our study provides mechanistic insights of MFN1/2 activity that enable mitochondrial fusion and further

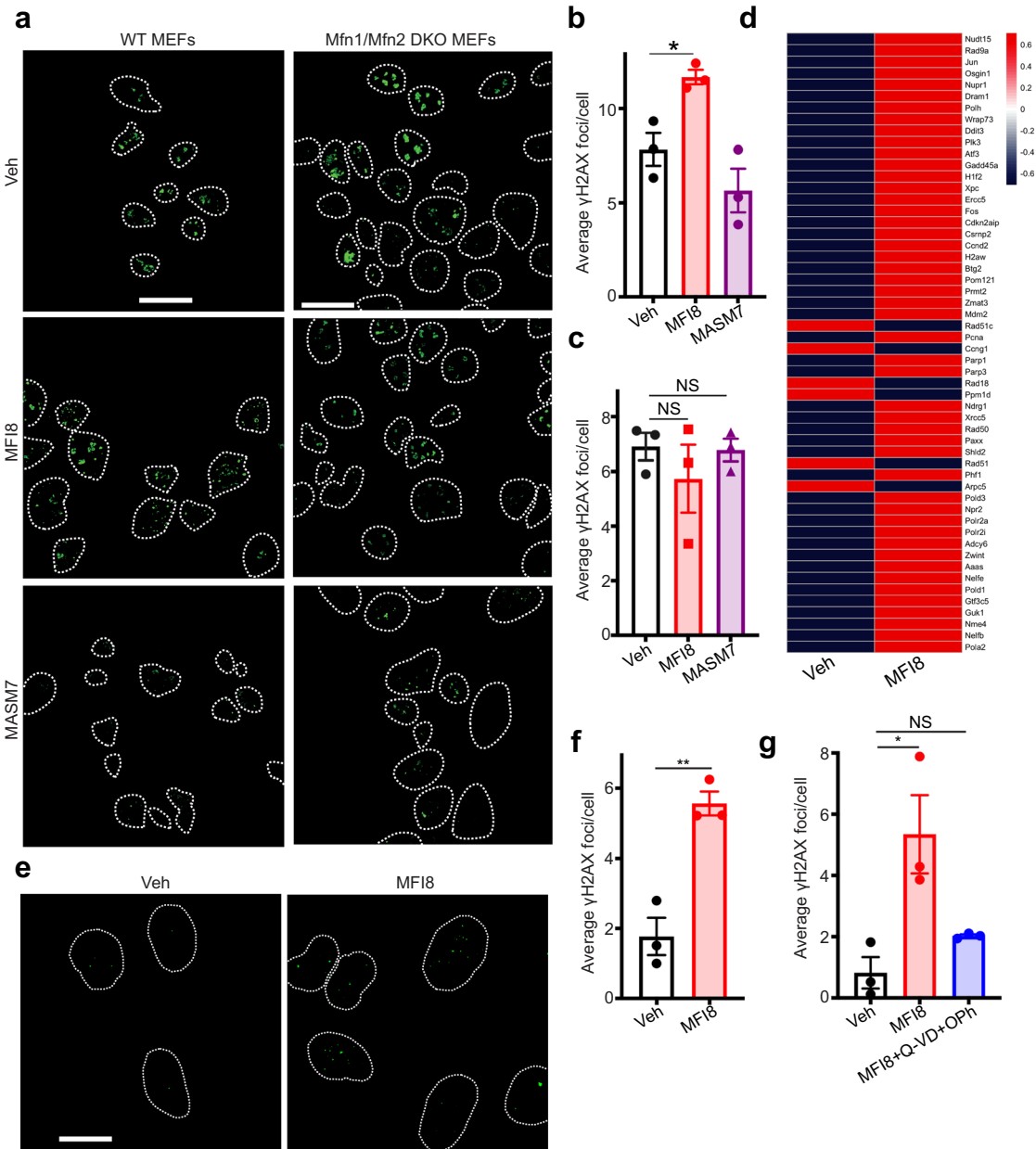

**Fig. 6 MFI8 induces DNA damage. a** MEFs and *Mfn1/Mfn2* DKO MEFs were immunostained for γH2AX. Cells were treated with MFI8 (20 μM, 6 h) or MASM7 (1 μM, 6 h). Scale bar 20 μm. **b** Quantification of γH2AX foci of WT MEFS (**a**). Error bars represent mean ± SEM of three independent biological replicates. **c** Quantification of γH2AX foci of *Mfn1/Mfn2* DKO MEFs (**a**). Error bars represent mean ± SEM of three independent biological replicates. **d** Relative variance-stabilized expression levels of genes that are involved in DNA damage response. **e** U2OS cells were immunostained for γH2AX. Cells were treated with MFI8 (20 μM, 6 h). Scale bar 20 μm. **f** Quantification of γH2AX foci of (**e**). Error bars represent mean ± SEM of three independent biological replicates. **g** Quantification of γH2AX foci from U2OS cells treated with MFI8 or the combination of MFI8 with Q-VD-OPh. Error bars represent mean ± SEM of three independent biological replicates. Statistics were obtained using two-tailed unpaired t-test for panel (**f**) or one way ANOVA for panels (**b**, **c**, **g**): *$p < 0.05$, **$p < 0.01$, ***$p < 0.001$, ****$p < 0.0001$. Source data are provided as a Source Data file.

delineates the role of MFN1/2 in controlling mitochondria function and signaling (Fig. 8). Small molecule MFN1/2 activators and inhibitors provided here will facilitate further investigation of MFN1/2 in physiological mechanisms and disease models and provide a blueprint for the discovery of novel therapeutics to target mitochondrial fusion.

## Methods

**Reagents**. MASM7 was obtained from Enamine (cat. # EN300-396282). Screened MASMs were purchased from Enamine, ChemBridge and ChemDiv. MFI8 was obtained from ChemBridge (cat. # 7681311) and also synthesized in house in a more stable form as a HCl salt. MFI8 was freshly dissolved in DMSO at 10 mM

prior each experiment. MFI22-26 were also synthesized in house as HCl salts. The rest of the screened MFIs were purchased from Enamine, ChemBridge, ChemDiv, Vitas M, and UORSY. All compounds were >95–98% pure, dissolved in 100% DMSO to prepare a 10 mM stock solution, and diluted in aqueous buffers or cell culture medium for assays. BV6 (B4653) and Q-VD-Oph (A1901) were purchased from APExBio. Staurosporine (S1421) was purchased from Selleck. Myxothiazol (T5580) was purchased from Sigma. Rotenone (36165) was purchased from Tocris. 367–384Gly or 398–418Gly peptides were synthesized by Genscript[18].

**Cell Lines**. MEFs (WT, *Mfn1* KO, *Mfn2* KO, and *Mfn1/Mfn2* DKO) were provided by David Chan's laboratory. *Drp1*flox/flox MEFs were generated in house. *Apaf-1* KO MEFs were provided by Xuejun Jiang's laboratory. U2OS cells were provided from Stephen Tait's laboratory. All cells maintained in DMEM (Life Technologies)

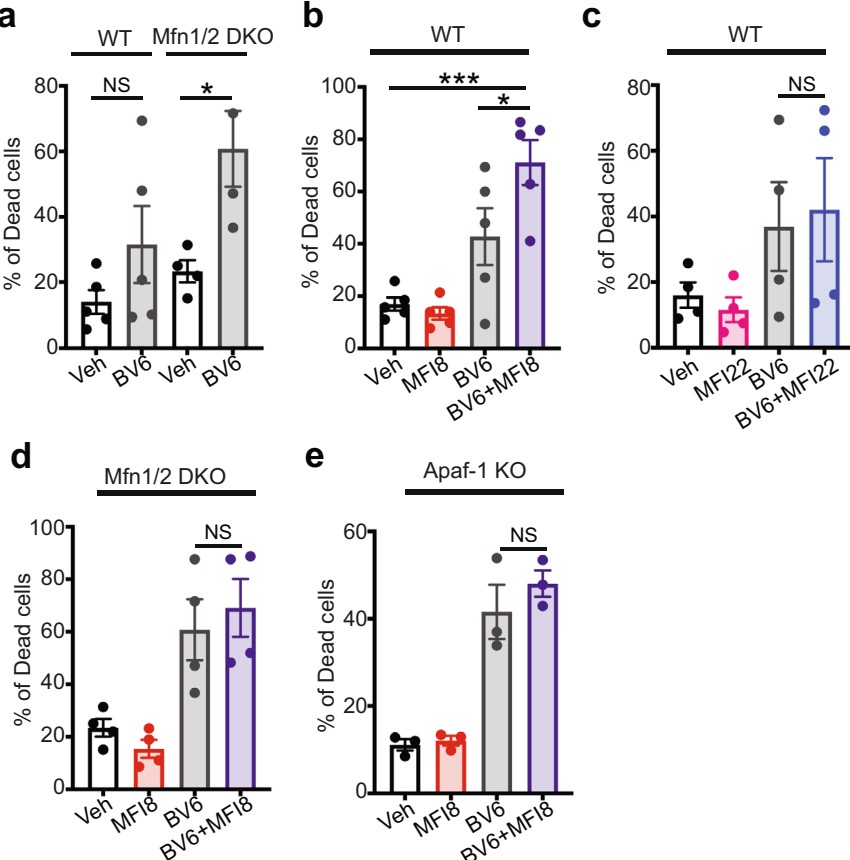

**Fig. 7 MFI8 induces cell death in combination with BV6 SMAC mimetic.** Evaluation of dead cells using Annexin V/PI staining as readouts via flow cytometry. Cells that were positive to Annexin V, PI, or both, were considered as dead. Cells were treated with MFI8 or MFI22 (20 μM, 6 h) and BV6 (20 μM, 6 h). **a** Percentage of dead cells in WT and *Mfn1/Mfn2* DKO MEFs upon treatment with BV6. Data represent mean ± SEM from $n = 5$ independent experiments for WT MEFs and $n = 4$ independent experiments for *Mfn1/Mfn2* DKO MEFs. **b** Percentage of dead cells in WT MEFs upon treatment with BV6, MFI8, or the combination of both. Data represent mean ± SEM from $n = 5$ independent experiments. **c** Percentage of dead cells in WT MEFs upon treatment with BV6, MFI22, or the combination of both. Data represent mean ± SEM from $n = 4$ independent experiments. **d** Percentage of dead cells in *Mfn1/Mfn2* DKO MEFs upon treatment with BV6, MFI8, or the combination of both. Data represent mean ± SEM from $n = 4$ independent experiments. **e** Percentage of dead cells in *Apaf-1* KO MEFs upon treatment with BV6, MFI8, or the combination of both. Data represent mean ± SEM from $n = 3$ independent experiments. Statistics were obtained using one way ANOVA for panels (**a**, **b**): *$p < 0.05$, **$p < 0.01$, ***$p < 0.001$, ****$p < 0.0001$. Source data are provided as a Source Data file.

supplemented with 10% FBS, 100 U ml$^{-1}$ penicillin/streptomycin, and 2 mM L-glutamine.

**Generation of *DRP1* KO cells.** Approximately 100,000 *DRP1*$^{flox/flox}$ MEFs were seeded on chamber slides (MatTek Corporation: 35 mm dishes, No. 1.5, 14 mm glass diameter) and infected with Adeno-Cre or Adeno-β Galactosidase at MOI of 10. The following day the media was changed and cells were infected again with Adeno-Cre or Adeno-β Galactosidase at MOI of 10 and cultured for 24 h. Finally, cells were treated with MASM7 or DMSO and imaged to assess mitochondrial morphology using live cell imaging.

**Structural model of MFN2.** The structural model of MFN2 was calculated based on the I-TASSER (Iterative Threading ASSEmbly Refinement) hierarchical approach to protein structure using bacterial dynamin-like protein (PDB: 2J69) and truncated crystal structure of MFN2 (PDB ID: 6JFL). Energy minimization and analysis of the top-ranked structure was performed with MAESTRO tools (Maestro, Schrödinger Release 2016–2020, Schrödinger, LLC). PyMOL (The PyMOL Molecular Graphics System. Schrödinger Release 2016–2020, Schrödinger, LLC) was used for preparing the figures.

**In silico small molecule library preparation.** eMolecules (www.emolecules.com) library of purchasable compounds was converted to 3D structures using LIGPREP (LigPrep, Schrödinger Release 2016, Schrödinger, LLC) and EPIK (Epik, Schrödinger Release 2016, Schrödinger, LLC) generating an in silico library of approximately 13.8 million compounds containing compounds with different ionization state at pH 7.0 ± 2.0, stereochemistry and tautomeric form, excluding

potential Pan Assay Interference Compounds (PAINS) using PAINS definitions included in Canvas. Conformation analysis of ligands was calculated using the OPLS3 force field.

**3D Pharmacophore model generation and screen.** Phase (Phase, Schrödinger Release 2016, Schrödinger, LLC) module was used to generate a pharmacophore hypothesis and a 3D pharmacophore screen[25]. The coordinates of the HR1 helix residues Val372, Met376, and His380 from the structural model of MFN2 were used to assign pharmacophore points in 3D coordinates. Pharmacophore hypothesis included 5 features as defined in Phase for 3 hydrophobic groups to mimic the sidechain residues of Val273 and Met376 and an aromatic ring with a hydrogen-bond donor to mimic the sidechain of His380. The pharmacophore screen used the in silico library of compounds prepared from the eMolecules library in pre-existing conformations with the requirement to satisfy at least 4 out of the 5 pharmacophore features of the hypothesis. The top 1000 compounds ranked based on the Phase Score were selected for further visual analysis and clustered for diversity using dendritic fingerprints in Canvas[53]. Physicochemical and AMDET properties including Lipinski rules, permeability, logP, metabolic liabilities, and hERG inhibition were evaluated using QikProp (QikProp, Schrödinger Release 2016, Schrödinger, LLC). The highest 8 ranked compounds and the 10 most diverse compounds yielded selected molecules for experimental validation. MASM7 and MFI8 were checked for potential Pan Assay Interference Compounds (PAINS)[54] and has not been reported as a hit in previous screens in Pubchem database.

**Molecular dynamics simulations and molecular docking.** Hydrogen atoms and bond orders of a full-length (757 residues) MFN2 model were assigned to the

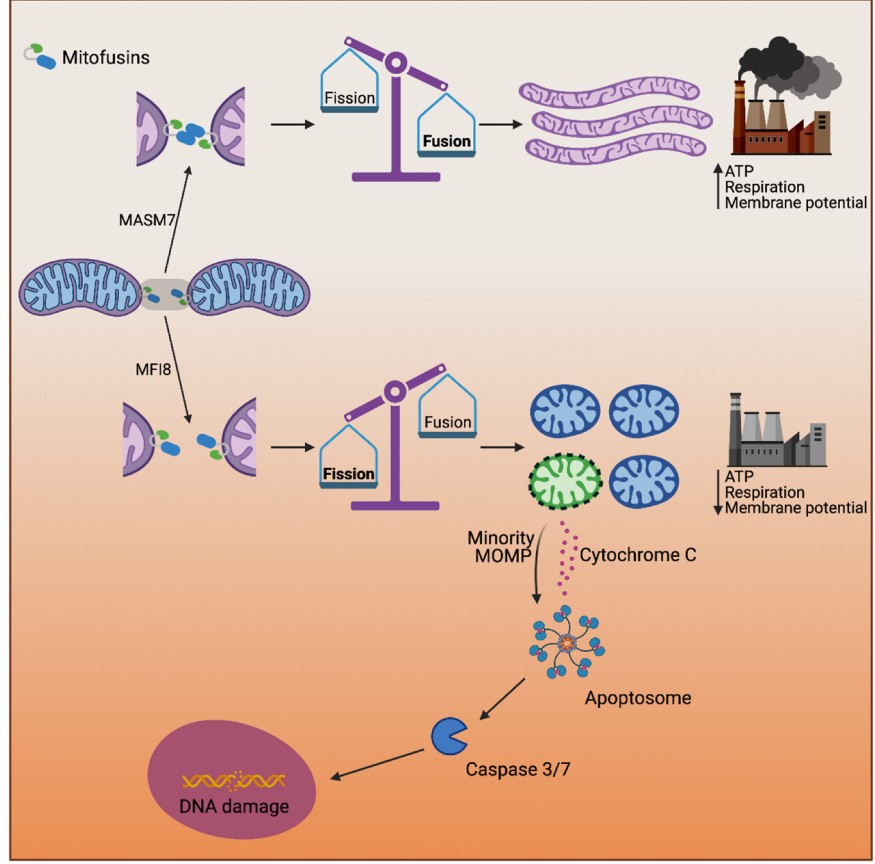

**Fig. 8 Schematic representation of the mechanism of action of MASM7 and MFI8 and their effects in mitochondrial fusion, mitochondrial functionality, and signaling.** The figure was created with BioRender.com.

protein by the "Protein Preparation Wizard" in MAESTRO (Schrodinger LLC, 2021). The structural model of MFN2 was generated using the bacterial dynamin-like protein (PDB: 2J69) as a template. Molecular dynamics (MD) simulations were performed using DESMOND (Schrodinger LLC, 2021). The system was solvated in a truncated orthorhombic box of size $128.3 \times 85.5 \times 78.6$ Å using a TIP3P water model and buffer minimize box volume option in the "System Builder" tool. The system was neutralized by adding 4 Cl− ions and a salt concentration of 0.15 M of NaCl was included in the simulation box to reproduce physiological conditions. Three independent 250 ns MD simulations were performed using OPLS3e force field and the NPT ensemble at 300 K and constant pressure of 1.0325 Atm. The Nose–Hoover Chain thermostat and Martyna–Tobias–Klein barostat were used to maintain the temperature and pressure, respectively. Before production runs, the system was relaxed and pre-equilibrated using the standard equilibration protocol implemented in DESMOND (Multisim protocol). The convergence of MD simulations was assessed by means of the RMSD of protein Cα-atom positions in time. For docking of MASM7, MFI8, and analogs the full-length MFN2 model in the closed conformation was used only as there is no available such full-length crystal structure. The Induced Fit Docking mode in GLIDE (Schrodinger LLC, 2021) was used after relaxing the HR1–HR2 interaction. Ligands were converted to 3D all atom structure using LIGPREP (Schrödinger, LLC, 2021) and assigned partial charges with EPIK (Schrödinger, LLC, 2021). S685A and D625A/L724 MFN2 mutants were created in MAESTRO from the MFN2 model and were prepared for docking and MD analysis as above. After induced fit docking, the energetically top binding poses of MASM7, MFI8, and analogs were selected. To assess the stability of the ligand–HR2 interactions, MD simulations (50 ns) of ligand complexes with WT or mutated MFN2-HR2 were performed as described above, starting from the best docking poses in each case. Analysis of trajectories was performed using MAESTRO simulation event analysis tools (Schrodinger LLC) and interactive plots in MAESTRO. Interatomic distances and RMSD data obtained from MD analysis were plotted using GraphPad Prism (version 9). PyMOL (Schrodinger LLC, 2021) was used to plot structures of docked ligand complexes and MD trajectory snapshots.

**Live cell imaging**. MEFs were seeded on chamber slides (MatTek Corporation: 35 mm dishes, No. 1.5, 14 mm glass diameter) at ~70% confluence. Cells were treated with MASM7 or MFI8. Then, cells were stained with MitoTracker Green (200 nM, Invitrogen) for 20 min at 37 °C. After treatments, media was replaced

with FluoroBrite™ DMEM (Invitrogen) supplemented with 10% FBS, 100 U ml$^{-1}$ penicillin/streptomycin, 2 mM L-glutamine and MASM7 or MFI8 prior image acquisition. Images were taken with Leica SP5 inverted confocal microscope.

**Measurement of mitochondrial aspect ratio (Mito AR)**. Images that depict stained mitochondria with mitotracker green were analyzed using FIJI We measured for each mitochondrion its length and width. Then, we divided the length by the width for each mitochondrion to get its mitochondrial aspect ratio. >200 mitochondria were measured per treatment.

**Immunofluorescent analysis and microscopy**. Cells were treated with MASM7, MFI8, and or Q-VD-OPh at ~70% confluence for 6 h. After treatments, cells were washed with PBS and fixed with 4% PFA for 15 min. Then cells were permeabilized with 0.1% Triton X-100 in PBS for 10 min. Cells were blocked with 5% BSA in PBS-T and incubated overnight with primary antibodies as indicated; γH2AX (Sigma; 05-636, 1:100 dilution). After incubation with primary antibodies, cells were washed with PBS and incubated with the appropriate mouse secondary antibody (ThermoFisher Scientific; A-11001) in the blocking solution. After PBS washes, coverslips were dipped in water and mounted on glass slides using Vectashield containing DAPI (Vector laboratories). Images were taken with Leica SP5 inverted confocal microscope. Data were analyzed with ImageJ.

**Mutagenesis**. Mutagenesis on the HR2 domain for the NMR analysis or the *MFN2* gene that was used for the packaging of the adenovirus was performed by using QuikChange Lightning Site-Directed mutagenesis Kit (Qiagen; Cat. 210518). Primers were purchased from Integrated DNA Technologies (Supplementary Table 1).

**Recombinant HR2 protein production**. Human MFN2 residues 678–757 corresponding to the HR2 domain were cloned into a pET-28 vector fused to a His-tag and transformed into BL21(DE3) CodonPlus (DE3)-RIPL *E. coli* cells. Cells were grown at 37 °C in 1 L of LB media to an OD$_{600}$ of 0.8, cells were then harvested and resuspended in 1 L of Luria Broth media or M9 media supplemented with 1.5 gr/L of $^{15}$N ammonium chloride grown for 45 min at 37 °C and induced at 18 °C for 16 h with 1 mM isopropyl β-d-1-thiogalactopyranoside. MFN2-HR2 or $^{15}$N-MFN2-HR2 domain was purified from bacterial pellets by high-pressure homogenization in lysis buffer (20 mM Tris.HCl pH 7, 250 mM KCl, 25 mM imidazole,

and Roche complete EDTA free protease inhibitor cocktail) and ultracentrifuged at $45,000 \times g$ for 45 min. The supernatant was applied to pre-equilibrated 1 mL HisPur Ni-NTA Resin washed in lysis buffer and eluted using elution buffer (20 mM Tris.HCl pH 6, 250 mM KCl, 400 mM imidazole). MFN2-HR2 or [15]N-MFN2-HR2 was further purified by size exclusion chromatography (Superdex 75 Increase 10/300 GL column) in gel filtration buffer (20 mM potassium phosphate pH 6, 150 mM KCl). Fractions containing the MFN2-HR2 domain were confirmed by SDS-PAGE, pooled, and concentrated to 50 μM in NMR buffer (20 mM potassium phosphate pH 6, 150 mM KCl, 10% $D_2O$) using a 10 KDa cut-off Centricon spin concentrator (Millipore) for prompt use in biochemical and NMR studies.

**NMR experiments**. The uniformly [15]N-labeled protein samples were prepared by growing the bacteria in a minimal medium, as described above. Correlation [1]H-[15]N-HSQC spectra of 50 μM MFN2-HR2 in the presence and absence of MASM7 or MFI8 or 367–384Gly or 398–418Gly were recorded on a BRUKER AVANCE IIIHD 600 MHz system equipped with a 5 mm H/F-TCI CryoProbe at 25 °C. All experiments were performed using an independent sample for each experimental measurement as a 400 μL sample in a 5 mm Shigemi; all samples were DMSO matched with 2% $d_6$-DMSO. Spectra were processed using qMDD (mddnmr v2.0) and NMRPIPE and analyzed using Analysis (CCPNMR)[55,56].

**Microscale thermophoresis**. Freshly purified His-tagged MFN2-HR2 domain was used for Microscale Thermophoresis (MST) binding studies. For protein labeling, a fresh stock of 5 μM His-tag-RED-tris-NTA 2nd generation dye (Nanotemper) in 25 mM Hepes pH 7.5, 100 mM NaCl, 0.005% Tween-20 (assay buffer) was used to label 500 nM of MFN2-HR2 in the same buffer. The labeling reaction was incubated for 30 min at RT and centrifuged at $15,000 \times g$ at 4 °C for 10 min. Labeled protein from the supernatant was kept on ice and used immediately. Compounds were serially diluted in 100% DMSO. Immediately before mixing with labeled protein, the dilution series was transferred in assay buffer to reach 2% DMSO. 10 μl of each compound in the final series was mixed with 10 μl of labeled protein and samples were incubated for 5 min at 30 °C before MST measurement using a MonolithNT.115 instrument (Nanotemper). Peptides were diluted in assay buffer plus 2% DMSO and treated as above. Final MST conditions: 25 mM Hepes pH 7.5, 100 mM NaCl, 0.005% Tween-20, 1% DMSO, 125 nM labeled HR2, 25 nM NTA dye, temperature 30 °C, LED power 40%, MST power Medium, before MST 3 s, MST-on, 10 s, after MST: 1 s. Samples were measured in standard capillaries (Nanotemper) and showed no aggregation according to post-run analysis using the Monolith data collection software (Nanotemper). Kd values were obtained from non-linear regression fits of normalized data to a three or four parameter logistic curve, using GraphPad Prism 9.

**In vitro MFN2 oligomerization**. MEFs were seeded in $8 \times 15$ cm$^2$ dishes and grown at ~90% confluence. Then, mitochondria were isolated according to previously published protocol[28]. Briefly, cells were harvested, pelleted and washed with cold PBS. Then, cells were resuspended in cold mitochondrial isolation buffer (0.2 M sucrose, 10 mM Tris-MOPS pH 7.4, 1 mM EGTA, 5 mM Mg(OAc)$_2$, 50 mM KOAc, 1 × HALT protease inhibitors, 0.5 mM PMSF) and homogenized in dounce homogenizer with 20 strokes. Isolated mitochondria were incubated with 2 mM GTP, 10 μM MASM7 and 40 μM MFI8 at 37 °C for 30 min. Equal volume of 2× lysis buffer was added to each reaction to have final concentration of (50 mM Bis-Tris, 50 mM NaCl, 10% Glycerol, and 1% wt/vol Digitonin), then samples were incubated on ice for 15 min. Lysates were centrifuged at $16,000 \times g$ at 4 °C for 30 min. Subsequently, supernatant was mixed with NativePAGE 5% G-250 Sample Additive to a final concentration of 0.25%.

**Blue native PAGE**. The procedure for the Blue native PAGE (BN-PAGE) electrophoresis was adapted from previously published protocol[28]. Lysates from the in vitro MFN2 oligomerization experiment were resolved with a Novex Native PAGE 4–16% Bis-Tris Protein Gels (Invitrogen) at 4 °C. The gels were firstly run in the dark cathode buffer at 40 V for 30 min and then at 100 V for 30 min. Then, the dark cathode buffer was replaced with light cathode buffer and the gels were run at 100 V for 30 min and 200 V for 1 h and 50 min. Next, the gels were transferred to polyvinylidene fluororide (PVDF) membranes at 30 V for 16 h using a transfer buffer (Tris 25 mM, 192 mM glycine, 20% methanol). Once the transfer was completed, the membranes were incubated 8% acetic acid for 15 min and subsequently washed with water for 5 min. Then, membranes were dried at 37 °C for 20 min, rehydrated in 100% methanol, and washed with water. Finally, membranes were blocked for 1 h in PBS-T containing 5% milk and 1% BSA and incubated overnight with an anti-MFN2 antibody (Cell Singaling; # 9482S) at a 1:1000 dilution.

**Cellular extract thermal shift assay (CETSA)**. MEFs were seeded in $8 \times 15$ cm$^2$ dish and grown at ~90% confluence. The media was then removed and cells were washed with PBS. Cells were harvested using a cell scraper and centrifuged at $300 \times g$ in 4 °C for 3 min. Then, cells were resuspended in 4 ml PBS. A suspension of 1 ml cells was treated with MASM7 (100 μM), MFI8 (100 μM), the combination

of both compounds or DMSO (100 μM), and left rotating for 1 h at room temperature. Upon completion of the treatment, 50 μl of the suspension of cells were transferred to PCR tubes and heated in a Biorad C1000 Touch Thermal Cycler for 3 min using a temperature gradient (25, 37, 39, 42, 46, 51, 56, and 59 °C). All cells were lysed by four cycles of freeze-thawing using liquid nitrogen. Samples were then centrifuged at $20,000 \times g$ for 15 min. Equal volumes of supernatants were run on 10-well 4–12% NuPAGE SDS-PAGE gels (Invitrogen) and analyzed by western blot. ACTIN, which is temperature insensitive under these conditions, served as loading control. Results were quantitated by densitometric analysis as described below.

**Densitometric analysis**. Densitometric data from western blot scanned films were obtained using Image Studio software (LI-COR). Data were corrected to loading control (total ACTIN) and normalized to the treated bands that correspond to 25 °C (100%) and blot backgrounds (0%). Tm values were obtained from non-linear regression fits of normalized data to a four-parameter logistic curve (4PL), using GraphPad Prism 8.

**Western blot**. Western blots were performed from whole cell lysates (WCL) prepared in lysis buffer containing 50 mM Tris-HCl pH 7.5, 1% Triton X-100, 150 mM NaCl, 5 mM MgCl$_2$, 1 mM EGTA, and 10% glycerol in the presence of protease inhibitor cocktail (Roche). WCL were separated on a 4–12% NuPAGE MES gel (Invitrogen), transferred into a PVDF membrane, block for 1 h, and immunoblot with the corresponding antibodies.

**Mitochondrial polarization assay (mtΔΨ) in cellulo**. MEFs ($10^4$ cells/well) were seeded in a 96-well black plate and treated with MASM7, MFI8, or ABT-737/S63845 for 6 h. Following treatments, cells were stained with 250 nM TMRE (Sigma; Cat. 87917) for 20 min at 37 °C. Subsequently, cells were washed with thrice with PBS. Fluorescence intensity was detected by a M1000 microplate reader (TECAN, Ex: 540 nm/Em: 579 nm).

**Mitochondrial polarization assay (mtΔΨ) in isolated mitochondria**. Mitochondria were isolated based on established protocol[57]. Isolated mitochondria (50 μg) were treated with MASM7 (10 μM, 30 min) or CCCP (30 μM, 30 min). Subsequently, mitochondria were washed with twice with PBS. Fluorescence intensity was detected by a M1000 microplate reader (TECAN, Ex: 540 nm/Em: 579 nm).

**Mitochondrial respiration in cellulo**. Mitochondrial oxygen consumption rates (OCR) were assessed using a XF24 Analyzer (Seahorse Biosciences, Billerica MA, USA). In brief, $3 \times 10^4$ MEFs were cultured in a XF24 cell culture microplate containing DMEM supplemented with 10% fetal bovine serum. Cells were treated with MASM7 (1 μM) or MFI8 (20 μM) 6 h prior to OCR analysis. Mitochondrial respiration was assessed by the sequential addition of oligomycin (1 μM), carbonyl cyanide-4-(trifluoromethoxy)phenylhydrazone (FCCP, 2 μM) and rotenone (1 μM)-antimycin (1 μM) as previously reported[58]. OCR was normalized to cell number or total protein content per well for each condition tested.

**Mitochondrial respiration in isolated mitochondria**. Mitochondria were isolated based on established protocol[57]. Isolated mitochondria were treated with MFI8 (20 μM) for 30 min prior to OCR analysis. OCR was measured using the Mitocell (MT200), a Clarktype electrode from Strathkelvin instruments. Isolated mitochondria (50 μg) from murine cardiomyocytes were loaded into the 50 μl magnetically stirred respiration chamber containing EBm buffer (1 M sucrose, 0.01 M Tris/HCl, 1 M MgCl$_2$, 0.1 M EGTA/Tris, 2 mM KH$_2$PO$_4$, pH 7.4). Glutamate and malate were added to the vessel at final concentrations of 0.25 and 0.125 M, respectively. Respiration was measured after the addition of substrates (glutamate and malate) and termed "state 2 respiration". ADP was then added at a final concentration of 800 μM to measure "state 3 respiration." Addition of oligomycin at final concentration 800 ng/ml was used to quench the reaction and measure proton leakage. Finally, FCCP was added at final concentration 200 nm to measure maximum (uncoupled) respiration.

**Metabolomics**. MEFs were treated with MASM7 and MFI8. Cells were harvested using a cell scraper and centrifuged at $300 \times g$ in 4 °C for 3 min. Each cell pellet sample was suspended in 250–700 μL of 80% aqueous methanol in an Eppendorf tube. The samples were vortex mixed for 15 s and sonicated in an ice-water bath for 5 min, followed by centrifugal clarification at $20,000 \times g$ and 5 °C in an Eppendorf 5424R centrifuge. The clear supernatants were collected. A standard stock solution of TCA cycle carboxylic acids, NAD and NADH was prepared in 80% methanol as S1. This standard solution S1 was serially diluted 1 to 5 (v/v) with the same solvent to make standard solutions S2 to S10. 20 μL of each standard solution and an aliquot of the clear supernatant from each cell was mixed with 20 μL of an internal standard solution containing 9 [13]C- or deuterium labeled analogs of the TCA cycle carboxylic acids (except isocitric acid), 20 μL of 200 mM 3-NPH solution and 20 μL of 150 mM of EDC solution. The mixtures were allowed to react at 30 °C for 30 min. After reaction, 120 μL of water was added to each solution. 10 μL of the

resultant solutions was injected into a C18 UPLC column to quantitate the TCA cycle carboxylic acids by UPLC-MRM/MS with (−) ion detection, according to the established procedure[59].

**Caspase-3/7 activation assay.** Cells ($5 \times 10^3$ cells/well) were seeded in a 96-well white plate and treated with MASM7, MFI8. Caspase 3/7 activation was measured after 6 h by addition of the Caspase-Glo 3/7 reagent according to the manufacturer's protocol (Promega). Luminescence was detected by a F200 PRO microplate reader (TECAN). Caspase assays were performed in at least triplicate and the data normalized to vehicle-treated control wells. Dilutions of MASM7 or MFI8 were performed using a TECAN D300e Digital Dispenser from 10 mM stocks.

**Cytochrome c release.** MEFs were seeded in a 10 cm$^2$ dish and grown at ~70% confluence. Following 6 h treatment with MFI8, cells were harvested and lysed in digitonin buffer [20 mM Hepes, pH 7.2, 10 mM KCl, 5 mM MgCl$_2$, 1 m MEDTA, 1 mM EGTA, 250 mM sucrose, 0.025%Digitonin (from 5% w/v stock) and complete protease inhibitors (Roche Applied Science)] on ice for 10 min. Then, supernatants were isolated by centrifugation at $15,000 \times g$ for 10 min. Mitochondrial pellets solubilized in 1%Triton X-100/PBS for 1 h at 4 °C. Solubilized Pellets were subjected to a $20,000 \times g$ spin for 10 min. Samples were prepared for western blot analysis and separated by 4–12% NuPage (Life Technologies).

**Cell viability assay.** Cells ($5 \times 10^3$ cells/well) were seeded in a 96-well white plate and treated with serial dilutions of MASM7 or MFI8. Cell viability was assayed after 72 h by addition of CellTiter-Glo reagent according to the manufacturer's protocol (Promega). Luminescence was measured using a F200 PRO microplate reader (TECAN). Viability assays were performed in at least triplicate and the data normalized to vehicle-treated control wells. Dilutions of MASM7 or MFI8 were performed using a TECAN D300e Digital Dispenser from 10 mM stocks.

**Cell death assay.** Cells ($2 \times 10^5$ cells/well) were seeded in a 6-well plate and treated with the indicated drugs for 6 h. Cells were dissociated using Accutase (Thermo Fisher; Cat. #00-4555-56) in order to avoid accidental exposure of phosphatidylserine on the outer plasma membrane. Cell death was evaluated with Dead Cell Apoptosis Kit with Annexin V Alexa Fluor™ 488 & Propidium Iodide (Thermo Fisher; Cat. # V13241) according to the manufacturer's protocol. Data was acquired by BD LSRII flow cytometer system using BD FACSDiva software. Data was analyzed by FlowJo (BD).

**Minority MOMP.** U2OS cells (100,000) were seeded onto glass coverslips and transfected with 250 ng each of CytoGFP and Mito-mCherry. 16 h after transfection, cells were treated with compounds for 3 h and then fixed in 4% paraformaldehyde. Cells were imaged on a Zeiss LSM880 with Airyscan using a 63 × 1.4NA objective. Subsequent to acquisition, images were processed using the Airyscan processing function in ZEN software. Minority MOMP was quantified by manually scoring the co-localisation of CytoGFP and mito-mCherry.

**Analysis of mitochondrial biogenesis-related genes by q-PCR.** Total RNA was extracted by using RNeasy Mini Kit (Qiagen) followed by cDNA synthesis using the SuperScript IV VILO Master Mix (Thermo Scientific), both performed according to the manufacturer's protocol. For the q-PCR reaction, Power SYBR Green master mix (Thermo Scientific) was used according to the manufacturer's instructions. Briefly, each reaction consisted of 10 ng cDNA, 5 μL Power SYBR Green master mix, 200 nM primers (forward and reverse), and RNase-free water up to 10 μL. q-PCR was performed on the ViiA 7 Real-Time PCR System (Thermo Scientific) with the following cycle parameter: 95 °C for 10 min, 40 cycles of 95 °C for 15 s, and 60 °C for 1 min. q-PCR products were analyzed by melting curves for unspecific products or primer dimer formation. *Rpl39* was used as housekeeping gene and $2^{-\Delta\Delta CT}$ method was applied to determine the relative mRNAs expression. Primers were purchased from Integrated DNA Technologies (Supplementary Table 2).

**Determination of mtDNA to genomic DNA ratio by q-PCR.** DNA was extracted using DNeasy Blood and Tissue kit (Qiagen) according to the manufacturer's protocol. q-PCR was used to determine the ratio of mitochondrial DNA (mtDNA) and genomic DNA (gn DNA). Each reaction consisted of 5 ng of DNA, 5 μL Power SYBR Green master mix (Thermo Scientific), 200 nM primers (forward and reverse), and RNase-free water up to 10 μL. q-PCR was performed on the ViiA 7 Real-Time PCR System (Thermo Scientific) at 95 °C for 10 min, 40 cycles of 95 °C for 15 s, and 60 °C for 1 min, followed by melting curve analysis. $2^{-\Delta\Delta CT}$ method was applied to determine the mtDNA/nDNA ratio, being $\Delta CT = CT(mtDNA\ gene) - CT(nDNA\ gene)$. Primers were purchased from Integrated DNA Technologies (Supplementary Table 3).

**RNA-seq.** Sequencing results were demultiplexed and converted to FASTQ format using Illumina bcl2fastq software. The sequencing reads were adapter and quality

trimmed with Trimmomatic[60] and then aligned to the mouse genome (build mm10/GRCm38) using the splice-aware STAR aligner[61]. The featureCounts program[62] was utilized to generate counts for each gene based on how many aligned reads overlap its exons. These counts were then normalized and used to test for differential expression using negative binomial generalized linear models implemented by the DESeq2 R 1.26.0 package[63].

**Chemical syntheses.** All chemical reagents and solvents were obtained from commercial sources and used without further purification. FastWoRX™ was purchased from Faster Chemistry LLC. Microwave reactions were performed using an Anton Paar Monowave 300 reactor. Chromatography was performed on a Teledyne ISCO CombiFlash $R_f$ 200i using disposable silica cartridges. Analytical thin layer chromatography (TLC) was performed on Merck silica gel plates and compounds were visualized using UV. NMR spectra were recorded on a Bruker 600 spectrometers. $^1$H chemical and $^{13}$C chemical shifts (δ) are reported relative to tetramethyl silane (TMS, 0.00 ppm) as internal standard or relative to residual solvent signals. Mass spectra were recorded by the Proteomics Facility at the Albert Einstein College of Medicine.

Synthesis of compound 8 (MASM7). See synthetic route at Supplementary Fig. 16a.

Synthesis of compound 4:

Step 1. Cyclopropanecarbohydrazide (2): A mixture of methyl cyclopropanecarboxylate (26.31 g, 26.28 mmol) of 1 and hydrazine hydrate (30 g) was refluxed for 12 h and then placed in vacuum desiccators over sulfuric acid for several days. The crude product was recrystallized from benzene containing a small amount of ethanol and yielded hydrazide 2 (22.4 g, 85%).

Step 2. 2-(cyclopropanecarbonyl)-N-phenylhydrazine-1-carbothioamide (3). Hydrazide 2 (2.08 g, 20.8 mmol) and phenyl isothiocyanate (2.16 g, 16.0 mmol) were dissolved in THF (80 mL), and the mixture was heated to reflux for 7 hours. After cooling, the insoluble product was collected by filtration and washed with water, yielding 3 (3.52 g, 94%).

Step 3. 5-cyclopropyl-4-phenyl-4H-1,2,4-triazole-3-thiol (4). A solution of potassium hydroxide (2.16 g, 38.5 mmol) in water (50 mL) was stirred while 3 (3.50 g, 14.9 mmol) was added. The solution was warmed on a steam bath for 1 hour. After cooling, the solution was poured into a dilute hydrochloric acid solution. The insoluble product was collected by filtration and washed with water, giving 4 (2.36, g 73%).

$^1$H NMR (400 MHz, DMSO-d6) δ 13.60 (s, 1H), 7.55 (m, 3H), 7.46 (d, 2H), 1.47 (m, 1H), 0.90 (m, 2H), 0.83 (m, 2H). $^{13}$C NMR (101 MHz, DMSO-d6) δ 168.06, 154.14, 134.28, 129.83, 128.80, 7.47, 6.73. APSI MS: calculated for C$_{11}$H$_{12}$N$_3$S (M + H)$^+$ 218.1 found 218.2.

Synthesis of compound 8:

Step 4. 2-amino-5,6-dihydro-4H-cyclopenta[b]thiophene-3-carboxamide (6). To the stirred solution of cyclopentanone (25.71 g, 306 mmol), 2-cyanoacetamide 5 (25.0 g, 297 mmol), sulfur powder (9.80 g, 306 mmol) in ethanol (400 ml) was added morpholine (53.3 mL, 618 mmol) and the reaction mixture was stirred at room temperature for 6 h. The reaction mixture was concentrated, diluted with EtOAc and washed with H$_2$O (2 × 300 mL). The separated organic layer was dried over anhydrous Na$_2$SO$_4$, evaporated and crystallized from ethanol to get compound 6 (29.52 g, 53%).

Step 5. 2-(2-chloropropanamido)−5,6-dihydro-4H-cyclopenta[b]thiophene-3-carboxamide (7). To a mixture of 6 (3.00 g, 16.5 mmol) and triethylamine (1.83 g, 18.1 mmol) in CH$_2$Cl$_2$ (20 mL) was added 2-chloropropionyl chloride (2.30 g, 18.1 mmol) slowly at 0 °C, and the resulting mixture was stirred at room temperature for 2 h. The reaction mixture was concentrated to half of the volume to afford the solid product, which was filtered, washed with CH$_2$Cl$_2$ (2 mL), and dried to provide compound 7 (3.68 g, yield 82%).

$^1$H NMR (DMSO-d6, 400 MHz) δ 12.67 (s, 1H), 7.66 (s, 1H), 6.74 (s, 1H), 4.97 (q, 1H), 2.92 (t, 2H), 2.80 (t, 2H), 2.35 (p, 2H), 1.67 (d, 3H),. $^{13}$C NMR (101 MHz, DMSO-d6) δ 167.45, 166.50, 147.80, 139.74, 132.83, 112.51, 55.12, 29.48, 28.73, 28.16, 22.04. APSI MS: calculated for C$_{11}$H$_{14}$ClN$_2$O$_2$S (M + H)$^+$ 273.1 found 273.0.

Step 6. 2-(2-((5-cyclopropyl-4-phenyl-4H-1,2,4-triazol-3-yl)thio)propanamido) −5,6-dihydro-4H-cyclopenta[b]thiophene-3-carboxamide (8). To a mixture of compound 4 (3.30 g, 15.2 mmol) and KOH (0.940 g, 16.8 mmol) in methanol (100 mL), compound 7 (5.40 g, 19.8 mmol) was added and the reaction mass was vigorously stirred for 1 hour at 35 °C. The volatiles were evaporated, and the residue was diluted with water. The product was extracted with dichloromethane. The organic extract was dried over Na$_2$SO$_4$ and evaporated under reduced pressure. The residue was purified by silica gel column chromatography to give the target compound 8 (MASM7, 2.50 g (5.51 mmol), 36%).

$^1$H NMR (DMSO-d6, 400 MHz) δ 0.91 (m, 2H), 1.48 (d, 3H), 1.54 (m, 1H), 2.34 (m, 2H), 2.78 (t, 2H), 2.89 (t, 2H), 4.35 (q, 1H), 6.66 (s, 1H), 7.53 (m, 6H), 12.35 (s, 1H). $^{13}$C NMR (101 MHz, DMSO-d6) δ 167.69, 167.34, 157.86, 148.26, 147.53, 139.61, 133.42, 132.38, 130.26, 127.88, 112.08, 45.32, 29.53, 28.74, 28.18, 18.14, 7.89, 6.25. APSI MS: calculated for C$_{22}$H$_{24}$N$_5$O$_2$S$_2$ (M + H)$^+$ 454.1 found 454.2.

Synthesis of 4-Chloro-2-(1-((2,3-dimethylphenyl)amino)ethyl)phenol (11 or MFI8). See the synthetic route at Supplementary Fig. 16b.

4-Chloro-2-(1-((2,3-dimethylphenyl)amino)ethyl)phenol (11 or MFI8): 1-(5-chloro-2-hydroxyphenyl)ethan-1-one (880 mg, 5.16 mmol), 2,3-dimethylaniline

(940 mg, 7.74 mmol), acetic acid (4.8 mL) and methanol (1.2 mL) were combined in a microwave vial. The vial was capped and heated to 100 °C for 120 min. After cooling to room temperature. NaBH$_3$CN (486 mg, 7.74 mmol) was added and the mixture was stirred for 30 min. The reaction mixture was diluted with water (100 mL) and FastWoRX (4.5 g) was added along with CH$_2$Cl$_2$ (15 mL). The CH$_2$Cl$_2$ was then removed on rotavap then the solids were filtered and washed with water, aq. NaHCO$_3$, water, and finally air-dried. The dry polymer was loaded in a cartridge and the product was purified by column chromatography (12 g silica; 0–30% CH$_2$Cl$_2$ in hexanes). In some cases, the product was contaminated with excess primary aniline at this stage. The impurity can be removed by dissolving in minimum CH$_2$Cl$_2$ and adding hexanes to precipitate the product. The liquid phase is then removed and the solids were washed with hexanes to give pure product. The fractions containing the product were collected and reduced to about half the volume before HCl (1 M in Et$_2$O; 10 mL) was added, causing precipitation of the HCl salt which was collected by filtration (440 mg, 1.41 mmol, 27%).

TLC: $R_f$ = 0.52 (Hexanes:CH$_2$Cl$_2$ 1:2; UV). $^1$H NMR (600 MHz, D$_2$O): δ 7.24 (d, $J$ = 2.6 Hz, 1H), 7.23–7.17 (m, 2H), 7.06 (t, $J$ = 7.8 Hz, 1H), 6.95 (d, $J$ = 8.0 Hz, 1H), 6.78 (d, $J$ = 8.7 Hz, 1H), 4.90 (q, $J$ = 6.9 Hz, 1H), 2.21 (s, 3H), 2.20 (s, 3H), 1.65 (d, $J$ = 7.0 Hz, 3H). $^{13}$C NMR (151 MHz, D$_2$O): δ 153.34, 140.28, 131.42, 131.01, 130.73, 130.66, 128.66, 126.41, 124.59, 123.14, 121.81, 117.27, 58.19, 19.36, 15.71, 13.15. ESI-MS: calculated for C$_{16}$H$_{19}$ClNO (M + H)$^+$ 276.1150 found 276.1150.

### Synthesis of 2-(1-((2,3-Dimethylphenyl)amino)ethyl)phenol (12 or MFI22).

2-(1-((2,3-Dimethylphenyl)amino)ethyl)phenol (MFI22): synthesized following the general procedure from 1-(2-hydroxyphenyl)ethan-1-one (200 mg, 1.47 mmol) and 2,3-dimethylaniline (240 mg, 1.98 mmol) yielding 32 mg (9%).

TLC: $R_f$ = 0.47 (Hexanes:CH$_2$Cl$_2$ 1:2; UV). 1H NMR (600 MHz, D$_2$O) δ 7.34–7.25 (m, 3H), 7.13 (t, $J$ = 7.8 Hz, 1H), 7.04 (d, $J$ = 8.0 Hz, 1H), 6.94 (t, $J$ = 7.6 Hz, 1H), 6.92 (d, $J$ = 8.6 Hz, 1H), 4.99 (q, $J$ = 6.9 Hz, 1H), 2.29 (d, $J$ = 14.1 Hz, 6H), 1.73 (d, $J$ = 7.0 Hz, 3H). $^{13}$C NMR (151 MHz, D2O) δ 154.43, 140.19, 131.59, 131.07, 130.92, 130.71, 128.99, 126.38, 121.78, 121.58, 120.62, 115.87, 58.77, 19.37, 15.89, 13.14. ESI-MS: calculated for C$_{16}$H$_{20}$NO (M + H)$^+$ 242.1139 found 242.1538.

### Synthesis of 4-Chloro-2-(((2,3-dimethylphenyl)amino)methyl)phenol (13 or MFI23).

4-Chloro-2-(((2,3-dimethylphenyl)amino)methyl)phenol (13 or MFI23): synthesized following the general procedure from 5-chloro-2-hydroxybenzaldehyde (204 mg, 1.30 mmol) and 2,3-dimethylaniline (240 mg, 2.0 mmol) yielding 34 mg (10%).

TLC: $R_f$ = 0.46 (Hexanes:CH$_2$Cl$_2$ 1:2; UV). $^1$H NMR (600 MHz, D$_2$O) δ 7.35–7.31 (m, 2H), 7.22 (t, $J$ = 7.8 Hz, 1H), 7.18 (d, $J$ = 2.7 Hz, 1H), 7.11 (d, $J$ = 8.0 Hz, 1H), 6.93 (d, $J$ = 8.7, 1H), 4.53 (s, 2H), 2.31 (s, 3H), 2.28 (s, 3H). $^{13}$C NMR (151 MHz, D$_2$O) δ 154.25, 140.28, 132.39, 131.45, 131.23, 131.03, 130.39, 126.79, 124.14, 120.87, 118.48, 116.76, 50.74, 19.33, 12.73. ESI-MS: calculated for C$_{15}$H$_{17}$ClNO (M + H)$^+$ 262.0993 found 262.0993.

### Synthesis of 4-Chloro-2-((m-tolylamino)methyl)phenol (14 or MFI24).

4-Chloro-2-((m-tolylamino)methyl)phenol (14 or MFI24): synthesized following the general procedure from 5-chloro-2-hydroxybenzaldehyde (204 mg, 1.30 mmol) and m-toluidine (209 mg, 1.95 mmol) yielding 50 mg (16%).

TLC: $R_f$ = 0.38 (Hexanes:CH$_2$Cl$_2$ 1:2; UV). $^1$H NMR (600 MHz, D$_2$O) δ 7.39 (t, $J$ = 7.8 Hz, 1H), 7.34 (d, $J$ = 7.7 Hz, 1H), 7.32 (dd, $J$ = 8.7, 2.6 Hz, 1H), 7.18 (s, 1H), 7.16–7.13 (m, 2H), 6.93 (d, $J$ = 8.7 Hz, 1H), 4.57 (s, 2H), 2.35 (s, 3H). $^{13}$C NMR (151 MHz, D2O) δ 154.06, 140.97, 133.74, 131.40, 131.10, 130.49, 129.85, 124.12, 123.15, 119.60, 118.47, 116.75, 51.21, and 20.22. ESI-MS: calculated for C$_{14}$H$_{15}$ClNO (M + H)$^+$ 248.0837 found 248.0837.

### Synthesis of 4-Chloro-2-(1-(phenylamino)ethyl)phenol (15 or MFI25).

4-Chloro-2-(1-(phenylamino)ethyl)phenol (15 or MFI25): synthesized following the general procedure from 1-(5-chloro-2-hydroxyphenyl)ethan-1-one (222 mg, 1.30 mmol) and aniline (182 mg, 2.0 mmol) yielding 22 mg (7%).

TLC: $R_f$ = 0.39 (Hexanes:CH$_2$Cl$_2$ 1:2; UV). $^1$H NMR (600 MHz, D$_2$O) δ 7.40–7.35 (m, 3H), 7.28–7.23 (m, 2H), 7.18 (dd, $J$ = 8.6, 2.5 Hz, 1H), 7.15 (d, $J$ = 2.6 Hz, 1H), 6.80 (d, $J$ = 8.7 Hz, 1H), 4.90 (q, $J$ = 6.9 Hz, 1H), 1.68 (d, $J$ = 7.0 Hz, 3H). $^{13}$C NMR (151 MHz, D$_2$O) δ 153.12, 133.06, 130.52, 129.92, 129.85, 128.81, 124.56, 123.15, 123.03, 117.26, 59.30, and 15.97. ESI-MS: calculated for C$_{14}$H$_{15}$ClNO (M + H)$^+$ 248.0837 found 248.0836.

### Synthesis of N-(1-(3-chlorophenyl)ethyl)−2,3-dimethylaniline (16 or MFI26).

N-(1-(3-chlorophenyl)ethyl)−2,3-dimethylaniline (16 or MFI26): synthesized following the general procedure from 1-(3-chlorophenyl)ethan-1-one (200 mg, 1.30 mmol) and 2,3-dimethylaniline (235 mg, 1.9 mmol) yielding 30 mg (9%).

TLC: $R_f$ = 0.31 (Hexanes:CH$_2$Cl$_2$ 1:2; UV). $^1$H NMR (600 MHz, D$_2$O) δ 7.48 (m, 1H), 7.37 (t, $J$ = 7.9 Hz, 1H), 7.34 (t, $J$ = 1.9 Hz, 1H), 7.33 (d, $J$ = 7.5 Hz, 1H), 7.25 (d, $J$ = 7.8 Hz, 1H), 7.15 (t, $J$ = 7.8 Hz, 1H), 6.89 (d, $J$ = 8.0 Hz, 1H), 2.29 (s, 3H), 2.14 (s, 3H), 1.85 (d, $J$ = 6.9 Hz, 3H). NOTE: The signal from the benzylic proton is obscured by the solvent signal. $^{13}$C NMR (151 MHz, D$_2$O) δ 140.42, 136.50, 134.15, 131.39, 131.08, 130.58, 130.47, 129.91, 128.30, 126.67, 126.52,

121.94, 62.35, 19.34, 16.90, and 13.14. ESI-MS: calculated for C$_{16}$H$_{19}$ClN (M + H)$^+$ 260.1201 found 260.1199.

**Statistical analysis**. The results are presented as mean ± SEM with a minimum of three replicates unless otherwise specified. Statistical analyses were performed by Student's t-tests or one-way or two-way ANOVA using Prism 8.1 (GraphPad software). When the overall ANOVA revealed a significant effect, the data were further analyzed with the Dunnett, Sidak, or Tukey post hoc test to determine specific group differences. $P$ values indicated on the graphs: $*p < 0.05$, $**p < 0.01$, $***p < 0.001$, $****p < 0.0001$.

**Reporting summary**. Further information on research design is available in the Nature Research Reporting Summary linked to this article.

## Data availability

Data generated or analyzed during this study are included in this article and in supplementary information files and are available from the corresponding author. The RNA-seq data generated in this study have been deposited in NCBI Gene Expression Omnibus (GEO) under the accession code GSE186924. The following publicly available data sets were used in the production of this manuscript: PDB ID: 6JFL, 2J69, 6JFL, 6JFK, 5GOE, and 1T3J. Source data are provided with this paper.

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

## Acknowledgements

We thank David Chan for providing MEF cell lines. Studies were supported by NIH grants P01AG031782, R01CA178394, and the Irma T. Hirschl Trust Career Award to E.G., R01HL138475, and Fondation Leducq (RA15CVD04) to R.N.K. L.A.K. is supported by a Foundation Grant by the Canadian Institute for Health Research (CIHR), L.A.K. is a Canada Research Chair in Molecular Cariology, I.R.N. holds a CIHR post-doctoral fellowship award. NMR, chemistry, and imaging resources are supported from NIH grants 1S10OD016305 and P30 CA013330. Cartoons in Fig. 8 were created with Biorender.com.

## Author contributions

E.Z. performed biochemical and cellular studies, microscopy, metabolism, and small molecule design studies. B.A. performed biophysical and molecular dynamics studies, K.V.M.M. performed NMR studies. N.B. performed in silico screening studies. T.P.G performed NMR studies. A.T.O. contributed to cell death studies. I.R.N., V.M., R.N.K., and L.A.K. performed seahorse studies. L.U.N performed compound synthesis and analytical characterization. J.S.R. and S.W.G.T. performed minority MOMP studies. I.D. and A.T. performed RNAseq analysis. Y.C., R.P., A.J.H.W., and R.N.K., contributed to the generation of MFN2 mutant cells and respiration studies. C.M. and P.W. contributed to small molecule screening. E.G. performed in silico screening studies, desiged and supervised the study, and wrote the manuscript with E.Z., which was reviewed and edited by all authors.

## Competing interests

E.Z., N.B., R.N.K., and E.G. have filled patent applications submitted by Albert Einstein College of Medicine covering compounds, compositions, and methods for controlling mitofusins for the treatment of diseases and disorders. E.G. is a co-founder and scientific advisor of BAKX Therapeutics, Selphagy Therapeutics now part of Life Biosciences, and Stelexis Therapeutics. None of the above companies have sponsored this research. All other authors declare no competing interests.
