## [Peer Review File · Nature Communications]

Modulating mitofusins to control mitochondrial function and signalingREVIEWER COMMENTS

Reviewer #1 (Remarks to the Author):

Zacharioudakis et al use a rational pharmacophore approach to in silico screen for small molecules that modulate the fusion activity of MFN1 and MFN2. The authors identify and characterize a Mitofusin Activator Small Molecule, MASM7, that promotes a tethering conformation, and a Mitochondrial Fusion Inhibitor, MIF8, that inhibits inter-MFN interactions. These tools are then used to investigate the effects of elongation and fragmentation on mitochondrial respiration and cell death.

The development of small molecules to specifically regulate mitochondrial fusion is an important contribution to a field that lacks such pharmacological tools. The chemical and cellular characterization indicates that MASM7 and MIF8 directly target the HR2 domain of MFN2, and are not affecting morphology through indirect effects. However, the cell death data are less clearly interpretable and there are instances where the data are overinterpreted and the language should be modified.

Specific comments:

- The CESTA experiments show that MASM7 thermally stabilizes MFN2 ($T_m = 42.9^\circ\text{C} - 45.2^\circ\text{C}$), but MIF8 alone does not ($T_m = 42.9 - 43.7$). Adding both molecules also stabilizes MFN2 ($42.9 - 44.05$), albeit slightly less so than MASM7 alone. The authors conclude that MIF8 is “revoking MASM7-induced MFN2 stabilization”. Is the difference in T_m between single and double compound treatment significant? How should this be interpreted if both proteins are binding to MFN2 in this context? There is no clear conclusion to be drawn from the experiment.

- Similarly, the conclusions drawn from the CESTA experiment that MASM7-induced MFN2 stabilization is a result of increased MFN oligomerization is speculative and not supported by the experiments.

-mtDNA : nuclear DNA ratio is not an appropriate measure for mitochondrial mass in MASM7 and MIF8 treated cells, as mtDNA levels are modified by changes in morphology. A better method would be to use total mitochondrial area and/or to measure total levels of mitochondrial biogenesis proteins such as Tom20.

- Data in Figure 3 indicate that MASM7 depends on residues D725/L727 and that MIF8 depends on S685, but are not sufficient to conclude that the small molecules directly bind these residues.

- Why are the total levels of MFN2 in the BN-PAGE so different between conditions when total MFN levels were not affected when measured by Western Blot?

- There is an increase in cytC release with increased doses of MIF8, which the authors interpret as fragmentation-induced MOMP. However, the mitochondria are already hyper-fragmented at lower doses of MIF8, suggesting that MIF8, and not the mitochondrial fragmentation, may be responsible for MOMP, especially if MFN1/2 double knockout cells don't show the same spontaneous cytC release. If adding MIF8 to MFN1/2 double knockout cells also induces cytC release, it would suggest that MOMP is a result of non-specific MIF8 effects and not due to morphology per se.

- Several experiments are not described in the methods section, most notably, how the mitochondrial aspect ratio was measured. This is a key readout for several experiments and important to explain. Other examples are the experiment using Floxed-DRP1 cells, which is not described in the methods, nor are the BN-PAGE conditions. Please make sure that all experiments in the manuscript are well described in the methods section.

- There are several typos throughout the text. For instance, "acknowledgements" and "contributions" are misspelled on page 26, and "predominantly" on page 18.

Reviewer #2 (Remarks to the Author):

The authors present a well-grounded work on novel modulators for mitochondrial fusion targeting mitofusin and combining in silico screening and biological assays.

As a molecular modelling expert, I will mainly focus my review concerning the in silico work. In the two first result sections, a ligand-based screening is performed to identify a small set of molecules activating or inhibiting the mitochondrial fusion via mitofusin. Those sets of molecules are then subjected to in vitro and in vivo tests. In the next sections, a tremendous in vitro/in vivo work is realized to ensure that hits are binding at the right spots (HSQC-NMR spectra), to get features onto the conformational changes and MFN2 oligomerization, to observe the impact of hits on the mitochondrial respiration and functionality as well as their induction to cell death. The manuscript is really well written and the biological work is highly impressive. However, I would like the authors to provide more insights from the in silico work. Therefore, I recommend the publication of the manuscript after major revision (mainly on the in silico sections). Here are my comments:

1) The authors based their *in silico* study on homology modeling and on Xray data without accounting for the dynamics of the system, could the authors comment on this choice. Could they briefly present and characterize the weak fluctuations of the binding zones to justify such choice.

2) The authors mentioned their previous work on a helical peptide promoting the pro-tethering conformation of MFN2 but did not compare the promotion of the mitochondrial fusion using the new MASM7 compound. Similar comment can be done for the MF18 hit and the helical peptides.

3) For the MASM pharamcophore model, His380 has been included but its role in the interaction with HR2 amino acid is not clear.

4) One major missing point in the *in silico* analysis is the insights at the atomic level by performing at last a docking step for the hit compounds (MASM7 and MF18) followed by short MD simulations, such features could clearly give straightforward insights about inhibition/activation mechanism compared to the apo system considering anti- and pro-tethering conformation. Inclusion of the membrane and observe the impact of the hits onto the structural features would clearly be a major advances in terms of understanding at the atomic level.

5) Data reported in the main manuscript and Extended data quantifying Mito AR in MEF has a +/-SEM are obtained based only on 2 independent biological replicates. Such SEM is extremely large for MASM21 and MASM23, did the authors check further analysis of those compounds ? Why such large SEM is observed for those compounds ?

6) The authors mentioned that MASM7 and MF18 “has not been reported as a hit in previous screens in Pubchem database” but those compounds are commercially available, what are their original usage ?

Reviewer #3 (Remarks to the Author):

The authors identified small molecules that inhibit activity of mitofusins, MFN1 and MFN2, and used them to understand the molecular mechanism of the mitofusion and signaling by the inhibition. In brief, the authors conducted *in silico* screening and selected 18 compounds to experimentally test its inhibition of mitochondrial fusion, and identified MASM7 inhibits MFN1 and MFN2 activity and verified the inhibition by QSAR. In the process of QSAR, the authors also found a compound that promotes

mitochondrial fusion, MFI8. The authors tried to characterize the compound interaction site on HR2 domain by NMR. Based on these observations, the authors concluded the inhibition/promotion ability of these compounds. Next, using these compounds as tools to modulate the “mitofusion function”, the authors characterized mitofusion effect on mitochondrial respiration and functionality, and mitochondrial outer membrane permeabilization and cell death. Reproducibility of most of the experiments, but not all, were confirmed. Multiple experiments were done to derive conclusions. Overall, finding the compounds and utilization of compounds are exciting. However, I have four major concerns.

1. Unfortunately, their NMR data is not consistent within the presented results. The NMR spectrum of MFN2-HR2 domain shown in Extended Fig. 3c (a part is shown as Fig. 3c) is different from that in Extended Fig. 3a (a part is shown as Fig. 3a). The spectrum in Extended Fig. 3c clearly indicates partial fragmentation, as evident from the two resonances at 1H 7.72 ppm and 15N 127 ppm, and 1H 7.68 ppm and 15N 128 ppm, as well as the intrinsic resonance at 1H 7.9 ppm and 15N 127.7 ppm. The partial degradation may be a reason why the authors observed more significant changes in Fig. 3c, compared to Fig. 3a.

2. If the compounds really bind to the protein, it has to show more pronounced changes. Suppose that the tiny intensity or chemical shift changes are due to a 5~10% bound form of the 50 uM protein in the presence of a 200 uM inhibitor. Then, the compound dissociation constant is calculated to be 2~4 mM using a simple 1:1 interaction model (assume that the experiment with/without the compounds were done using the same experimental parameters). Thus, the NMR data presented in this manuscript show no effect or interaction at a non-specific binding level. Although the authors expected “pi-stacking” or “hydrophobic interaction”, such effect is not evident in the NMR titration data. It is possible that the binding site is different from what was expected, or can be detected only when the full-length protein is used or in the membrane environment. Assays using mutant proteins (Fig. 3e and 3f) cannot tell the binding site because the mutations themselves alter the Mito AR. Thus, in the current manuscript, there is no explicit data that confirm the compound binding sites.

3. In the section of “A small molecule inhibitor of mitochondrial fusion via mitofusins”, details about the interaction site discussed in the second paragraph, for example, “pi-stacking” or “hydrophobic interaction”, is speculative. Given that there is no structural confirmation about the binding site, this binding site discussion in the second paragraph sounds too weak to write in the Results.

4. In the sections of “Modulation of mitofusins activity alters mitochondrial respiration and functionality” and “Inhibition of mitofusins activity induces minority MOMP and primes cells for cell death”, the authors use the word “modulation of mitofusins” or “inhibition of mitofusins” quite generally. Although the discovered compounds in this manuscript exhibit such alternations of mitochondrial respiration and functionality and relations to cell death, it is unknown whether any other inhibitors (or promoters) of mitofusins exhibit the same effect, or only the specific ones that interact

with HR1 or HR2 domain and/or structurally modulate the dimerization do the same effect. The authors may need more defined discussion. Otherwise, it may sound a bit over-interpretation.

Minor

(1) In the abstract, spelling out SMAC.

(2) It is ideal to cite references in figure captions for panels that were previously found. For example, cartoons in Figure 1a.

(3) Figure 4a and 4c need markers for Oligo, CCCP etc. Otherwise, it is unclear what caused the OCR changes.

Reviewer #4 (Remarks to the Author):

This manuscript describes novel and exciting results on discovery and characterization of small molecules capable of regulating directly either positively or negatively MFN1/2 activity and subsequently mitochondrial fusion by manipulating the MFN1/2 tethering permissive structure. These compounds, will allow a more precise evaluation of the role of MFN1/2 in the pathogenesis of several diseases, thus offering novel pharmacological opportunities. Generally, the authors are providing convincing demonstration of the target engagement using in vitro biochemical and biophysical assays as well as functional assays on cellular level. However, there are a few areas in which the execution of experiments falls short of providing robust conclusions and need improvements before accepting for publication.

The provided SAR studies for the reported activator (MASM7) and inhibitor (MF18) are limited. Although the activity of the synthesized analogues is supporting the generated pharmacophore model to some level, further validation is needed. In particular the HSQC NMR binding studies should be extended with additional mutant proteins based on the pharmacophore models (e.g. L692A which is also used in reconstituted MEFs) as well as by testing several analogs to further validate the importance of the certain functional moieties. In addition, the chemical shifts presented in the HSQC NMR spectra are not very significant. It will be important authors to perform quantitative analysis of the protein-ligand interactions, especially for MASM7 which has EC50 of 75 nM. Furthermore, knowing that MFN1 and 2 are highly homologous proteins, there is lack of discussion/studies to clarify the level of selectivity of the reported compounds. This is important to be known since MFNs have mitochondrial and other functional roles,

beyond mediating mitochondrial fusion.

MFI8 was discovered by virtual screening and pharmacophore model developed based on the reported helical peptide from the MFN2. Thus, comparative studies and testing the peptide and MFI8 in parallel will provide further validation of the generated pharmacophore model (for example comparison of their HSQC NMR spectra), and better understanding of the relative potency of this novel small-molecule inhibitor.

Throughout the manuscript all presented data are obtained by using a single concentration of the agonist (1 μ M) and inhibitor (20 μ M). It is necessary to be demonstrated if these effects are dose-dependent. Thus, certain binding and functional key experiments should be repeated by using concentration range of the small-molecules to show dose-dependent response.

The results from the combination treatment with Smac bivalent mimetic are very interesting. The authors should provide rational why they tested XIAP inhibitor for the synergistic studies and not with other anti-apoptotic inhibitors, for example Bcl-2 family inhibitors.

In the discussion the authors are commenting the opportunity for the development of MFN2 agonists as novel therapeutics for CMT2A. They should extend the discussion to cover the reported mutations in CMT2A and how are related to the pharmacophore/binding model of the reported agonist.

We greatly appreciate all the reviewers' insights and valuable comments. Their comments have been addressed in the revised manuscript as described below.

To facilitate the review process, we indicated in each answer the figure panels or pages where the new results and text are displayed in the revised manuscript. Changes that address the reviewers' comments in the manuscript text are marked in blue fonts.

We hope that with the additional experimental data and revision of the manuscript, the reviewers will find the revised manuscript suitable for publication.

Reviewer #1 (Remarks to the Author):

Zacharioudakis et al use a rational pharmacophore approach to in silico screen for small molecules that modulate the fusion activity of MFN1 and MFN2. The authors identify and characterize a Mitofusin Activator Small Molecule, MASM7, that promotes a tethering conformation, and a Mitochondrial Fusion Inhibitor, MFI8, that inhibits inter-MFN interactions. These tools are then used to investigate the effects of elongation and fragmentation on mitochondrial respiration and cell death.

The development of small molecules to specifically regulate mitochondrial fusion is an important contribution to a field that lacks such pharmacological tools. The chemical and cellular characterization indicates that MASM7 and MFI8 directly target the HR2 domain of MFN2, and are not affecting morphology through indirect effects. However, the cell death data are less clearly interpretable and there are instances where the data are overinterpreted and the language should be modified.

We thank the reviewer for the positive remarks of our work and experimental suggestions to strengthen our study. We hope the additional experiments that we provide have addressed the reviewer's comments.

Specific comments:

- The CESTA experiments show that MASM7 thermally stabilizes MFN2 ($T_m = 42.9\text{C} - 45.2\text{C}$), but MFI8 alone does not ($T_m = 42.9 - 43.7$). Adding both molecules also stabilizes MFN2 ($42.9 - 44.05$), albeit slightly less so than MASM7 alone. The authors conclude that MFI8 is "revoking MASM7-induced MFN2 stabilization". Is the difference in T_m between single and double compound treatment significant? How should this be interpreted if both proteins are binding to MFN2 in this context? There is no clear conclusion to be drawn from the experiment.

We thank the reviewer for the opportunity to clarify further these data. The combination (MFI8+MASM7) treatment appears to stabilize MFN2 in comparison to the Veh if we consider only the mean value of the T_m , while if we consider the standard deviation, based on three independent experiments, along with the mean value (Veh: $T_m=42.9 \pm 0.8$ °C vs MFI8+MASM7: $T_m=44.05 \pm 2.6$ °C), MFN2 is not significantly stabilized by the combination of MASM7 and MFI8. Furthermore, our data clearly showed that MASM7 has a higher capacity to promote MFN2 stabilization in comparison to the Veh (Veh: $T_m=42.9 \pm 0.8$ °C vs MASM7: $T_m=45.2 \pm 0.5$ °C), when treated alone. These data show that MFI8 reduces MFN2 engagement by MASM7.

Therefore, we have decided to amend the text in the revised version of our manuscript and conclude “that MFI8 reduced the capacity of MASM7 to induce MFN2 stabilization” (page 15).

- Similarly, the conclusions drawn from the CESTA experiment that MASM7-induced MFN2 stabilization is a result of increased MFN oligomerization is speculative and not supported by the experiments.

We agree with the reviewer. Thus, we have removed the relevant text as a conclusion from the CESTA experiment in our revised manuscript (page 15).

-mtDNA : nuclear DNA ratio is not an appropriate measure for mitochondrial mass in MASM7 and MIF8 treated cells, as mtDNA levels are modified by changes in morphology. A better method would be to use total mitochondrial area and/or to measure total levels of mitochondrial biogenesis proteins such as Tom20.

We would like to thank the reviewer for this suggestion. To address this point we have measured the total levels of Tomm20 using western blot analysis and found that neither MASM7 nor MFI8 treatment altered significantly the total levels of Tomm20 (Supplementary Fig. 9c,d). Moreover, we used mitotracker green, a dye that stains mitochondria independently of their membrane potential, as a fluorescent marker to assess mitochondrial mass by measuring the fluorescent intensity upon staining with the dye (Supplementary Fig. 9e). Neither MASM7 nor MFI8 altered significantly the fluorescent intensity of mitotracker green.

- Data in Figure 3 indicate that MASM7 depends on residues D725/L727 and that MIF8 depends on S685, but are not sufficient to conclude that the small molecules directly bind these residues.

We agree with the reviewer. Our experiments do not support that MASM7 directly binds to D725/L727 and MFI8 directly binds to S685, but rather that those residues are important for the direct binding of the molecules on the HR2 domain of MFN2. Thus, we have modified accordingly the text in our revised manuscript (page 13).

- Why are the total levels of MFN2 in the BN-PAGE so different between conditions when total MFN levels were not affected when measured by Western Blot?

We agree with the reviewer that the total levels of MFN2 appear different between conditions in the BN-PAGE gel (Fig. 3d). This is because the capacity of MFN2 to oligomerize varies depending on the condition. Specifically, we and others have observed that GTP can induce the formation of MFN2 higher order oligomers (~450 kD) when treated to isolated mitochondria (Samanas et al., Life Sci Alliance, 2020). Thus, all the GTP-treated MFN2 appears as higher total MFN2 levels than non GTP-treated MFN2 under the non-denaturing conditions of the BN-PAGE gel (Fig. 3d). Importantly, we have performed the same experiment as in Fig. 3d but instead of separating the MFN2 oligomers with BN-PAGE gel we used denaturing conditions in an SDS-PAGE gel, which allowed us to detect MFN2 irrespective of its oligomerization status (Supplementary Fig 8a). Our findings now show that the total levels of MFN2 are comparable/similar among the different conditions when detected by western blot (Supplementary Fig 8a). Therefore, we conclude that neither MASM7 nor MFI8 significantly alter the total levels of MFN2 and the differences in the ratio of higher order MFN2 oligomers to dimers that are observed in the with BN-PAGE gel are attributed to the capacity of MASM7 and MFI8 to manipulate MFN2 oligomerization in a GTP dependent manner (Fig. 3d).

- There is an increase in cytochrome c release with increased doses of MIF8, which the authors interpret as fragmentation-induced MOMP. However, the mitochondria are already hyper-fragmented at lower doses of MIF8, suggesting that MIF8, and not the mitochondrial fragmentation, may be responsible for MOMP, especially if MFN1/2 double knockout cells don't show the same spontaneous cytochrome c release. If adding MIF8 to MFN1/2 double knockout cells also induces cytochrome c release, it would suggest that MOMP is a result of non-specific MIF8 effects and not due to morphology per se.

We agree with the reviewer's point about the possibility of non-mitofusin specific effects of MIF8 that can induce MOMP. Thus, we decided to test whether MIF8 can induce cytochrome c release in MFN1/2 double knockout MEFs. MIF8 did not induce any cytochrome c release in MFN1/2 double knockout MEFs (Supplementary Fig. 11c). Furthermore, such result comes in agreement with our previous data that showed MIF8 induces caspase 3/7 activation, an event that is downstream of MOMP and cytochrome c release, in a strictly mitofusin dependent manner (Fig. 5b). Taken all together, these results demonstrate that the induction of MOMP by MIF8 treatment is a result of the specific targeting of MIF8 towards mitofusins rather than other uncharacterized non-mitofusin specific effects of MIF8.

- Several experiments are not described in the methods section, most notably, how the mitochondrial aspect ratio was measured. This is a key readout for several experiments and important to explain. Other examples are the experiment using Floxed-DRP1 cells, which is not described in the methods, nor are the BN-PAGE conditions. Please make sure that all experiments in the manuscript are well described in the methods section.

We thank the reviewer for noticing these missing experimental details. We have now included a detailed explanation about measurement of the mitochondrial aspect ratio and previously non-described experiments in the methods. Changes in the text are highlighted in blue.

- There are several typos throughout the text. For instance, "acknowledgements" and "contributions" are misspelled on page 26, and "predominantly" on page 18.

We thank the reviewer, we corrected several typos.

Reviewer #2 (Remarks to the Author):

The authors present a well-grounded work on novel modulators for mitochondrial fusion targeting mitofusin and combining in silico screening and biological assays. As a molecular modelling expert, I will mainly focus my review concerning the in silico work. In the two first result sections, a ligand-based screening is performed to identify a small set of molecules activating or inhibiting the mitochondrial fusion via mitofusin. Those sets of molecules are then subjected to in vitro and in vivo tests. In the next sections, a tremendous in vitro/in vivo work is realized to ensure that hits are binding at the right spots (HSQC-NMR spectra), to get features onto the conformational changes and MFN2 oligomerization, to observe the impact of hits on the mitochondrial respiration and functionality as well as their induction to cell death. The manuscript is really well written and the biological work is highly impressive. However, I would like the authors to provide more insights from the in silico work. Therefore, I recommend the publication of the manuscript after major revision (mainly on the in silico sections). Here are my comments:

We thank the reviewer for the positive remarks of our work and appreciate his/her suggestions on how to improve our manuscript.

1) The authors based their *in silico* study on homology modeling and on Xray data without accounting for the dynamics of the system, could the authors comment on this choice. Could they briefly present and characterize the weak fluctuations of the binding zones to justify such choice.

To account for the dynamics of the full length MFN2 structure, we performed molecular dynamics simulations (3 independent simulations) of the MFN2 structural model for 250 ns and evaluated the fluctuations of the HR2 binding zones used in the pharmacophore models for MASM and MF1 compounds. The molecular dynamics simulations confirmed that there are weak fluctuations in the binding zones as shown in Supplementary Figure 1. We discuss these findings in page 6 and 9 for each binding zone.

2) The authors mentioned their previous work on a helical peptide promoting the pro-tethering conformation of MFN2 but did not compare the promotion of the mitochondrial fusion using the new MASM7 compound. Similar comment can be done for the MF18 hit and the helical peptides.

That's a great suggestion. We now provide new set of data where we compare MASM7 with 367-384Gly (pro-fusion peptide) and MF18 with 398-418Gly (anti-fusion peptide) in the following assays: a) *in vitro* binding to the HR2 domain using Microscale Thermophoresis (MST) as a readout (Fig. 3a); b) in mitochondrial morphology using mitochondrial aspect ratio (Mito AR) as a readout (Fig. 1h,2f); c) HSQC NMR studies (Supplementary Fig. 5,6). Briefly, both small molecules had a comparable potency with their corresponding peptides in the aforementioned assays.

3) For the MASM pharmacophore model, His380 has been included but its role in the interaction with HR2 amino acid is not clear.

We can see the point of the reviewer that from the view of HR1-HR2 helices in Figure 1c the His380 seems not in the right distance to interact with the Asp725 as we mentioned in the text. However, depending on the conformation of Asp725 and His380 these residues can come very close to interact. We present below another calculated conformation of the MFN2 structural model in which the Asp725 is in close distance to form a hydrogen bond with His380.

4) One major missing point in the *in silico* analysis is the insights at the atomic level by performing at least a docking step for the hit compounds (MASM7 and MF18) followed by short MD simulations, such features could clearly give straightforward insights about inhibition/activation mechanism compared to the apo system considering anti- and pro-tethering conformation. Inclusion of the membrane and observe the impact of the hits onto the structural features would clearly be a major advances in terms of understanding at the atomic level.

We agree with the reviewer that further insights at the atomic level regarding the mechanism of activation/inhibition of MFN2 would be a major advance. Indeed, proper analysis of this system would require inclusion of the native membrane environment of mitofusin proteins in addition to experimental validation of the conformational changes and structures. We believe that this work requires a major dedicated effort as part of a separate study and beyond the work for this manuscript. We hope that reviewer agrees with our view that this cannot be addressed correctly without significant effort and validation.

5) Data reported in the main manuscript and Extended data quantifying Mito AR in MEF has a +/- SEM are obtained based only on 2 independent biological replicates. Such SEM is extremely large for MASM21 and MASM23, did the authors check further analysis of those compounds ? Why such large SEM is observed for those compounds ?

We agree with the reviewer that MASM21 and MASM23 presented large error bars during the quantification of Mito AR. We have now increased the biological replicates that concern the quantification of Mito AR. We found the error bars to be reduced and both compounds have more reproducible results between independent biological experiments. (Supplementary Fig. 2e,3d). These results are also in agreement with the new binding results by Microscale Thermophoresis (MST) in Supplementary Figure 4.

6) The authors mentioned that MASM7 and MFI8 “has not been reported as a hit in previous screens in Pubchem database” but those compounds are commercially available, what are their original usage ?

Indeed, the compounds were available commercially in screening libraries and have not been assigned a target or specific usage to the best of our knowledge. We apologize if this was confusing.

Reviewer #3 (Remarks to the Author):

The authors identified small molecules that inhibit activity of mitofusins, MFN1 and MFN2, and used them to understand the molecular mechanism of the mitofusion and signaling by the inhibition. In brief, the authors conducted in silico screening and selected 18 compounds to experimentally test its inhibition of mitochondrial fusion, and identified MASM7 inhibits MFN1 and MFN2 activity and verified the inhibition by QSAR. In the process of QSAR, the authors also found a compound that promotes mitochondrial fusion, MFI8. The authors tried to characterize the compound interaction site on HR2 domain by NMR. Based on these observations, the authors concluded the inhibition/promotion ability of these compounds. Next, using these compounds as tools to modulate the “mitofusion function”, the authors characterized mitofusion effect on mitochondrial respiration and functionality, and mitochondrial outer membrane permeabilization and cell death. Reproducibility of most of the experiments, but not all, were confirmed. Multiple experiments were done to derive conclusions. Overall, finding the compounds and utilization of compounds are exciting. However, I have four major concerns.

We really appreciate the reviewer’s careful analysis of our work and positive comments about our findings, and we hope the additional experiments that we provide have addressed the reviewer’s concerns.

1. Unfortunately, their NMR data is not consistent within the presented results. The NMR spectrum of MFN2-HR2 domain shown in Extended Fig. 3c (a part is shown as Fig. 3c) is different from that in Extended Fig. 3a (a part is shown as Fig 3a). The spectrum in Extended Fig. 3c clearly indicates partial fragmentation, as evident from the two resonances at ^1H 7.72 ppm and ^{15}N 127 ppm, and ^1H 7.68 ppm and ^{15}N 128 ppm, as well as the intrinsic resonance at ^1H 7.9 ppm and ^{15}N 127.7 ppm. The partial degradation may be a reason why the authors observed more significant changes in Fig. 3c, compared to Fig. 3a.

We thank the reviewer for raising this point. We agree with the reviewer that the NMR spectrum for the MFN2-HR2 domain is different between the previous Fig. 3a and Fig. 3c. We apologize for any confusion, which arose from an impurity co-purified with the MFN2-HR2 in the Fig 3c.

We have repeated the purification of MFN2-HR2 several times with a more stringent selection of fractions of purified HR2 domain after gel filtration chromatography and acquired MFN2 spectra over several hours. We observed good spectra for the HR2 and no evidence of degradation as in Fig.3a (now Supplementary Fig. 5a). Furthermore, we have repeated the HSQC experiment of MFN2-HR2 domain with or without MFI8 and MASM7 (Supplementary Fig. 5a,6a). Again, we observed specific cross peak broadening and cross peak shifting induced by MFI8 and MASM7 at different cross peaks for each compound. We also evaluated the corresponding peptides that are expected to bind to the same HR2 regions as MFI8 and MASM7 under the same conditions and found similar binding profiles with each compound/peptide pair (Supplementary Fig. 5b,6b). These results and other results with MST described below, suggest that MFI8 binds MFN2-HR2 independently of the co-purified protein that was part of the previous sample in Figure 3c.

2. If the compounds really bind to the protein, it has to show more pronounced changes. Suppose that the tiny intensity or chemical shift changes are due to a 5~10% bound form of the 50 μM protein in the presence of a 200 μM inhibitor. Then, the compound dissociation constant is calculated to be 2~4 mM using a simple 1:1 interaction model (assume that the experiment with/without the compounds were done using the same experimental parameters). Thus, the NMR data presented in this manuscript show no effect or interaction at a non-specific binding level. Although the authors expected “ π -stacking” or “hydrophobic interaction”, such effect is not evident in the NMR titration data. It is possible that the binding site is different from what was expected, or can be detected only when the full-length protein is used or in the membrane environment. Assays using mutant proteins (Fig. 3e and 3f) cannot tell the binding site because the mutations themselves alter the Mito AR. Thus, in the current manuscript, there is no explicit data that confirm the compound binding sites.

We thank the reviewer for raising this point. To address these concerns regarding the binding of the compounds in the NMR studies and to provide stronger evidence of the specific interaction of our molecules with MFN2, we have performed the following experiments: we used Microscale Thermophoresis (MST) technique to quantitatively measure binding to MFN2-HR2 domain and determined the K_d of MASM7 and MFI8 as 1130 nM and 7760 nM, respectively (Fig 3a). Using MST, we also measured the affinity of the corresponding peptides 367-384Gly and 398-418Gly that are expected to bind in similar regions within HR2 domain with MASM7 and MFI8 respectively, and found that these peptides also bind to the MFN2-HR2 domain with a K_d of 3026 nM for 367-384Gly and 1598 nM for 398-418Gly (Fig 3a). Furthermore, using MST we evaluated analogues of MASM7 and MFI8 and found good correlation between binding to the HR2 domain and mitochondrial morphology changes in cells based on the Mito AR (Supplementary Figures 2e,3d,4). For example, compounds MASM19 and MASM22 showed no measurable binding by MST and also no increase of Mito AR. Similarly compounds MFI22 and MFI26 showed no

measurable binding by MST and also no reduction of Mito AR in cells. Furthermore, MASM23 showed a weaker binding than MASM7 to the HR2 domain which is consistent with its reduced capacity to increase Mito AR in cells compared to the MASM7's effect. Moreover, MFI23 showed weaker binding than MFI8 to the HR2 domain which is consistent with its reduced capacity to decrease Mito AR in cells compared to the MFI8's effect.

As we mentioned in point #1, we have repeated the HSQC NMR studies of the MFN2-HR2 with MASM7 and MFI8, and this time we also included titrations with 367-384Gly and 398-418Gly peptides. We observed comparable effects in terms of cross peak broadening and shifting of specific peaks between MASM7 and 367-384Gly peptide. Of note, the same peaks are highlighted in the HSQC spectra upon titration of MASM7 and 367-384Gly (Supplementary Fig 5a,b). These chemical shift perturbations were not present upon titration of MASM7 with an MFN2-HR2 mutant that is expected to disrupt the binding of MASM7 (Supplementary Fig. 5c), based on the pharmacophore design. Such result is consistent with the reduced capacity of MASM7 to increase Mito AR in the presence of this MFN2 mutant. Likewise, we observed comparable effects in terms of cross peak broadening and shifting of distinct peaks between MFI8 and 398-418Gly peptide (Supplementary Fig. 6a,b). Additionally, these chemical shift perturbations were not present upon titration of MFI8 with a MFN2-HR2 mutant that is expected to disrupt the binding of MFI8, based on the pharmacophore design (Supplementary Fig. 6c). Such result is consistent with the reduced capacity of MFI8 to decrease Mito AR in the presence of this MFN2 mutant.

We agree with the reviewer that more pronounced changes of the cross peaks in the HSQC experiments may be expected based on the Kds determined by MST. This is likely affected by the NMR conditions, which require much higher concentration (50 μ M) of the MFN2-HR2 than the MST technique (125 nM) and may reduce the availability of the MFN2-HR2 binding sites due to HR2-intermolecular interference. Nevertheless, the effects observed in the HSQC spectra are specific and consistent with the binding of each compound/peptide pair. Additionally, the presence of the mutants in the NMR titrations most likely disrupt direct binding of the small molecules. Taken all together we believe that our data support both MASM7 and MFI8 binding to the HR2 domain specifically and in a distinct manner.

3. In the section of "A small molecule inhibitor of mitochondrial fusion via mitofusins", details about the interaction site discussed in the second paragraph, for example, "pi-stacking" or "hydrophobic interaction", is speculative. Given that there is no structural confirmation about the binding site, this binding site discussion in the second paragraph sounds too weak to write in the Results.

We thank the reviewer for this suggestion. We have removed the paragraph from the results.

4. In the sections of "Modulation of mitofusins activity alters mitochondrial respiration and functionality" and "Inhibition of mitofusins activity induces minority MOMP and primes cells for cell death", the authors use the word "modulation of mitofusins" or "inhibition of mitofusins" quite generally. Although the discovered compounds in this manuscript exhibit such alternations of mitochondrial respiration and functionality and relations to cell death, it is unknown whether any other inhibitors (or promoters) of mitofusins exhibit the same effect, or only the specific ones that interact with HR1 or HR2 domain and/or structurally modulate the dimerization do the same effect. The authors may need more defined discussion. Otherwise, it may sound a bit over-interpretation.

We thank the reviewer for raising this point. To the best of our knowledge, there are no other reports that investigated the effect of mitofusin activators on mitochondrial respiration. Hence, our findings that concern the effect of mitofusin activators on mitochondrial respiration are novel. It is

also noteworthy that mitofusin activators do not have any effect on cell death priming (Fig. 5f, Supplementary Fig. 11b).

Regarding the mitofusin inhibitor, our manuscript reports the first small molecule that inhibits mitofusin's fusogenic activity. Thus, our findings that concern the mitofusin inhibitor-induced alterations in mitochondrial functionality (e.g. respiration) and cell death priming (e.g. minority MOMP, caspase activation) are also novel. As such, it does not have any other inhibitors to which to compare.

Minor

(1) In the abstract, spelling out SMAC.

We corrected this.

(2) It is ideal to cite references in figure captions for panels that were previously found. For example, cartoons in Figure 1a.

We corrected this.

(3) Figure 4a and 4c need markers for Oligo, CCCP etc. Otherwise, it is unclear what caused the OCR changes.

We corrected this.

Reviewer #4 (Remarks to the Author):

This manuscript describes novel and exciting results on discovery and characterization of small molecules capable of regulating directly either positively or negatively MFN1/2 activity and subsequently mitochondrial fusion by manipulating the MFN1/2 tethering permissive structure. These compounds, will allow a more precise evaluation of the role of MFN1/2 in the pathogenesis of several diseases, thus offering novel pharmacological opportunities. Generally, the authors are providing convincing demonstration of the target engagement using in vitro biochemical and biophysical assays as well as functional assays on cellular level. However, there are a few areas in which the execution of experiments falls short of providing robust conclusions and need improvements before accepting for publication.

We really appreciate the reviewer's positive remarks about our findings and suggestions on how to improve the quality of our manuscript. We have performed several experiments based on the reviewer's suggestions and hope that the reviewer will find our revised manuscript appropriate for publication.

The provided SAR studies for the reported activator (MASM7) and inhibitor (MF18) are limited. Although the activity of the synthesized analogues is supporting the generated pharmacophore model to some level, further validation is needed. In particular the HSQC NMR binding studies should be extended with additional mutant proteins based on the pharmacophore models (e.g. L692A which is also used in reconstituted MEFs) as well as by testing several analogs to further validate the importance of the certain functional moieties. In addition, the chemical shifts presented in the HSQC NMR spectra are not very significant. It will be important authors to perform quantitative analysis of the protein-ligand interactions, especially for MASM7 which has EC50 of 75 nM. Furthermore, knowing that MFN1 and 2 are highly homologues proteins, there is lack of discussion/studies to clarify the level of selectivity of the reported compounds. This is

important to be known since MFNs have mitochondrial and other functional roles, beyond mediating mitochondrial fusion.

We thank the reviewer for his/her suggestions. To address the comment regarding the quantitative analysis of the specific interaction of our molecules with MFN2, we used Microscale Thermophoresis (MST), a technique that allowed us to quantitatively measure the binding of our molecules to the MFN2-HR2 domain, and determined the K_d of MASM7 and MFI8 to be 1130 nM and 7760 nM, respectively (Fig 3a). Moreover, to address the comments regarding the validation of our pharmacophore models and expansion of our SAR studies, we used MST to evaluate the binding of MASM7 and MFI8 analogues to the MFN2-HR2 domain. We found a good correlation between the binding studies to the MFN2-HR2 domain in vitro and mitochondrial morphology changes in cells based on Mito AR (Supplementary Figures 2e,3d,4). For example, MASM19 and MASM22 showed no measurable binding by MST and also no increase of Mito AR. Similarly compounds MFI22 and MFI26 showed no measurable binding by MST and also no reduction of Mito AR in cells. Furthermore, MASM23 showed a weaker binding than MASM7 to the HR2 domain, which is consistent with its reduced capacity to increase Mito AR in cells compared to the MASM7's effect. Moreover, MFI23 showed a weaker binding than MFI8 to the HR2 domain, which is consistent with its reduced capacity to decrease Mito AR in cells compared to the MFI8's effect. Notably, we also used MST to measure the affinity of the corresponding peptides 367-384Gly and 398-418Gly that are expected to bind in similar sites within HR2 domain with MASM7 and MFI8, respectively, and found that these peptides also bind to the MFN2 HR2 domain with a K_d of 3026 nM for 367-384Gly and of 1598 nM for 398-418Gly (Fig 3a). Regarding the comments about the NMR binding studies, we have addressed similar comments from reviewer #3 in points 1 and 2. Taken all together our data from MST, NMR and functional studies with mutagenesis, support both MASM7 and MFI8 binding to the HR2 domain specifically and in a distinct manner.

We agree with the reviewer that there is a lack of discussion regarding the selectivity of our molecules towards MFN1 and MFN2. Our data suggest that MASM7 can activate both MFN1 and MFN2, while MFI8 can inhibit both MFN1 and MFN2 (Fig. 1i, 2g). Thus, neither of our molecules is selective towards MFN2 or MFN1. We now include this in our discussion of our revised manuscript (page 20):

“Our data indicated that MASM7 can activate both MFN2 and MFN1, while MFI8 can inhibit both MFN2 and MFN1 (Fig. 1i, 2g). This can be attributed to the high sequence homology between MFN1 and MFN2, and the conservation of the residues that are located in the binding region of each small molecule between MFN1 and MFN2.”

MFI8 was discovered by virtual screening and pharmacophore model developed based on the reported helical peptide from the MFN2. Thus, comparative studies and testing the peptide and MFI8 in parallel will provide further validation of the generated pharmacophore model (for example comparison of their HSQC NMR spectra), and better understanding of the relative potency of this novel small-molecule inhibitor.

We agree with the reviewer. Thus, we now provide a new set of experiments, where we compare our small molecules with the corresponding peptide that was used for the generation of their pharmacophore model (MFI8 vs 398-418Gly; MASM7 vs 367-384 Gly) in the following assays: a) in vitro binding to the HR2 domain using MST as a readout (Fig. 3a); b) in mitochondrial morphology using mitochondrial aspect ratio (Mito AR) as a readout (Fig. 1h,2f); c) HSQC NMR studies (Supplementary Fig. 5,6). Briefly, both small molecules had a comparable potency with their corresponding peptides in the aforementioned assays.

Throughout the manuscript all presented data are obtained by using a single concentration of the agonist (1 μ M) and inhibitor (20 μ M). It is necessary to be demonstrated if these effects are dose-dependent. Thus, certain binding and functional key experiments should be repeated by using concentration range of the small-molecules to show dose-dependent response.

That's a great suggestion. We now provide additional experiments that show a dose dependent effect for the following key experiments: a) effect of MASM7 and MFI8 on the mitochondrial membrane potential (Fig. 4e,f); b) effect of MFI8 in caspase 3/7 activation (Supplementary Fig. 11a); c) in vitro binding experiment of MASM7 and MFI8 on the HR2 domain using MST as a readout (Fig. 3a). Furthermore, we already have experiments that show a dose dependent effect of: a) MASM7 and MFI8 in mitochondrial morphology using mitochondrial aspect ratio (Mito AR) as a readout (Fig. 1h,2f); b) MFI8 in inducing cytochrome C release (Fig. 5d).

The results from the combination treatment with Smac bivalent mimetic are very interesting. The authors should provide rational why they tested XIAP inhibitor for the synergistic studies and not with other anti-apoptotic inhibitors, for example Bcl-2 family inhibitors.

We thank the reviewer for the positive remarks of our work that concerns the combination of XIAP inhibitor with MFI8. Our findings showed that MFI8 can induce minority MOMP, cytochrome c release and caspase 3/7 activation in an APAF-1 dependent manner. These results suggest that MFI8 can induce caspase activation, albeit at sub-lethal level, through apoptosome formation. XIAP inhibitors constitute a unique class of drugs that operate downstream of MOMP and cytochrome C release and do not require apoptosome formation to activate caspases 3/7 and subsequently induce cell death. With this element in mind, we posited that MFI8 through the induction of minority MOMP and subsequent formation of the apoptosome will lower the threshold for the activation of caspases 3/7 and subsequently apoptosis in response to XIAP inhibition by a Smac mimetic. Indeed, MFI8 was not capable of potentiating the capacity of the XIAP inhibitor to induce cell death when APAF-1 was knocked out (Fig. 6e). This result comes in agreement with our hypothesis that activation of early apoptotic events by MFI8, such as apoptosome formation, can potentiate the capacity of XIAP inhibitors to induce apoptosis that operate on later apoptotic events. On the contrary, Bcl-2 inhibitors induce apoptosis by inducing majority MOMP, which leads to caspase 3/7 activation at a lethal level through the apoptosome formation. Given that Bcl-2 inhibitors induce majority MOMP and MFI8 induces minority MOMP, we posited that the higher capacity of the Bcl-2 inhibitors to induce MOMP and subsequently activate apoptosis will minimize the effect of MFI8 in lowering the apoptotic threshold by activating the same upstream events of the apoptotic pathway. We have included an explanation for the choice of the XIAP inhibitor in our discussion section of the manuscript (page 22).

In the discussion the authors are commenting the opportunity for the development of MFN2 agonists as novel therapeutics for CMT2A. They should extend the discussion to cover the reported mutations in CMT2A and how are related to the pharmacophore/binding model of the reported agonist.

This is a great suggestion. We now extend our discussion in our revised manuscript to cover CMT2A mutations that correlate with our pharmacophore model or the binding region of the MFN2 agonists (page 23).

“Several mutations on the GTPase, HR1 and HR2 domain of the Mfn2 have been identified from patient samples and correlated with the development of the CMT2A disease. Interestingly, M376, a residue that was used for the development of the pharmacophore model for the discovery of the MFN2 activators, has been found to be mutated in patient samples to Val^{23,51}. Moreover,

Leu724, a residue that is located in the binding region of the MFN2 activators, has been found to be mutated in patient samples to Pro²³. These patient mutations highlight the importance of those residues in HR1 and HR2 domains for the function of the protein.”

REVIEWER COMMENTS

Reviewer #2 (Remarks to the Author):

I do thank the authors for their careful review and the number of analyses they did to complement their work.

Concerning the MD methodological part, no information about the Xray structure is given, nor missing residues or water box size. This is the minimum in order to drive the reader. In addition, the authors only focus on the main helices or binding site, could the authors provide information showing that the simulation and the system are converged, stabilized after 250 ns.

Many questions were are still arising from the pharmacophores and the binding site, could the authors at least run in replicate MD simulations (50 for 100 ns for instance) to ensure the their hits are staying in the same binding site ? Similarly running MD simulation with mutational points could also serve as validation.

Lastly, after a second reading, the writing part is so clear and seems to be dedicated to a highly specific audience which is clearly not the case of Nature Communications. The authors should consider such aspects before submitting/writing their article.

After a second reading and looking at the second version of the manuscript I am not willing to accept the article to be published in Nature Communications.

Reviewer #3 (Remarks to the Author):

The authors answered to my questions and revised manuscripts adequately.

if the authors will do a minor revision again, I suggest them to change the number of digits shown in the Figure 3a (the newly added ones). I suggest to round to significant digits to two or so. I can not believe the fit curve has a significance of four digits. in other word, change 3028 nM to 3.0 μ M. Also, if possible add uncertainty of the KDs.

Reviewer #4 (Remarks to the Author):

The authors have addressed all reviewer's comments in particular performing additional experiments and thus strengthening the reported findings. New reported results from MST (Figure 3a and Suppl. Fig. 4) and HSQC NMR, using wild type and mutants of MFN2-HR2 domain, are providing conclusive evidence for direct binding and specific interactions of MASM7 and MF18 with the HR2 domain. Importantly the authors have evaluated and compared the direct binding of the new mitofusins modulators with previously reported helical peptides, 367-384Gly and 398-418Gly, confirming that they bind with similar binding affinity and interact with similar residues. The rigorous of the study is also improved by providing additional results for the dose-dependent effect of MASM7 and MF18 (Fig. 4e and 4f; and Suppl. Fig. 11a). The extended discussion is also improving the quality of the revised manuscript. Thus, I am finding the revised manuscript suitable for publication and I am supporting its acceptance.

Reviewer #5 (Remarks to the Author):

NCOMMS-21-16647A

Concerns:

These compounds were selected based on a pharmacore hypothesis developed from predicted structures of mitofusin (Fig 1C). While the overall shape may be conserved, the specific side-chain interactions and solvent exposure domains may not be represented in the model. Indeed, in the available crystal structure of truncated MFN2, 706-757 is one of four helices in the extended helical bundle adjacent to the globular GTPase domain and the side chain interactions in the crystal differ from those predicted in the structural model. Therefore, rigorous validation of candidates would be required.

The specificity of each molecule for mitofusin was determined primarily with a fragment predicted to represent HR2 of Mfn2 (residues 678 – 757). This fragment also may not have the same attributes as it would in the context of the full-length protein (fold, available side chains for interactions, solubility). Therefore, this fragment is not an ideal tool to determine specificity.

The NMR peak broadening and shifting of MFN2-HR2 in the presence of either MASM7 or MFI8 is very subtle and therefore it is difficult to conclude that there is specific binding. ITC would be a more conventional approach that would provide not only affinity but the thermodynamics of binding.

The relative expression level of MFN2 WT and the alanine substitution variants in MFN1/MFN2 DKO cells should be quantified by Western blot. Differences in expression level could affect interpretation of results. Importantly, it appears that D725A/L727A is non-functional as mitochondria are fragmented in cells expressing this protein. Therefore, negative data from this construct is hard to interpret.

The increased T_m for MFN2 observed in the presence of MASM7 was partially validated by showing that the same stabilization was not observed for D725A/L727A. The non-functional double alanine substitution is not the best control as it is non-functional. Preference would be to use MFN2-L727A.

The analysis of mitochondrial function revealed that the small molecules regulate respiration and gene expression. This is interesting but does not provide rigorous support of specificity for mitofusin proteins. Indeed, it could be that primary changes to respiration increase mitochondrial membrane potential and that increases mitochondrial fusion in cells. Using MFN1/MFN2 DKO cells would not control for this because there cannot be fusion in the absence of both mitofusins. A more appropriate control would be cells that possess a partial loss-of-function variant of mitofusin whose activity would be augmented by the small molecule.

Given that MFI8 induced mitochondrial fragmentation and loss of membrane potential, it is not surprising that another effect is activation of MOMP and caspases. The protection of MFN1-MFN2 DKO cells is difficult to interpret as these cells have adapted to the presence of dysfunctional mitochondria with low membrane potential. This also fails to provide rigorous validate that mitofusin is the only target of MFI8.

Reviewer #2 (Remarks to the Author):

I do thank the authors for their careful review and the number of analyses they did to complement their work.

We would like to thank the reviewer for his/her positive remarks on our work and suggestions on how to improve the quality of our manuscript. We have now performed several *in silico* experiments based on the reviewer's suggestions and hope that the reviewer will find our revised manuscript suitable for publication.

Concerning the MD methodological part, no information about the X-ray structure is given, nor missing residues or water box size. This is the minimum in order to drive the reader. In addition, the authors only focus on the main helices or binding site, could the authors provide information showing that the simulation and the system are converged, stabilized after 250 ns.

The information about the X-ray crystal structure that was used for the construction of the full-length MFN2 structural model was given in the methods section: "Structural model of MFN2" preceding the "Molecular Dynamics Simulations". We apologize if this was not clear previously. We have now included the information about the X-ray crystal structure also in the new methods section "Molecular Dynamics Simulations and Molecular Docking" (page 27).

We also apologize for the missing details in the methods for the Molecular Dynamics simulations. We have now included additional experimental details in the "Molecular Dynamics Simulations and Molecular Docking" section including the water box size (page 27). We included a figure describing the water box size, total number of atoms, including residues of protein model and solvent atoms (Supplementary Fig. 1a). We also provided a figure of the RMSD of the system over the course of the MD simulations that shows the system is converged and stabilized (Supplementary Fig. 1b).

Many questions were still arising from the pharmacophores and the binding site, could the authors at least run in replicate MD simulations (50 for 100 ns for instance) to ensure their hits are staying in the same binding site? Similarly running MD simulation with mutational points could also serve as validation.

We followed reviewers' suggestion to provide further validation of the pharmacophore model and binding site with docking and molecular dynamics simulations beyond the experimental validation. Firstly, we have performed induced fit docking for both MASM7 and MF18 in the HR2 binding regions of our full-length MFN2 structural model that were predicted from their respective pharmacophore model (Supplementary Fig. 4a,6a). Both compounds were found to interact with the same residues of the HR2 domain as the residues of the HR1 domain that were used for the generation of each pharmacophore model (Fig. 1c,1d,2a,2b, Supplementary Fig. 4a,6a). Secondly, we have used these top-scoring poses as a starting point to perform MD simulations with our MFN2 structural model when complexed with MASM7 or MF18. The MD analysis showed that both MASM7 and MF18 were able to retain the interactions with the predicted residues of the HR2 domain during the simulation time (Supplementary Fig. 4d,6d).

Moreover, we performed similar MD simulations using MASM22 and MF125, analogues of MASM7 and MF18 that are not able to bind to the MFN2-HR2 domain (Supplementary Fig. 7) and

alter mitochondrial aspect ratio (Supplementary Fig. 3e,5d), respectively. It is noteworthy that those two analogues were predicted to have poor fitting to our pharmacophore models, because they harbor structural alterations that are not compatible with our pharmacophore models. For example, the cyclopropane of MASM7 is substituted with a bulky benzyl ring in MASM22 and our previous SAR analysis showed that MASM7 cannot accommodate bulky substituents at this position (Supplementary Fig. 3e,7). Whereas MFI25 lacks the two methyl groups of the benzyl ring (ring B) that were predicted to mimic the hydrophobic interactions of the pharmacophore model (Fig. 2b, Supplementary Fig. 5b,6b). Interestingly, we observed that neither MASM22 nor MFI25 were able to form or retain all the interactions with the predicted residues of the HR2 domain as MASM7 and MFI8 did, respectively (Supplementary Fig. 4b,4d,6b,6d). Consistently, MASM22 had a higher RMSD than MASM7 and MFI25 had a higher RMSD than MFI8, suggesting that both MASM22 and MFI25 present higher mobility and less interactions overall with MFN2 during our MD simulations (Supplementary Fig. 4e,6e).

Finally, we have mutated the residues of the HR2 domain that were predicted to interact with our hits (MASM7: D725A/L727A; MFI8: S685A) and performed MD simulations with MASM7 and MFI8. We observed that these mutations significantly reduced the capacity of each small molecule to interact with the HR2 domain in our MD simulations (Supplementary Fig. 4c,4d,6c,6d).

Taken all together, we now provide new set of data from a series of in silico experiments that validate the pharmacophore model and the binding site for both MASM7 and MFI8. Importantly, our in silico data come in agreement with our experimental data that show both of our molecules bind specifically to the HR2 domain of MFN2 in vitro using MST and protein NMR as a readout (Fig. 3a, Supplementary Fig. 8,9) and in cellulo using CETSA and mitochondrial fusion as phenotypic readout (Fig. 3b,3c,3f,3g, Supplementary Fig. 11b-c).

Lastly, after a second reading, the writing part is so clear and seems to be dedicated to a highly specific audience which is clearly not the case of Nature Communications. The authors should consider such aspects before submitting/writing their article.

We thank the reviewer for suggesting to improve clarity, although the reviewer in the previous round of review wrote "The manuscript is really well written". We have provided several clarifications and improvements in the text in the previous and current round of review, as well as additional data to support our conclusions, so we hope that the reviewer now finds the writing part suitable.

We also find it difficult to understand how our manuscript is being described as having been dedicated to a highly specific audience when it includes experiments and methods from several disciplines and expertise such as mitochondrial dynamics, metabolism, cell death, biochemical structure and function, chemical biology and computational drug design. The fact that 4 reviewers with very different expertise were required to review this manuscript suggests the opposite, and that our manuscript will attract interest from a broad audience that aligns very well with Nature Communications.

After a second reading and looking at the second version of the manuscript I am not willing to accept the article to be published in Nature Communications.

Reviewer #3 (Remarks to the Author):

The authors answered to my questions and revised manuscripts adequately.

if the authors will do a minor revision again, I suggest them to change the number of digits shown in the Figure 3a (the newly added ones). I suggest to round to significant digits to two or so. I can not believe the fit curve has a significance of four digits. in other word, change 3028 nM to 3.0 μ M. Also, if possible add uncertainty of the KDs.

We truly appreciate the reviewer's positive remarks about our revised manuscript. That's a great suggestion, we have now amended our manuscript according to the reviewer's suggestion.

Reviewer #4 (Remarks to the Author):

The authors have addressed all reviewer's comments in particular performing additional experiments and thus strengthening the reported findings. New reported results from MST (Figure 3a and Suppl. Fig. 4) and HSQC NMR, using wild type and mutants of MFN2-HR2 domain, are providing conclusive evidence for direct binding and specific interactions of MASM7 and MF18 with the HR2 domain. Importantly the authors have evaluated and compared the direct binding of the new mitofusins modulators with previously reported helical peptides, 367-384Gly and 398-418Gly, confirming that they bind with similar binding affinity and interact with similar residues. The rigorous of the study is also improved by providing additional results for the dose-dependent effect of MASM7 and MF18 (Fig. 4e and 4f; and Suppl. Fig. 11a). The extended discussion is also improving the quality of the revised manuscript. Thus, I am finding the revised manuscript suitable for publication and I am supporting its acceptance.

We would like to thank the reviewer for his/her positive remarks about our revised manuscript and for supporting its acceptance.

Reviewer #5 (Remarks to the Author):

NCOMMS-21-16647A

We really appreciate the reviewer's careful analysis of our work. We have now performed several experiments based on the reviewer's suggestions and hope that the reviewer will find our revised manuscript suitable for publication.

Concerns:

These compounds were selected based on a pharmacore hypothesis developed from predicted structures of mitofusin (Fig 1C). While the overall shape may be conserved, the specific side-chain interactions and solvent exposure domains may not be represented in the model. Indeed, in the available crystal structure of truncated MFN2, 706-757 is one of four helices in the extended helical bundle adjacent to the globular GTPase domain and the side chain interactions in the crystal differ from those predicted in the structural model. Therefore, rigorous validation of candidates would be required.

Firstly, we wish to clarify that our full-length structural model was generated using the I-TASSER approach for structure prediction as we previously reported (Franco et al. Nature 2016) considering homologues structures such as the inactive conformation of bacterial dynamin-like protein (PDB: 2J69) and the truncated human MFN2 (PDB ID: 6JFL). Importantly, the available

truncated crystal structure of MFN2 is an engineered structure (to make a construct amenable for crystallization) that misses a significant portion of the HR1 and HR2 domains and it has a linker between HR1 and HR2 to promote the interaction of a partial HR2 domain with a partial HR1 domain (Li et al. 2019). However, this linker could affect the relevant orientations and the interactions between the two partial HR1 and HR2 domains in the crystal structure. Notably, this truncated MFN2 crystal structure omits other parts of the full length MFN2 structure (residues 400-706) including a transmembrane domain and the HR1 and HR2 domain interactions that were used in our pharmacophore modeling for the mitofusin inhibitor. Therefore, we could not use the truncated MFN2 crystal structure for the design of the pharmacophore model of our mitofusin inhibitor. Moreover, the truncated crystal structure of MFN2 has a linker that links the partial HR1 domain with the partial HR2 domain at a position that is very close to our pharmacophore model residues for the mitofusin activator. The interacting side chains in our model are not aligned with the truncated MFN2 structure. This can be the result of the engineering approach, specifically the introduction of the linker along with the truncation sites of the HR1 and HR2 domain may slightly alter the topology for the HR1 and HR2 domains and subsequently the interactions between those two domains.

Our previous structure-function study with a full-length structural model of MFN2 (Franco et al. 2016) proposed a conformation for MFN2 that is similar to the bacterial dynamin-like protein. Such conformation enabled the design of an activator (367-384 Gly) and inhibitor (398-418 Gly) mini-peptide of MFN2 that derived from different regions of the HR1 domain and interact with different regions of the HR2 domain. We previously showed that the activator peptide promoted activation of MFN2 and mitochondrial fusion, while the inhibitor peptide inhibited activation of MFN2 and mitochondrial fusion. Notably, the activator peptide includes the residues that were used in our pharmacophore model for the identification of our small molecule activator (MASM7), while the inhibitor peptide includes the residues that were used in our pharmacophore model for the identification of our small molecule inhibitor (MF18). In our current manuscript we used protein NMR and MST to show that the activator peptide and the inhibitor peptide have similar binding characteristics with our small molecule activator and inhibitor towards the HR2 domain of MFN2, respectively, (Fig 3a, Supplementary Fig. 8,9). Importantly, we now provide a new set of data from molecular dynamic simulations of our MFN2 structural model complexed with MASM7 or MF18 where we show that both small molecules interact with the predicted residues on the HR2 domain from their respective pharmacophore models (Supplementary Fig. 4,6). Moreover, we used site-directed mutagenesis on the HR2 domain in *in vitro* binding experiments (Supplementary Fig. 8,9) and in cellulo protein engagement experiments (CETSA: Supplementary Fig. 11b-d) to provide evidence that our small molecules interact specifically with their respective region of the HR2 domain as predicted by the pharmacophore model for each small molecule. We have also used site-directed mutagenesis in several *in cellulo* functional experiments (mitochondrial fusion: Fig. 3b,c; mitochondrial membrane potential: Supplementary Fig. 13c,g) to provide further support that the effects of our small molecules on mitochondrial morphology and function are specific to MFN2. Notably, we used structure activity relationships (SAR) for both of our molecules in binding experiments *in vitro* (MST: Supplementary Fig.7) and in *in cellulo* functional experiments (mitochondrial fusion: Supplementary Fig. 3d,e and Supplementary Fig. 5c,d) to validate our pharmacophore model and provide further support about the specificity of our small molecules towards MFN2 using a chemical approach. To conclude, we believe that all these experiments provide a rigorous validation for our candidates and the design of our pharmacophore model.

The specificity of each molecule for mitofusin was determined primarily with a fragment predicted to represent HR2 of Mfn2 (residues 678 – 757). This fragment also may not have the

same attributes as it would in the context of the full-length protein (fold, available side chains for interactions, solubility). Therefore, this fragment is not an ideal tool to determine specificity.

We can see the point of the reviewer about the HR2 fragment of MFN2. To the best of our knowledge there are no reports that describe the expression of the full length recombinant MFN2. However, what has been expressed is an engineered and truncated form of MFN2 that is missing many critical residues of HR1 and HR2 domains and the transmembrane domain (residues 400-708) (Cao et al Nature 2017, Li et al. Nat. Commun 2019). Of note, the truncated form of MFN2 that has been expressed before misses the part of the HR2 domain (residues 400-418) where our small molecule inhibitor binds to. We had tried in the past to express the full length of MFN2 but we were not successful as the full length protein was prone to aggregation. Thus, we chose to express the HR2 domain of MFN2 in vitro to assess the binding of our small molecules as this is where the predicted binding site locates based on our pharmacophore model. Furthermore, we used site-directed mutagenesis on selected residues of the MFN2-HR2 domain that were predicted to disrupt the binding of our small molecules based on the pharmacophore model for each compound and showed that both MASM7 and MF18 were not able to bind to the MFN2-HR2 domain mutants in vitro (Supplementary Fig. 8,9). Such results suggest that MASM7 and MF18 bind specifically to the WT MFN2-HR2 domain in vitro. Importantly, we used the same strategy of site-directed mutagenesis in cells that expressed the full length of MFN2 harboring mutations in the same residues of the HR2 domain that were used for the in vitro binding experiments and monitored mitochondrial fusion upon treatment with our compounds (Fig. 3b,c). Our results showed that MASM7 was not able to promote mitochondrial fusion in cells that expressed the MFN2 mutants but only in cells that expressed the WT MFN2 (Fig. 3b). Likewise, MF18 was not able to inhibit mitochondrial fusion in cells that expressed the MFN2 mutants but only in cells that expressed the WT MFN2 (Fig. 3c). These results, which come in agreement with our results from the in vitro binding experiments where the fragment of the HR2 domain was used and further support that our small molecules interact specifically with the HR2 domain of the full length of MFN2 in cells.

Moreover, we have also shown in vitro by NMR (Supplementary Fig. 8,9) and MST (Fig. 3a) that both HR1-derived mini peptides: 367-384 Gly and 398-418 Gly bind to the MFN2-HR2 domain fragment. The aforementioned peptides are derived from different regions of the HR1 domain and bind to the same regions on the HR2 domain as our small molecules (MASM7 and 367-384 Gly; MF18 and 398-418 Gly). Importantly, the HR1 domain interacts intra-molecularly with the HR2 domain in the full length of MFN2 (Franco et al Nature 2016). Taken all together, our data suggest that our recombinant HR2-domain fragment maintains the molecular properties of the HR2 domain of the full length of MFN2 at least for the binding regions of our small molecules, as interactions of fragments of the HR1 domain with our recombinant HR2 domain can be observed (Fig. 3a and Supplementary Fig. 8,9).

The NMR peak broadening and shifting of MFN2-HR2 in the presence of either MASM7 or MF18 is very subtle and therefore it is difficult to conclude that there is specific binding. ITC would be a more conventional approach that would provide not only affinity but the thermodynamics of binding.

We understand the reviewer's concern, but this comment has been addressed before (see response to reviewers #3 and 4). Additionally, we have already provided a new set of data in the revised version of our manuscript using MST (Fig. 3a) that address the concerns of the reviewers regarding binding affinity and specificity. Our MST data showed clearly that both MASM7 and MF18 bind to the MFN2-HR2 domain, which agrees with our NMR data (Supplementary Fig. 8,9).

Importantly, we have also used mutagenesis (genetic approach) in protein NMR experiment (Supplementary Fig. 8,9) and structure activity relationship (SAR) (chemical approach) in MST studies (Supplementary Fig. 7) to support that our molecules bind specifically to the MFN2 HR2 domain. Taken all together, we have used two distinct biophysical methods (protein NMR and MST) to provide evidence that our molecules bind to the MFN2-HR2 domain (same construct in both methods), as well as a genetic and a chemical approach to provide evidence that our molecules bind in a specific fashion to the MFN2-HR2 domain. Thus, we believe that we have already addressed satisfactorily the binding of our small molecules to HR2 domain of MFN2.

The relative expression level of MFN2 WT and the alanine substitution variants in MFN1/MFN2 DKO cells should be quantified by Western blot. Differences in expression level could affect interpretation of results. Importantly, it appears that D725A/L727A is non-functional as mitochondria are fragmented in cells expressing this protein. Therefore, negative data from this construct is hard to interpret.

That's a great suggestion. We now show that the alanine substitution variants of MFN2 were expressed at similar levels in MFN1/MFN2 DKO MEFs (Supplementary Fig. 10c).

The increased T_m for MFN2 observed in the presence of MASM7 was partially validated by showing that the same stabilization was not observed for D725A/L727A. The non-functional double alanine substitution is not the best control as it is non-functional. Preference would be to use MFN2-L727A.

We thank the reviewer for this suggestion. To address this comment, we now provide a new set of data where we show that MASM7 did not alter the T_m of the L727A MFN2 mutant (Supplementary Fig. 11d). Importantly, this data come in agreement with our previous data showing MASM7 was not able to promote mitochondrial fusion in cells that express L727A MFN2 mutant (Fig. 3b). Taken all together our data suggest that MASM7 bind to specifically to the HR2 domain of MFN2 and promotes mitochondrial fusion.

The analysis of mitochondrial function revealed that the small molecules regulate respiration and gene expression. This is interesting but does not provide rigorous support of specificity for mitofusin proteins. Indeed, it could be that primary changes to respiration increase mitochondrial membrane potential and that increases mitochondrial fusion in cells. Using MFN1/MFN2 DKO cells would not control for this because there cannot be fusion in the absence of both mitofusins. A more appropriate control would be cells that possess a partial loss-of-function variant of mitofusin whose activity would be augmented by the small molecule.

We thank the reviewer for this interesting suggestion. To the best of our knowledge, mutations on MFN2 can either be loss of function mutations (e.g. CMT2A MFN2 variants) or have no effect in the function of the protein. Thus, we cannot comply with this request of the reviewer as none of the above cases fall in the partial loss of function category. In addition, in our opinion even if a mutation on MFN2 leads to a partial loss of function of the protein but it is not in the binding region of MASM7 it will not provide evidence that MASM7 acts specifically through MFN2.

Firstly, we showed that mitofusins regulate mitochondrial functionality (membrane potential and respiration) using a genetic approach:

1. Reconstitution of MFN2 in MFN1/MFN2 DKO MEFs increased membrane potential compared to the reconstitution of β -Gal (Supplementary Fig. 13b).
2. Knockout of MFN1 and MFN2 reduced respiration in cells (Fig. 4a,b).

Given that mitofusins reside on the outer mitochondrial membrane and their primary function is to tether the outer mitochondrial membranes from adjacent mitochondria, our data support that mitofusins positively regulate mitochondrial function by firstly regulating mitochondrial fusion. Secondly, our previous data showed that both MASM7 and MFI8 bind specifically to the HR2 domain of MFN2 and can modulate the activity of both MFN1 and MFN2 (Fig. 1i,2g,3a,3f,3g Supplementary Fig. 8,9,11b-d). With these two elements in mind, we decided to use MFN1/MFN2 DKO cells to investigate if the effects of our molecules in mitochondrial function are mitofusin dependent/specific. Indeed, our data showed that neither MASM7 nor MFI8 was able to alter membrane potential and respiration in the absence of mitofusins (Fig. 4a-d, Supplementary Fig. 14d). Thus, we placed in our model the effects of our small molecules on mitochondrial function downstream of mitofusin activity and mitochondrial fusion (Fig. 7).

In contrast to our model, the reviewer here suggests an alternative explanation in which MASM7 increases mitochondrial fusion in cells because it firstly increases respiration and membrane potential. This idea places mitochondrial fusion downstream of respiration and membrane potential. With this element in mind, if MASM7 had off-target effects that affect primarily respiration we should have seen an increase in respiration even in MFN1/MFN2 DKO cells, as the lack of mitochondrial fusion in MFN1/MFN2 DKO cells will be redundant for the MASM7 induced increase in respiration. In stark contrast with this idea, our results showed that MASM7 cannot increase respiration in MFN1/MFN2 DKO cells, which suggests that MFN1/2 are required for MASM7-induced increase in respiration (Figure 4a,b). Importantly, MFN1/MFN2 DKO cells responded to the mitochondrial stress treatments (e.g. oligomycin, FCCP) suggesting that mitochondria from those cells are still able to produce ATP using OXPHOS and respiration (Figure 4a,c). To further support the specificity of our molecule, we have already shown that the L727A mutation on the HR2 domain of MFN2 reduces the capacity of MASM7 to increase membrane potential in cells in comparison to the WT MFN2 (Supplementary Fig. 13c). Such result comes in agreement with our data that showed that this mutation also reduced the capacity of MASM7 to increase fusion (Fig. 3b) and the T_m of MFN2 in the CETSA experiment (Supplementary Fig. 11d). Importantly, this MFN2 mutant did not alter mitochondrial fusion in comparison to the WT MFN2 in the absence of MASM7 (Fig. 3b).

Furthermore, we now provide new data showing that MASM7 was not able to increase mitochondrial membrane potential in isolated mitochondria in vitro in conditions that mitochondrial fusion cannot occur (Supplementary Fig. 13d). Such result demonstrates that mitochondrial fusion is required for MASM7-induced increase in membrane potential and rules out any other off-target effects of MASM7 that can directly increase membrane potential. Taken all together our data suggest that MASM7 requires MFN1/2 and functional mitochondrial fusion to increase in membrane potential and respiration.

Given that MFI8 induced mitochondrial fragmentation and loss of membrane potential, it is not surprising that another effect is activation of MOMP and caspases. The protection of MFN1-MFN2 DKO cells is difficult to interpret as these cells have adapted to the presence of dysfunctional mitochondria with low membrane potential. This also fails to provide rigorous validate that mitofusin is the only target of MFI8.

We understand reviewer's concern, but we wish to point out that previously reviewer #1 suggested to us in the previous round of revision to test if MFI8 can induce cytochrome c release in the

MFN1/MFN2 DKO MEFs. The rationale behind this experiment was to use MFN1/MFN2 DKO MEFs as a control cell line to validate if MFI8-induced MOMP and cytochrome c release in WT MEFs is attributed to non-specific/off target effects of MFI8 or if it is specific to the inhibition of mitofusins. We found this request from reviewer #1 very reasonable, thus we proceeded and did the experiment. Importantly, our results showed that MFI8 treatment to MFN1/MFN2 DKO MEFs was not able to induce cytochrome c-release, suggesting that this effect is specific through the inhibition of mitofusins (Supplementary Fig. 14c).

To support that mitochondria from MFN1/MFN2 DKO cells can undergo MOMP and release cytochrome c we previously used the treatment of ABT-737/S63845 as a positive control in our experiment (Supplementary Fig. 14c). Notably, both of ABT-737 and S63845 are established BH3 mimetics that induce MOMP and cytochrome c release by decreasing membrane potential. In line with this idea, we now provide a new set of data where we show that treatment of ABT-737/S63845 to MFN1/MFN2 DKO cells was capable of reducing mitochondrial membrane potential (Supplementary Fig. 14d). Collectively, our data support that MFN1/MFN2 DKO cells have mitochondria that can undergo MOMP and release cytochrome c because of the decrease in the membrane potential when challenged by ABT-737/S63845. Importantly, treatment of MFI8 to MFN1/MFN2 DKO cells did not decrease membrane potential (Supplementary Fig. 14d). Taken all together, our data show that MFI8 decreases membrane potential in a mitofusins dependent manner (Fig. 4f, Supplementary Fig. 14d). Therefore, the rescue in membrane potential (Supplementary Fig. 14d), cytochrome c release (Supplementary Fig. 14c) and caspase activation (Fig. 5b) that we observed in MFN1/MFN2 DKO MEFs upon MFI8 treatment is attributed to the absence of mitofusins, which is the target of MFI8, rather than the adaptation of cells to the dysfunctional mitochondria.

Moreover, we have also shown that MFI22, an analogue of MFI8 that is not able to bind to the MFN2-HR2 domain (Supplementary Fig. 7) and inhibit mitochondrial fusion (Supplementary Fig. 5c,d), was not able to induce caspase 3/7 activation in WT MEFs (Fig. 5a). These results indicate that inhibition of mitofusins activity and mitochondrial fusion is required for the induction of caspase activation.

Taken all together, we have used MFN1/MFN2 DKO MEFs (after validation that these cells can undergo MOMP and cytochrome c release) as a genetic approach and MFI22 as a chemical approach to demonstrate that MFI8 induces MOMP and caspase activation by specifically inhibiting mitofusins.

REVIEWERS' COMMENTS

Reviewer #5 (Remarks to the Author):

1. **Author response:** “Firstly, we wish to clarify that our full-length structural model was generated using the I-TASSER approach for structure prediction as we previously reported (Franco et al. Nature 2016) considering homologues structures such as the inactive conformation of bacterial dynamin-like protein (PDB: 2J69) and the truncated human MFN2 (PDB ID: 6JFL). Importantly, the available truncated crystal structure of MFN2 is an engineered structure (to make a construct amenable for crystallization) that misses a significant portion of the HR1 and HR2 domains and it has a linker between HR1 and HR2 to promote the interaction of a partial HR2 domain with a partial HR1 domain (Li et al. 2019). However, this linker could affect the relevant orientations and the interactions between the two partial HR1 and HR2 domains in the crystal structure. Notably, this truncated MFN2 crystal structure omits other parts of the full length MFN2 structure (residues 400-706) including a transmembrane domain and the HR1 and HR2 domain interactions that were used in our pharmacophore modeling for the mitofusin inhibitor. Therefore, we could not use the truncated MFN2 crystal structure for the design of the pharmacophore model of our mitofusin inhibitor. Moreover, the truncated crystal structure of MFN2 has a linker that links the partial HR1 domain with the partial HR2 domain at a position that is very close to our pharmacophore model residues for the mitofusin activator. The interacting side chains in our model are not aligned with the truncated MFN2 structure. This can be the result of the engineering approach, specifically the introduction of the linker along with the truncation sites of the HR1 and HR2 domain may slightly alter the topology for the HR1 and HR2 domains and subsequently the interactions between those two domains.”

Reviewer Response: While the truncated Mfn2 crystal structure omits a large portion of the protein, this internal deletion is consistent with what has been observed with other dynamin-related proteins in the field. It is common for the C-terminal domain to be near the N-terminal domain in the 3-dimensional structure. Nonetheless, neither the truncated crystal structure nor the I-TASSER predictions are likely to fully represent the structure of MFN2, thus making small molecule design and validation very challenging and thus the work requires rigorous validation.

2. **Author response:** “To the best of our knowledge, mutations on MFN2 can either be loss of function mutations (e.g. CMT2A MFN2 variants) or have no effect in the function of the protein. Thus, we cannot comply with this request of the reviewer as none of the above cases fall in the partial loss of function category”

Reviewer Response: In Detmer and Chan, JCB 2007, several different CMT2A variants were studied by expressing in either Mfn2-null or Mfn1/2-null cells. Most variants restored the reticular mitochondrial morphology in mfn2-null cells, suggesting some fusion activity is retained. Furthermore, expression of

Mfn1-V69F, L76P, R274Q or W740S in Mfn1/2-null cells also restored reticular mitochondrial morphology, consistent with the conclusion that these variants possess some fusion activity. More recently, Samanas et al, Life Sci Alliance 2020 reported four variants with amino acid substitutions that had reduced fusion activity, but clearly more than Mfn2-null mitochondria alone.

3. **Author response:** “Indeed, our data showed that neither MASM7 nor MFI8 was able to alter membrane potential and respiration in the absence of mitofusins (Fig. 4a-d, Supplementary Fig. 14d). Thus, we placed in our model the effects of our small molecules on mitochondrial function downstream of mitofusin activity and mitochondrial fusion.”

“With this element in mind, if MASM7 had off-target effects that affect primarily respiration we should have seen an increase in respiration even in MFN1/MFN2 DKO cells, as the lack of mitochondrial fusion in MFN1/MFN2 DKO cells will be redundant for the MASM7 induced increase in respiration.”

Reviewer Response: The Mfn1/2-null cells are severely abnormal with many organelles that lack mitochondrial membrane potential (Chen et al., JBC 2005). Indeed, it has been shown that replication of mitochondrial DNA is impaired in these cells (Silva-Ramos et al., Plos Genetics, 2019). Therefore, one would not expect to increase mitochondrial respiration in Mfn1/2-null cells, which lack the molecular components required to increase respiration. However, the increased respiration observed in wild type cells could be due to any number of changes in the complete electron transport chain components and full complement of wild type mitochondrial DNA. The two systems simply cannot be compared.

Reviewer #5 (Remarks to the Author):

1. Author response: “Firstly, we wish to clarify that our full-length structural model was generated using the I-TASSER approach for structure prediction as we previously reported (Franco et al. Nature 2016) considering homologues structures such as the inactive conformation of bacterial dynamin-like protein (PDB: 2J69) and the truncated human MFN2 (PDB ID: 6JFL). Importantly, the available truncated crystal structure of MFN2 is an engineered structure (to make a construct amenable for crystallization) that misses a significant portion of the HR1 and HR2 domains and it has a linker between HR1 and HR2 to promote the interaction of a partial HR2 domain with a partial HR1 domain (Li et al. 2019). However, this linker could affect the relevant orientations and the interactions between the two partial HR1 and HR2 domains in the crystal structure. Notably, this truncated MFN2 crystal structure omits other parts of the full length MFN2 structure (residues 400-706) including a transmembrane domain and the HR1 and HR2 domain interactions that were used in our pharmacophore modeling for the mitofusin inhibitor. Therefore, we could not use the truncated MFN2 crystal structure for the design of the pharmacophore model of our mitofusin inhibitor. Moreover, the truncated crystal structure of MFN2 has a linker that links the partial HR1 domain with the partial HR2 domain at a position that is very close to our pharmacophore model residues for the mitofusin activator. The interacting side chains in our model are not aligned with the truncated MFN2 structure. This can be the result of the engineering approach, specifically the introduction of the linker along with the truncation sites of the HR1 and HR2 domain may slightly alter the topology for the HR1 and HR2 domains and subsequently the interactions between those two domains.”

Reviewer Response: While the truncated Mfn2 crystal structure omits a large portion of the protein, this internal deletion is consistent with what has been observed with other dynamin-related proteins in the field. It is common for the C-terminal domain to be near the N-terminal domain in the 3-dimensional structure. Nonetheless, neither the truncated crystal structure nor the I-TASSER predictions are likely to fully represent the structure of MFN2, thus making small molecule design and validation very challenging and thus the work requires rigorous validation.

Author response: We agree with the reviewer that neither the truncated crystal structure nor our homology model based on available dynamin related proteins may fully represent the actual structure of the full length of MFN2. However, the reviewer’s comment was regarding a difference in the exact alignment of the residues of HR1-HR2 interactions. Our pharmacophore models are based on residues that belong to the HR1 domain for which both the model and truncated crystal structure agree that they are part of the HR1 helix. Using the coordinates of HR1 helix from our model, our in silico screening has led to the identification of an MFN1/2 agonist and antagonist, which were shown to mimic HR1 residues and modulate MFN2 activity by interacting specifically with the HR2 domain MFN2 (Fig. 1h, 2f, 3a, Supplementary Fig. 8, 9). Such outcome comes in agreement with the predictions from our in silico studies that used the I-TASSER predicted full length MFN2 model.

2. Author response: “To the best of our knowledge, mutations on MFN2 can either be loss of function mutations (e.g. CMT2A MFN2 variants) or have no effect in the function of the protein. Thus, we cannot comply with this request of the reviewer as none of the above cases fall in the partial loss of function category”

Reviewer Response: In Detmer and Chan, JCB 2007, several different CMT2A variants were studied by expressing in either Mfn2-null or Mfn1/2-null cells. Most variants restored the reticular

mitochondrial morphology in *mfn2*-null cells, suggesting some fusion activity is retained. Furthermore, expression of *Mfn1*-V69F, L76P, R274Q or W740S in *Mfn1/2*-null cells also restored reticular mitochondrial morphology, consistent with the conclusion that these variants possess some fusion activity. More recently, Samanas et al, *Life Sci Alliance* 2020 reported four variants with amino acid substitutions that had reduced fusion activity, but clearly more than *Mfn2*-null mitochondria alone.

Author response: We thank the reviewer for providing information about MFN1 and MFN2 mutants that were previously shown to restore mitochondrial fusion when reconstituted in cells that lacks MFNs. In our opinion, the choice of the MFN1/2 mutants to show that MASM7 increases respiration and membrane potential by modulating MFN1/2 activity in a specific manner should be based on the idea that those mutations should directly disrupt binding of MASM7 to the HR2 domain of MFNs and subsequently revoke the capacity of MASM7 to increase respiration and membrane potential. It is noteworthy that the aforementioned MFN1 mutants, as well as the reported MFN2 mutants (S378P, A383V, Q386P, C390F) in the Samanas article, are not in the vicinity of the region of the HR2 domain where MASM7 binds to. Thus, the use of those mutants in control experiments to show that MASM7 increases membrane potential and respiration by specifically modulating MFN1/2 activity is not considered appropriate and any rescue in the activity by MASM7 would not warranty specificity towards MFN2.

Notably we have already shown that substitution of Leu727 with Ala significantly reduced the capacity of MASM7 to promote mitochondrial fusion in cells and increase membrane potential and the T_m of MFN2 (CETSA experiment) in comparison to the WT MFN2 (Fig. 3b, Supplementary Fig. 11b, 11d, 13c). Importantly, Leu727 resides in the region of the HR2 domain of MFN2 that was predicted and later shown to affect binding of MASM7 (Fig. 1c). Moreover, the substitution of Leu727 with Ala did not alter the capacity of MFN2 to promote mitochondrial fusion at the baseline level, suggesting that the fusogenic activity of MFN2 is retained (Fig. 3b). We also have shown that MASM7 cannot increase membrane potential of isolated mitochondria *in vitro* when mitochondrial fusion cannot take place (Supplementary Fig. 13d). Taken all together, we believe that we have provided a rigorous validation for the specificity of MASM7 towards MFN1/2 and the downstream effects on mitochondrial respiration and membrane potential.

3. Author response: “Indeed, our data showed that neither MASM7 nor MFI8 was able to alter membrane potential and respiration in the absence of mitofusins (Fig. 4a-d, Supplementary Fig. 14d). Thus, we placed in our model the effects of our small molecules on mitochondrial function downstream of mitofusin activity and mitochondrial fusion.”

“With this element in mind, if MASM7 had off-target effects that affect primarily respiration we should have seen an increase in respiration even in MFN1/MFN2 DKO cells, as the lack of mitochondrial fusion in MFN1/MFN2 DKO cells will be redundant for the MASM7 induced increase in respiration.”

Reviewer Response: The *Mfn1/2*-null cells are severely abnormal with many organelles that lack mitochondrial membrane potential (Chen et al., *JBC* 2005). Indeed, it has been shown that replication of mitochondrial DNA is impaired in these cells (Silva-Ramos et al., *Plos Genetics*, 2019). Therefore, one would not expect to increase mitochondrial respiration in *Mfn1/2*-null cells, which lack the molecular components required to increase respiration. However, the increased respiration observed in wild type cells could be due to any number of changes in the complete electron transport chain components and full complement of wild type mitochondrial DNA. The two systems simply cannot be compared.

Author response: We respectfully disagree with the reviewer. Our data come in agreement with previously published data by Chen et al. showing that deletion of MFN1 and MFN2 reduced but it did not completely diminish respiration in cells (our manuscript: Fig. 4a,b; Chen et al. manuscript: Fig. 5g). Moreover, Silva-Ramos et al. showed that deletion of MFN1 and MFN2 reduced but it did not completely diminish the expression of electron transport chain (ETC) components (Silva-Ramos et al. manuscript: Fig. 1h). Importantly, treatment of ABT-737/S63845 but not MFI8 was capable of reducing membrane potential in Mfn1/Mfn2 DKO MEFs, suggesting that deletion of MFN1/2 does not diminish completely membrane potential in cells (Fig. 5h). Taken all together Mfn1/Mfn2 DKO cells express the components of the ETC and can use respiration to produce ATP, albeit to a lesser extent than cells that express both MFN1/2. With these findings, we posited that mitochondria from cells that lack both MFN1/2 retain the necessary functionality to participate in control experiments to test the specificity of our molecules towards MFNs. Nevertheless, we now comment on the reduced mitochondrial function of Mfn1/Mfn2 DKO MEFs in the results section of our manuscript (page 16).

We have also shown that MASM7 was not capable of increasing membrane potential (Supplementary Fig. 13d) and MFI8 was not capable of decreasing respiration (Fig. 5h) in isolated mitochondria in vitro in conditions that mitochondrial fusion cannot take place. These results suggest that both MASM7 and MFI8 require functional mitochondrial fusion in order to alter mitochondrial functionality and rule out any direct effects of both compounds on the ETC components that could potentially alter mitochondrial functionality (i.e. respiration, membrane potential) irrespective of MFN1/2 targeting. In line with this idea, mutagenesis on the HR2 domain of MFN2 reduced the capacity of MASM7 and MFI8 to alter membrane potential (Supplementary Fig. 13c,13g). To conclude, we have provided a rigorous validation that MASM7 and MFI8 increase and decrease mitochondrial functionality by increasing and decreasing MFN1/2' fusogenic activity, respectively.